# Leaflet Tensions Control the Spatio-Temporal Remodeling of Lipid Bilayers and Nanovesicles

**DOI:** 10.3390/biom13060926

**Published:** 2023-05-31

**Authors:** Reinhard Lipowsky, Rikhia Ghosh, Vahid Satarifard, Aparna Sreekumari, Miftakh Zamaletdinov, Bartosz Różycki, Markus Miettinen, Andrea Grafmüller

**Affiliations:** 1Max Planck Institute of Colloids and Interfaces, Science Park Golm, 14424 Potsdam, Germany; 2Icahn School of Medicine Mount Sinai, New York, NY 10029, USA; 3Yale Institute for Network Science, Yale University, New Haven, CT 06520, USA; 4Department of Physics, Indian Institute of Technology Palakkad, Palakkad 678 623, India; 5Institute of Physics, Polish Academy of Sciences, Aleja Lotnikow 32/46, 02-668 Warsaw, Poland; 6Department of Chemistry, University of Bergen, 5020 Bergen, Norway

**Keywords:** synthetic biosystems, biomembranes, simulations, molecular dynamics, bilayer tension, leaflet tensions, engulfment, endocytosis

## Abstract

Biological and biomimetic membranes are based on lipid bilayers, which consist of two monolayers or leaflets. To avoid bilayer edges, which form when the hydrophobic core of such a bilayer is exposed to the surrounding aqueous solution, a single bilayer closes up into a unilamellar vesicle, thereby separating an interior from an exterior aqueous compartment. Synthetic nanovesicles with a size below 100 nanometers, traditionally called small unilamellar vesicles, have emerged as potent platforms for the delivery of drugs and vaccines. Cellular nanovesicles of a similar size are released from almost every type of living cell. The nanovesicle morphology has been studied by electron microscopy methods but these methods are limited to a single snapshot of each vesicle. Here, we review recent results of molecular dynamics simulations, by which one can monitor and elucidate the spatio-temporal remodeling of individual bilayers and nanovesicles. We emphasize the new concept of leaflet tensions, which control the bilayers’ stability and instability, the transition rates of lipid flip-flops between the two leaflets, the shape transformations of nanovesicles, the engulfment and endocytosis of condensate droplets and rigid nanoparticles, as well as nanovesicle adhesion and fusion. To actually compute the leaflet tensions, one has to determine the bilayer’s midsurface, which represents the average position of the interface between the two leaflets. Two particularly useful methods to determine this midsurface are based on the density profile of the hydrophobic lipid chains and on the molecular volumes.

## 1. Introduction

All biological membranes contain a single lipid bilayer as their basic building block. The importance of lipids was already realized by Langmuir and others at the beginning of the 20th century, using spreading experiments of lipid monolayers. Such a technique was also used by Gorter and Grendel who extracted lipids from red blood cells [1,2]. They found that the area of the monolayer was approximately twice the area of the cell and proposed that the cell should be covered by a lipid bilayer.

This proposal was confirmed, in the 1950s and 1960s, by imaging cross-sections of biomembranes via electron microscopy, which provided direct evidence that cell membranes are based on single bilayers and showed that these bilayers have a thickness of 4 to 5 nm [3]. Electron microscopy studies also demonstrated that bilayers can be formed by a single species of phospholipid molecules [4]. Therefore, lipid bilayers consisting of one or a few lipid components have become important model systems for biological membranes.

In order to avoid bilayer edges, at which the hydrophobic core of the bilayer would be exposed to the surrounding aqueous solution, a single bilayer forms a closed unilamellar vesicle. Both synthetic and cellular nanovesicles have been experimentally studied for a long time. Synthetic nanovesicles are assembled from lipid molecules, using a variety of preparation methods [5,6], which produce a wide range of vesicle sizes. Nanovesicles with a diameter below 100 nm have emerged as promising modules for lipid nanotechnology. Indeed, these vesicles have been shown to provide potent platforms for dermal and transdermal drug delivery [7,8,9,10,11]. In particular, they have been used to treat skin diseases such as skin cancer [12] and psoriasis [13]. More recently, such nanovesicles have been applied as nanocarriers for the packaging and delivery of small interfering RNA therapeutics, used to silence various pathways of gene expression [14,15,16], and for the delivery of mRNA vaccines, which have become crucial during the COVID-19 pandemic [15,17,18].

Electron microscopy methods also represent the main experimental approach to study the morphology of nanovesicles [19,20,21,22,23]. One disadvantage of these methods is, however, that they provide only a single snapshot of each vesicle. In contrast, computer simulations with molecular resolution can reveal the molecular dynamics of the lipids and the nanoscale dynamics of the bilayers, thereby monitoring and elucidating the spatio-temporal remodeling of *individual* nanovesicles.

Here, we review recent insights obtained from such molecular dynamics simulations. In fact, the main purpose of this review is to advertize and promote the concept of leaflet tensions, a new concept that has been recently introduced and further developed by our group [24,25,26,27,28,29,30,31,32]. Leaflet tensions are important because they control many properties of lipid bilayers and nanovesicles. In particular, they determine the stability and instability of lipid bilayers, the rates of transbilayer flip-flops, the shape transformations of nanovesicles, the engulfment and endocytosis of condensate droplets and nanoparticles, as well as the adhesion and fusion of nanovesicles. Some of these processes are displayed in Figure 1.

We emphasize that one should distinguish the individual leaflet tensions from the bilayer tension, which is equal to the sum of the two leaflet tensions. To avoid membrane rupture, the osmotic conditions have to be adjusted in such a way that both cellular and synthetic membranes experience a relatively low bilayer tension. However, even for vanishing bilayer tension, the individual leaflets can still experience significant leaflet tensions if one leaflet is stretched by a positive leaflet tension whereas the other leaflet is compressed by a negative leaflet tension. In such a situation, the lipid bilayer is subject to a transbilayer stress asymmetry, corresponding to the difference between the two leaflet tensions, which drives the remodeling processes in Figure 1. As we will show, the concept of stress asymmetry is important for both planar bilayers and nanovesicles. Stress asymmetry in planar bilayers has also been studied by Deserno and coworkers [33,34], who used the term “differential stress” instead of “stress asymmetry”.

As far as the simulation methods are concerned, our first molecular dynamics simulations [35,36] were based on a united-atom approach. In order to study larger membrane segments for longer times, our more recent simulations were primarily based on dissipative particle dynamics (DPD) [37,38,39], a coarse-grained method that was first applied to lipid bilayers in [40]. For the nanovesicle simulations, we used the DPD code implemented in LAMMPS [41,42]. The simulations in [27] were performed via the GROMACS engine [43] with the coarse-grained Martini force field [44]. The simulation method of dissipative particle dynamics as applied to lipid bilayers and nanovesicles is briefly summarized in Appendix A.

Compared to all-atom molecular dynamics, the coarse-grained DPD method used here has several advantages. First, one can study the behavior of relatively large bilayer segments, which are not accessible to all-atom simulations. Second, the molecular models used in DPD are built up from a small number of different beads and, thus, involve only a small number of force parameters for the interaction forces between the beads, see Appendix A. As a consequence, one is able to explore large regions of the parameter space and to study the relative importance of these parameters in a systematic manner. Third, DPD simulations explicitly include water, the universal solvent for all biomolecules, and reproduce the correct hydrodynamics, because all forces used in DPD conserve momentum. The resulting hydrodynamic interactions between different membrane segments are important for the lipid-water systems considered here because the remodeling of membrane shape and topology involves transient hydrodynamic flows, which affect the time evolution of the systems. Fourth, many remodeling processes are slowed down by free energy barriers. The time to cross such a barrier is stochastic in nature. In such a situation, one should sample a sufficiently large number of remodeling events in order to obtain a reliable statistics [30,31,45,46], which would be prohibitively expensive for all-atom molecular dynamics simulations.

Coarse-grained molecular models bridge the gap between all-atom models and theories of membrane elasticity. The latter theories been very useful to understand and explain the behavior of giant vesicles [47]. If one wants to obtain a reliable model for a specific biomolecular system, one can start from an all-atom model with certain all-atom force fields and map these force fields onto optimized force parameters for DPD. An early example for such a mapping was described for polymersomes as assembled from PEO-based block copolymers in water [48]. More recent examples include antimicrobal peptides in lipid bilayers [49], folding of polypeptides [50], self-assembly of surfactants [51], brushes of zwitterionic polymers [52], and large proteins such as cytolysin A [53]. Here, we will not focus on a specific membrane system but rather on the generic behavior of such systems on supramolecular and nanoscopic scales. We show that our DPD results are consistent with the fluid-elastic theories on mesoscopic scales and that they lead to new relationships between the corresponding fluid-elastic parameters. Most importantly, we identify the leaflet tensions as important fluid-elastic parameters, which can be directly controlled by the assembly of lipids in silico and which provide new insight into the supramolecular and nanoscopic behavior of lipid bilayers and nanovesicles. Furthermore, the computational protocols developed here for DPD can be combined with the mapping of all-atom force fields to optimized DPD force parameters, in order to study the remodeling of specific membrane systems.

This review is organized as follows. The next Section 2 describes basic aspects of lipid bilayers and nanovesicles in an intuitive manner, leaving technical details to later sections. In Section 3 and Section 4, we review the fluid-elastic behavior of planar bilayers with one and several lipid components, respectively. We examine the two-dimensional parameter space defined by the leaflet tensions of planar bilayers, distinguishing equal leaflet tension from opposite leaflet tension states. Planar bilayers undergo stress-induced flip-flops and structural instabilities for sufficiently large stress asymmetries between the leaflets as discussed in Section 5.

The fluid-elastic behavior of nanovesicles is discussed in Section 6, Section 7 and Section 8. In Section 6, we review the fluid-elastic behavior of vesicle bilayers and their shape transformations in response to changes in vesicle volume. For negative stress asymmetry, the nanovesicle can form in-buds with closed membrane necks that undergo fission (Section 6.5). The two-dimensional leaflet tension space for nanovesicles is described in Section 6.6, their instabilities induced by large stress asymmetries in Section 7. The unusual shape transformations of nanovesicles exposed to small solutes that adsorb onto the vesicle bilayers is discussed in Section 8.

The three Section 9, Section 10 and Section 11 deal with the engulfment and endocytosis of condensate droplets and rigid nanoparticles. The main conclusion of these three sections is that the engulfment process may proceed in an axisymmetric or non-axisymmetric manner and that these two endocytic pathways are controlled by the stress asymmetry between the leaflet tensions. Section 12 provides strong evidence that the adhesion and fusion of two nanovesicles is also controlled by this stress asymmetry. At the end, we provide a summary and an outlook on related research topics for future studies.

## 2. Basic Aspects of Lipid Bilayers and Nanovesicles

Lipid bilayers involve many length scales. A single phospholipid molecule has a volume of about 1 nm${}^{3}$ [54]. If $10^{4}$ lipid molecules are assembled into a molecular bilayer, this bilayer forms a nanovesicle with a diameter of about 36 nm and a volume of about $2.5\times 10^{4}$ nm${}^{3}$. These rough estimates already demonstrate that even small nanovesicles, traditionally called “small unilamellar vesicles” (SUVs), provide a computational challenge, when we want to look at them with molecular resolution. Furthermore, on the scale of several nanometers, these vesicles have unusual fluid-elastic properties that lead to a large variety of nonspherical shapes, to dynamic transformations between these shapes, and to a remodeling of the vesicle topology.

### 2.1. Assembly and Molecular Dynamics of Lipids

#### 2.1.1. Properties of Individual Lipid Molecules

Phospholipids are major components of cellular membranes in all living organisms [55,56]. Synthetic phospholipids as produced by chemical companies can be used to prepare biomimetic lipid vesicles. Natural and synthetic phospholipids have the same molecular architecture, which is captured by the coarse-grained lipid model in Figure 2. Each such molecule has a hydrophilic head, which is water-soluble, and two hydrocarbon chains, which are hydrophobic and water-insoluble. This molecular architecture is encoded in the coarse-grained lipid molecule displayed in Figure 2, which has been used in most of the simulation studies reviewed here [24,25,26,30,32].

#### 2.1.2. Assembly and Molecular Dynamics of Lipids

In water, lipids assemble into molecular bilayers as shown in Figure 3. The headgroups of the bilayers form two interfaces with the surrounding aqueous solutions, thereby shielding the hydrophobic chains from these solutions. In Figure 3a, which represents a simulation snapshot of a planar bilayer, the bilayer has edges which arise from the periodic boundary conditions imposed in the simulations. If we considered such a finite bilayer patch surrounded by an aqueous solution, the hydrophobic core of the bilayer would come into contact with this solution and would make a large positive contribution to the system’s free energy. To avoid these bilayer edges, the planar bilayer patch will close up to form a nanovesicle as in Figure 3b.

To set up the simulation studies, planar bilayers are prepared by the assembly of $N_{ll}$ lipids in their lower leaflet and $N_{ul}$ lipids in their upper leaflet. These bilayers are *symmetric* when both leaflets contain the same number of lipids, i.e., $N_{ul}=N_{ll}$, whereas *asymmetric* bilayers are obtained for $N_{ul}\neq N_{ll}$. The snapshot in Figure 3a displays an example for a symmetric bilayer with the additional property that both leaflets of this bilayer are tensionless. The latter property defines the reference state of the bilayer. Nanovesicles are assembled from $N_{il}$ lipids in their inner leaflet and $N_{ol}$ lipids in their outer leaflet. The snapshot in Figure 3b displays a nanovesicle with tensionless leaflets. The latter reference state is not symmetric with respect to the lipid numbers assembled in the two leaflets because it is always characterized by $N_{ol}>N_{il}$.

In addition to synthetic liposomes, cellular nanovesicles such as exosomes and extracellular vesicles are released from almost every type of living cell [57] and are crucial for the chemical communication between the cells. Exosomes have been proposed as possible biomarkers for diseases [58,59] and as potential drug delivery systems [60,61].

#### 2.1.3. Lateral Diffusion and Interleaflet Flip-Flops of Lipids

Cellular membranes are maintained in a fluid state which is characterized by fast lateral diffusion of the lipid molecules within one of the bilayer leaflets. The lateral diffusion was first probed by spin-labeled lipids [62,63] and steroids [64,65] which led to lateral diffusion constants of the order of 1 μm${}^{2}$ per second. Nowadays, the lateral diffusion of membrane molecules can be observed directly by fluorescence recovery after photobleaching (FRAP) [66] and by single particle tracking [67,68,69,70], two methods that have been applied to a large variety of biomimetic and biological membranes. These studies confirm that the lateral diffusion constants of membrane molecules are indeed of the order of 1 μm${}^{2}$ per second. Thus, if we labeled a single lipid and monitored its position within the bilayer leaflet, we would observe this molecule to perform a random walk, covering a few nanometers within a few nanoseconds. On short time scales, this lateral motion arises from local swaps of the labeled lipid with one of its neighboring lipids.

Each lipid molecule stays within one of the bilayer leaflets until it undergoes a transition or flip-flop to the other leaflet. During such an interleaflet flip-flop, the hydrophilic headgroup of the lipid has to be moved across the hydrophobic core of the bilayer which represents a significant energy barrier. For phospholipids, these barriers are quite large and lead to a wide range of flip-flop rates, with the typical time scale for one flip-flop varying from minutes to days [62,71,72,73,74,75,76,77,78,79]. For cholesterol and related sterols. which have a smaller headgroup and encounter a reduced energy barrier, the published values for the inverse flip-flop rates vary from seconds [80,81] to milliseconds [82].

#### 2.1.4. Fluid-Elastic Behavior of Molecular Bilayers and Nanovesicles

In their fluid state, lipid bilayers and biomembranes have unusual elastic properties. When we view the bilayers as thin sheets, these sheets can sustain two types of elastic deformations, in-plane stretching and compression as well as out-of-plane bending. Elastic stretching and compression is intimately related to membrane tension whereas bending deformations generate membrane curvature. Next, we will discuss the different notions of membrane tension and curvature without discussing the underlying computational methods, which will be addressed later.

### 2.2. Different Notions of Membrane Tension

The simplest notion of membrane tension is *bilayer tension* which acts to stretch or compress both leaflets of the bilayer simultaneously. When we stretch or compress a molecular bilayer, we will increase or decrease its area. A widely used experimental method to measure the bilayer tension is by micropipette aspiration of giant unilamellar vesicles (GUVs). Using this method, one can obtain the limiting bilayer tension, for which the vesicle membrane ruptures. Typical values for this tension of rupture are a few mN/m [83,84]. Because the area compressibility modulus is two orders of magnitude larger than the tension of rupture, the area of a lipid bilayer can only be stretched by a few percent before it ruptures. The smallest tensions that can be resolved using micropipette aspiration correspond to the smallest suction pressures that can be applied to the membranes in a reliable manner. These smallest suction pressures are about 1 Pa [85].

The lipid bilayer consists of two leaflets which are separated by a leaflet-leaflet interface. This interface is highlighted in both panels of Figure 3 by using different colors for the lipids in the different leaflets. It is then natural to assume that we can decompose the bilayer tension into two leaflet tensions. In molecular dynamics simulations, this decomposition can indeed be performed in a quantitative manner. When the lipid bilayer is in mechanical equilibrium, each leaflet must experience a laterally uniform leaflet tension because each leaflet represents a two-dimensional fluid. Furthermore, in the absence of lipid flip-flops between the leaflets, each leaflet contains a constant number of lipids which determines the leaflet tension in combination with geometrical constraints. For a planar bilayer as in Figure 3a, these constraints arise from the lateral dimensions of the simulation box while they arise from the vesicle volume for a vesicle bilayer as in Figure 3b. In the presence of lipid flip-flops, the constant lipid numbers must be replaced by the average lipid numbers in each leaflet.

#### 2.2.1. Bilayer Tension and Leaflet Tensions

The decomposition of the bilayer tension, Σ, into its two leaflet tensions, $\Sigma_{1}$ and $\Sigma_{2}$, is described by the simple relation $\Sigma =\Sigma_{1}+\Sigma_{2}$ as will be shown in Section 3.3 below. Each leaflet tension can be positive or negative depending on the number of lipids that are assembled in the leaflet. A positive leaflet tension implies that the leaflet is stretched whereas a negative leaflet tension describes a compressed leaflet. In order to avoid membrane rupture, the osmotic conditions in the aqueous solutions surrounding the bilayer membrane must be adjusted in such a way that the bilayer is subject to a relatively low bilayer tension. However, even for a tensionless bilayer with vanishing bilayer tension $\Sigma =\Sigma_{1}+\Sigma_{2}=0$, the two leaflets of the bilayer may still experience significant leaflet tensions. Indeed, a tensionless bilayer with Σ=0 only implies that $\Sigma_{2}=-\Sigma_{1}$. In general, the latter relation can be fulfilled whenever one leaflet is stretched and the other leaflet is compressed by the opposite leaflet tension.

On the other hand, if both leaflet tensions, $\Sigma_{1}$ and $\Sigma_{2}$, vanish, the bilayer tension $\Sigma =\Sigma_{1}+\Sigma_{2}$ must vanish as well. The special bilayer state with vanishing leaflet tensions, $\Sigma_{1}=\Sigma_{2}=0$ defines a unique reference state for all bilayers assembled with the same total number $N=N_{1}+N_{2}$ of lipids in the two leaflets 1 and 2. In this reference state, the lipids are packed in an optimal manner and attain an optimal area as well as an optimal volume per lipid.

In the following sections, we will consider the stress asymmetry $\Delta \Sigma \equiv \Sigma_{2}-\Sigma_{1}$ between the two leaflets. The stress asymmetry $\Delta \Sigma =2\Sigma_{2}=-2\Sigma_{1}$ for a tensionless bilayer with $\Sigma_{1}+\Sigma_{2}=0$. A nonzero stress asymmetry implies that the bilayer has a certain preferred curvature. If leaflet 1 is stretched with $\Sigma_{1}>0$ and leaflet 2 is compressed with $\Sigma_{2}<0$, the bilayer prefers to bulge towards leaflet 2 in order to reduce both the compression of leaflet 2 and the stretching of leaflet 1. On the other hand, the bilayer prefers to bulge towards leaflet 1, if leaflet 2 is stretched and leaflet 1 is compressed. Thus, a nonzero stress asymmetry between the two leaflets leads to a preferred curvature of the bilayer.

#### 2.2.2. Average Position of Leaflet-Leaflet Interface

To actually compute the leaflet tensions, we need to identify the spatial regions that are, on average, occupied by the two leaflets and to decompose the overall bilayer tension into two separate contributions from the two leaflets, see Figure 3. The average position of the leaflet-leaflet interface defines the midplane for planar bilayers and the midsurface for spherical nanovesicles. The position of the midsurface is easy to find for symmetric planar bilayers but requires some computational effort in all other cases, both for the midplane of asymmetric planar bilayers and for the midsurface of the spherical vesicle bilayers.

During the last eight years, two computational methods were found to be particularly useful for the computation of the midplanes and midsurfaces: the CHAIN protocol based on the density profile of the chain beads [24,25,28,30] and the VORON protocol based on the computation of molecular volumes via Voronoi tesselation [32]. The CHAIN protocol will be discussed in Section 3.4.2 for planar bilayers and in Section 6.3.1 for nanovesicles. The VORON protocol will be described in Section 3.4.3 for planar bilayers and in Section 6.3.2 for nanovesicles.

#### 2.2.3. Lipid Numbers and Membrane Tensions

During the assembly of lipids into planar bilayers and nanovesicles, the lipid numbers can be directly controlled as part of the simulation setup. In contrast, the bilayer and leaflet tensions are properties of the lipid assemblies that have to be determined from the analysis of the simulation data and require some computational effort. Thus, compared to the membrane tensions, lipid numbers seem to provide the more obvious control parameters. On the other hand, lipid numbers and membrane tensions are two physical quantities that are fundamentally different as can be understood from the following thought experiments.

Let us consider two vesicles consisting of two lipid bilayers, *a* and *b*, that are assembled from $N^{a}$ and $N^{b}$ lipids. When we imagine to fuse these two bilayers, we will end up with a larger bilayer, a∪b, that contains $N^{a}+N^{b}$ lipids. In contrast, when the bilayer tensions of the two vesicle bilayers are $\Sigma^{a}$ and $\Sigma^{b}$ before fusion, the bilayer tension of the fused bilayer a∪b will be different from $\Sigma^{a}+\Sigma^{b}$. Indeed, if the two bilayer tensions $\Sigma^{a}$ and $\Sigma^{b}$ were equal before fusion, the bilayer tension of the fused bilayer will be equal to $\Sigma^{a}=\Sigma^{b}$ as well. In general, the bilayer tension of the fused bilayer will attain an intermediate value between $\Sigma^{a}$ and $\Sigma^{b}$ after the fused bilayer has been equilibrated.

Using the terminology of thermodynamics [86], the lipid numbers and bilayer tensions are *extensive* and *intensive* variables, respectively. When we combine two systems, extensive variables are added whereas intensive variables are equilibrated. This distinction can be extended to the individual leaflets and to the associated leaflet tensions when we consider the hemifusion of two vesicle bilayers. As a result, we conclude that the lipid numbers of the leaflets represent extensive variables whereas the leaflet tensions represent intensive variables.

### 2.3. Emergence of Membrane Curvature at the Nanoscale

When we intend to determine the curvature-elastic properties of a planar bilayer, this bilayer should be prepared in a state of sufficiently low bilayer tension as illustrated in Figure 3 [36]. In addition, one should also note that the description of the bilayer shape in terms of curvature requires a certain amount of coarse graining. Indeed, because membranes are immersed in liquid water, each membrane molecule undergoes thermal motion with displacements both parallel and perpendicular to the membrane. The perpendicular displacements typically represent molecular protrusions that are induced by thermal noise and roughen the interface between this leaflet and the surrounding aqueous solution, as shown in Figure 4.

Therefore, in order to characterize a lipid bilayer by its curvature, one has to consider small membrane patches and to average over the molecular conformations within these patches. The minimal lateral size of these patches can be determined from the spectrum of the bilayer’s shape fluctuations and was found to be about 1.5 times the membrane thickness, see Figure 4, for both one-component [24,36] and two-component [25] lipid bilayers, see Section 4.1.4 and Figure 11 below. For a membrane with a thickness of 4 nm, this minimal size is about 6 nm. Because such a membrane patch contains 80–100 lipid molecules, membrane curvature should be regarded as an emergent property arising from the cooperative behavior of many lipid molecules on supramolecular scales.

### 2.4. Alternative Notions of Membrane Curvature

In the literature on lipid bilayers, alternative notions of membrane curvature have been proposed. In order to compare these notions with the supramolecular view emphasized here, it is instructive to start from the simple (and popular) view that individual lipid molecules have a certain shape [87] which leads to the notion of intrinsic lipid curvature and to the idea that the lipid molecules may become ‘frustrated’ when they are packed into a bilayer [88]. However, the view that a phospholipid has a certain fixed shape is problematic, in particular because this shape should depend on the local environment of the lipid and because this environment is variable for a fluid bilayer. Thus, we promote the view that membrane curvature emerges as a supramolecular property for small membrane patches that contain 80–100 lipids. For such patches, the crucial curvature parameter is provided by the spontaneous curvature, which describes the transbilayer asymmetry between the two leaflets of the bilayer membrane.

#### 2.4.1. Intrinsic Lipid Curvature from Inverted Hexagonal Phases

It is tempting to assume that a phospholipid as in Figure 2 has a certain, well-defined shape. Such a shape concept for individual lipid molecules has been used to explain the complex phase behavior of aqueous lipid dispersions, which involve a variety of non-bilayer phases. In particular, many phospholipids form an inverted hexagonal phase, in which the lipids are assembled around cylindrical water channels. These channels form a 2-dimensional hexagonal lattice which can be studied by X-ray and neutron scattering. In this way, one can measure the radius $R_{0}$ of the cylindrical interface between a single water channel and the glycerol backbone of the phospholipids [89]. This interface radius has been used to define the magnitude of the intrinsic lipid curvature $C_{0}$ via $|C_{0}|=1/R_{0}$. For those lipids that actually form an inverted hexagonal phase, the values of $R_{0}$ vary between 2.5 nm and 3 nm, which is smaller than the typical thickness of a lipid bilayer.

#### 2.4.2. Limitations of the Intrinsic Curvature Concept

The assumption that each lipid molecule has a certain, fixed shape is problematic for several reasons. First, most lipids that form an inverted hexagonal phase form other non-bilayer and bilayer phases as well [90]. In these different phases, the lipids are packed locally in a different manner. Therefore, the observed polymorphism of aqueous lipid dispersions does not support the idea that a certain type of lipid has always the same shape. Indeed, the shape of an individual lipid molecule must depend on the interactions of this molecule with the other molecules in its neighborhood, and this molecular neighborhood is clearly different when the lipid resides in different lipid-water phases.

Second, thermal noise will affect both the lipid molecule under consideration and its molecular neighborhood. One direct consequence of thermal noise is the lateral diffusion of an individual lipid within the lipid assembly. As a result, the diffusing lipid will encounter different local neighborhoods, in particular when this assembly contains several lipid components. Third, when we consider rigid lipid molecules with a non-cylindrical shape and try to pack them into the two leaflets of a bilayer, the lipids would become “frustated”. It was originally proposed [88] that such a frustration leads to packing defects or voids. However, this proposal ignores the flexibility of the lipids and the fluidity of the bilayers. Indeed, if the lipids form a fluid bilayer, packing defects or voids will be mobile and should be averaged out by local density fluctuations and lateral lipid diffusion.

#### 2.4.3. Spontaneous Curvature of Bilayer Membranes

For a symmetric bilayer of lipids, the intrinsic curvatures of the lipids in the two leaflets must, on average, cancel out, irrespective of the assumed shape of the lipids. However, bilayer membranes are typically not symmetric because the two leaflets can differ in their molecular composition and can be exposed to different aqueous solutions. This local asymmetry between the two leaflets defines the spontaneous curvature of the bilayers as originally introduced by Helfrich [91], in close analogy to the curvature elasticity of liquid crystals [92]. The corresponding elastic energy of biomembranes defines the spontaneous curvature model [93,94]. Some recent examples for such asymmetric bilayers are provided by bilayers exposed to asymmetric solutions of simple sugars [95] and to exterior solutions of His-tagged proteins that bind to anchor lipids in the outer bilayer leaflet [96]. In these latter experiments, the lipid membranes contained cholesterol which undergoes frequent flip-flops and implies that area-difference elasticity [97,98,99] plays no role, which is useful because the latter type of elasticity would otherwise involve additional parameters.

#### 2.4.4. Spontaneous Curvature of Reference States

For planar bilayers with a single lipid component, the reference state with vanishing leaflet tensions represents a symmetric bilayer, for which the preferred or spontaneous curvature must vanish. In contrast, the reference state of bilayers with several lipid components can exhibit a compositional asymmetry even if both leaflet tensions vanish. Therefore, the reference state of a multi-component bilayer can have a significant spontaneous curvature as demonstrated for planar bilayers with two phospholipid components [27], see Section 4.2.2 below.

Nanovesicles with a single lipid component also lead to an asymmetric reference state that contains a different number of lipids, $N_{il}$ and $N_{ol}>N_{il}$, in the inner and outer leaflet. The simulation data of nanovesicles with $N_{il}+N_{ol}=10,100$ lipids are consistent with a reference state that has vanishing leaflet tensions but a small, nonzero spontaneous curvature [28].

### 2.5. Dependence of Leaflet Tensions on Bilayer Geometry

In addition to the number of lipids assembled in the two leaflets, the leaflet tensions are also determined by the overall geometry imposed on the lipid bilayers.

#### 2.5.1. Geometric Control Parameters for Planar Bilayers

For planar bilayers, the leaflet tensions depend on the lipid numbers $N_{ll}$ and $N_{ul}$ assembled in the lower and upper leaflet of the bilayer as well as on the base area of the simulation box, see Figure 5. This figure displays three symmetric bilayers, which contain the same number of lipids, $N_{ll}=N_{ol}=841$, in each leaflet but differ in the base area of the simulation box. The symmetric bilayer in Figure 5b corresponds to the reference state of the bilayer with tensionless leaflets and leaflet tensions $\Sigma_{ll}=\Sigma_{ul}=0$. For this reference state, the area per lipid, *a*, has the optimal value $a^{0}=1.22\;d^{2}$ and the volume per lipid, *v*, has the optimal value $v^{0}=3.57\;d^{3}$ [32]. The symmetric bilayer in Figure 5a has a reduced base area compared to the reference state in Figure 5b whereas the symmetric bilayer in Figure 5c has an increased base area.

Because the two leaflets of the three planar bilayers in Figure 5 contain the same lipid numbers $N_{ll}$ and $N_{ul}=N_{ll}$, both leaflets experience the same leaflet tension, $\Sigma_{ll}=\Sigma_{ul}$, for all three cases. However, for the bilayer spanning the reduced base area in Figure 5a, both leaflets are compressed by negative leaflet tensions, $\Sigma_{ll}=\Sigma_{ul}<0$, while they are stretched by positive leaflet tensions, $\Sigma_{ll}=\Sigma_{ul}>0$, in Figure 5c. The molecular area per lipid is equal to $a=1.18\;d^{2}$ for the compressed bilayer in Figure 5a. and to $a=1.29\;d^{2}$ for the stretched bilayer in Figure 5c. The molecular volume per lipid, on the other hand, is hardly affected by changes in the base area and has the values $v=3.56\;d^{3}$ for the compressed bilayer in Figure 5a and $v=3.58\;d^{3}$ for the stretched bilayer in Figure 5c.

#### 2.5.2. Geometric Control Parameters for Vesicle Bilayers

For nanovesicles, the leaflet tensions depend on the lipid numbers $N_{il}$ and $N_{ol}$ assembled in the inner and outer leaflet of the vesicle bilayer as well as on (i) the vesicle volume and (ii) the curvature of the vesicle membrane. When we reduce the volume of a spherical vesicle, the bilayer membrane gains some excess area which allows the membrane to bend in such a way that both leaflets become more relaxed. As a consequence, a spherical vesicle can transform into rather different shapes as illustrated in Figure 6.

This figure displays two spherical nanovesicles, one of which transforms into a stomatocyte with an in-bud, the other into a dumbbell with an out-bud. The spherical nanovesicle in Figure 6a is bounded by an inner leaflet that contains $N_{il}=4400$ lipids and by an outer leaflet with $N_{ol}=5700$ lipids. A slight reduction of the vesicle volume, which mimicks the experimental procedure of osmotic deflation, leads to a tensionless bilayer, for which the inner leaflet is compressed whereas the outer leaflet is stretched. The spherical nanovesicle in Figure 6c is bounded by an inner leaflet that contains $N_{il}=3800$ lipids and by an outer leaflet with $N_{ol}=6300$ lipids. A slight reduction of the vesicle volume now leads to a tensionless bilayer, for which the inner leaflet is stretched whereas the outer leaflet is compressed [28].

The two spherical vesicles in Figure 6a,c have the same size and, thus, the same curvature of the bilayer membrane. Comparing the behavior of nanovesicles with different sizes shows that the fluid-elastic behavior of the nanovesicles depends on this curvature as well [30,32].

### 2.6. Topological Transformations of Closed Membrane Compartments

The shape changes of vesicles as displayed in Figure 6 are intimately related to two curvature-elastic moduli, the spontaneous curvature and the bending rigidity, κ, of the vesicle membrane. The bending rigidity describes the resistance of the membrane against local curvature changes. For phospholipid bilayers, the bending rigidity κ has a typical value of about $20\;k_{B}T≃10^{-19}$ J. In addition to shape changes, bilayer membranes can also undergo topological transformations by fission and fusion processes.

During fission, one membrane compartment is divided up into two daughter compartments. Three examples for a fission process are shown in Figures 22, 31 and 39 below. During fusion, two membrane compartments are combined into a single compartment as illustrated in Figures 33 and 44 further below. In the context of curvature elasticity, the energy change during such a topological transformation is proportional to the Gaussian curvature modulus $\kappa_{G}$ and to the integral over the Gaussian curvature of the membrane surface [91]. The modulus $\kappa_{G}$ is negative with a typical value $\kappa_{G}≃-\kappa ≃-20\;k_{B}T≃10^{-19}$ J [47]. For the closed surface of a vesicle, the integral over the Gaussian curvature is a topological invariant and equal to 2πχ with the Euler characteristic χ as follows from the Gauss–Bonnet theorem [100]. Thus, during a topological transformation from the initial vesicle state 1 to the final vesicle state 2, the change of the Gaussian curvature energy is given by
(1)$$\Delta E_{G}=2\pi \left(\chi_{2}-\chi_{1}\right)\kappa_{G}$$
which is negative and positive for fission and fusion processes, respectively, [47]. For the shape transformations in Figure 6, both the initial and the final vesicle state have the same topology and, thus, the same Euler characteristic, $\chi_{2}=\chi_{1}$. As a consequence, the change $\Delta E_{G}$ of the Gaussian curvature energy vanishes and plays no role for the shape transformations in Figure 6. The latter feature applies to all shape transformations, which proceed via smoothly curved membranes and leave the local bilayer structure intact.

## 3. Planar Bilayers with One Lipid Component

We now discuss the behavior of planar bilayers in more detail. In this section, we focus on bilayers with a single lipid component. In the next section, we will consider the behavior of bilayers with two and three lipid components.

### 3.1. Assembly of Lipids into Planar Bilayers

To set up the simulation, a planar bilayer is preassembled by placing lipid molecules onto two adjacent planes, thereby forming the initial states of the lower and upper leaflet of the bilayer. Once such an initial bilayer has been assembled with $N_{ll}$ lipids in the lower and $N_{ul}$ lipids in the upper leaflet, it spans the simulation box as in Figure 5. For the one-component bilayers discussed in this section, the lipids do not undergo flip-flops between the two leaflets which implies that the lipid numbers $N_{ll}$ and $N_{ul}$ do not change during the simulations.

The simulation box has the shape of a cuboid, see Figure 5. Periodic boundary conditions are imposed for all three directions parallel to the sides of the cuboid. The Cartesian coordinate perpendicular to the bilayer is denoted by *z*, the two lateral coordinates parallel to it by *x* and *y*.

### 3.2. Density and Stress Profiles

After a planar bilayer has been assembled, it can be characterized by its density and stress profiles, as illustrated in Figure 7. Because of the periodic boundary conditions, all profiles depend on the perpendicular coordinate *z* but are independent of the lateral coordinates *x* and *y*. Examples for the density profiles of water (W), headgroup (H), and lipid chain (C) beads are displayed in Figure 7a–c, the corresponding stress profiles are shown in Figure 7d–f. The computation of the stress profiles is described in Appendix B.

The profiles of the symmetric reference state with tensionless leaflets are depicted in panels a and d of Figure 7 for $N_{ll}=N_{ul}=841$ lipids in each leaflet. All profiles in these two panels are mirror symmetric with respect to the midplane at $z=z_{\mathrm{mid}}$, which we take to be the origin of the *z*-coordinate, which is shifted in such a way that $z_{\mathrm{mid}}\equiv 0$. The remaining panels of Figure 7 display two examples for asymmetric bilayers, which contain different numbers of lipids in the two leaflets. The asymmetric bilayer in panels b and e of Figure 7 contains $N_{ll}=815$ lipids in its lower leaflet and $N_{ul}=862$ lipids in its upper leaflet. The asymmetric bilayer in panels c and f of Figure 7 consists of a lower leaflet with $N_{ll}=801$ lipids and an upper leaflet with $N_{ul}=875$ lipids.

For all bilayers in Figure 7, the water (W) density has the constant value $\rho_{W}=3/d^{3}$ away from the bilayer, which ensures that the water model reproduces the correct water compressibility at room temperature $T=298{\;}^{\circ }$ K [39]. In the spatial region that is occupied by the bilayer, the density profile $\rho_{C}=\rho_{C}\left(z\right)$ of the hydrophobic chain groups has a pronounced maximum, which is taken to define the average position of the leaflet-leaflet interface in the CHAIN protocol, see Section 3.4.2 below. The density profiles $\rho_{H}$ of the hydrophilic headgroups exhibits two peaks which are located at the two bilayer-water interfaces. Inspection of Figure 7 reveals that the density profiles of the asymmetric bilayers in panels b and c are almost identical with those of the symmetric reference state in panel a. In contrast, the stress profiles of the asymmetric bilayers in panels e and f of Figure 7 are strongly asymmetric compared to the symmetric stress profile of the reference state in panel d.

### 3.3. Bilayer Tension and Leaflet Tensions

The bilayer tension Σ is obtained by integrating the stress profile s=s(z) over the *z*-coordinate. We denote the linear dimension of the simulation box in the *z*-direction by $L_{z}$ and consider the interval $-\frac{1}{2}\;L_{z}\leq z<+\frac{1}{2}\;L_{z}$ for the *z*-coordinate. The bilayer tension Σ is then given by [35]
(2)$$\Sigma =\int_{-\frac{1}{2}\;L_{z}}^{+\frac{1}{2}\;L_{z}}\;dz\;s\left(z\right)\;(\mathrm{planar}\;\mathrm{bilayer})$$

In practice, one can reduce the integration over *z* to a smaller interval because the stress profile decays rapidly to zero as we move away from the bilayer, see the examples in Figure 7d–f.

To determine the leaflet tensions $\Sigma_{ll}$ and $\Sigma_{ul}$, we need to decompose the bilayer into its two leaflets. This decomposition is based on the bilayer’s midsurface which corresponds to the average position of the molecular interface between the two leaflets. In the snapshots of Figure 3 and Figure 5, this interface is provided by the boundary surface between the blue and yellow hydrocarbon chains in the two leaflets.

As depicted in Figure 3 and Figure 5, the leaflet-leaflet interface undergoes shape fluctuations. Therefore, we need to consider the average position of this interface, obtained by sampling a large number of statistically independent conformations. Because of the periodic boundary conditions used in the planar bilayer simulations, the average position of the leaflet-leaflet interface is then fully characterized by its average *z*-value, $\langle z\rangle =z_{\mathrm{mid}}$, which defines the midplane of the bilayer membrane. The two leaflet tensions $\Sigma_{ll}$ and $\Sigma_{ul}$ of the lower and upper leaflet are then obtained from [24,25,27]
(3)$$\Sigma_{ll}\equiv \int_{-\frac{1}{2}\;L_{z}}^{z_{\mathrm{mid}}}\;dz\;s\left(z\right)\;\mathrm{and}\;\Sigma_{ul}\equiv \int_{z_{\mathrm{mid}}}^{+\frac{1}{2}\;L_{z}}\;dz\;s\left(z\right)\;.$$

To compute these two integrals, we need to determine the midplane position $z=z_{\mathrm{mid}}$, which represents the average position of the leaflet-leaflet interface.

### 3.4. Midplane of Leaflet-Leaflet Interface for Planar Bilayers

For a symmetric planar bilayer, the midplane is easy to compute because of the up-down symmetry of the bilayer or, equivalently, of the mirror-symmetric density profiles. For asymmetric planar bilayers, the identification of the midplane requires a certain computational effort.

#### 3.4.1. Midplane of Symmetric Planar Bilayers

For symmetric planar bilayers, the lower and upper leaflets contain the same number of lipids, i.e., $N_{ll}=N_{ul}$. In this case, the midplane coincides with the plane of symmetry. Thus, for a symmetric bilayer, any density profile, ρ=ρ(z), across the bilayer is mirror-symmetric with respect to $z=z_{\mathrm{mid}}$. This symmetry applies both to the reference state with tensionless leaflets and to all other bilayer states with $N_{ll}=N_{ul}$ as illustrated in Figure 5. In addition, the two leaflet tensions $\Sigma_{ul}$ and $\Sigma_{ll}$ are necessarily equal for a symmetric bilayer. It then follows that vanishing bilayer tension $\Sigma =\Sigma_{ul}+\Sigma_{ll}=2\Sigma_{ul}=2\Sigma_{ll}$ implies vanishing leaflet tensions $\Sigma_{ul}=\Sigma_{ll}=0$. Therefore, in order to obtain the reference state with tensionless leaflets for planar bilayers with a fixed total lipid number $N_{ll}+N_{ul}$, it is sufficient to examine the symmetric bilayer states with $N_{ll}=N_{ul}$, to measure the bilayer tension Σ for different base areas of the simulation box, and to interpolate the simulation data to determine the unique symmetric bilayer with vanishing bilayer tension Σ=0.

#### 3.4.2. Midplane from Density Profile of Hydrophobic Chains

On the other hand, for asymmetric bilayers, which are assembled from a different number of lipids in the two leaflets, the position of the midplane is not obvious. To compute the value of $z_{\mathrm{mid}}$ in the latter case, several computational protocols have been developed [24,25,27,32]. We focus here on two of these protocols, denoted by CHAIN and VORON.

In the CHAIN protocol, which is the simplest protocol, the position $z_{\mathrm{mid}}$ of the midplane is identified with the *z*-value, at which the density of the hydrocarbon chains has an extremum [24,25]. For the coarse-grained simulations used here, this extremum is a maximum, see Figure 7a–c.

The second protocol, called VORON, is based on Voronoi tessellation of the molecular groups [32] which allows us to calculate the volume per lipid and the volumes of the different subcompartments of the bilayer system.

#### 3.4.3. Midplane from Volume per Lipid via Voronoi Tessellation

Voronoi tessellation assigns a polyhedral cell to each bead. This cell is defined by the requirement that all points in the cell are closer to the center of the chosen bead than to the center of any other bead. Panels a and c of Figure 8 display an example for the tessellation of a planar bilayer conformation. Panels b and d of Figure 8 provide an example for the tessellation of an individual lipid molecule.

Summing up the volumes of those beads that belong to a particular subcompartment of the system, we obtain the volume $V_{lW}$ of the water subcompartment below the bilayer, the volume $V_{ll}$ of the lower leaflet, the volume $V_{ul}$ of the upper leaflet, and the volume $V_{uW}$ of the water subcompartment above the bilayer. The VORON protocol to compute the midplane position $z_{\mathrm{mid}}$ is based on these subcompartment volumes [32].

Let $A_{\|}$ be the base area of the simulation box parallel to the planar bilayer and let the *z*-coordinate perpendicular to this bilayer vary within the range $-\frac{1}{2}\;L_{z}\leq z\leq +\frac{1}{2}\;L_{z}$ and the base area is of the simulation box is denoted by $A_{\|}$. The cuboid geometry of the simulation box then implies that the midplane position $z_{\mathrm{mid}}$ satisfies the relationship
(4)$$\left(\frac{1}{2}\;L_{z}+z_{\mathrm{mid}}\right)\;A_{\|}=V_{ll}+V_{lW}$$
where $V_{ll}$ and $V_{lW}$ are the volumes of the lower leaflet of the planar bilayer and of the water subcompartment below this leaflet. Likewise, the midplane position $z_{\mathrm{mid}}$ can be obtained via the relation
(5)$$\left(\frac{1}{2}\;L_{z}-z_{\mathrm{mid}}\right)\;A_{\|}=V_{ul}+V_{uW}$$
in terms of the volumes $V_{ul}$ and $V_{uW}$ corresponding to the upper leaflet and to the upper water subcompartment. The two relations in Equations (Equation 4) and (Equation 5) both determine the midplane coordinate $z=z_{\mathrm{mid}}$ in terms of known geometric parameters. As a result, one obtains two $z_{\mathrm{mid}}$-values which are identical within the numerical accuracy.

### 3.5. Two-Dimensional Leaflet Tension Space for Planar Bilayers

The two leaflet tensions $\Sigma_{ll}$ and $\Sigma_{ul}$ define a two-dimensional parameter space for planar bilayers, as depicted in Figure 9 for bilayers with a total number of $N_{ll}+N_{ul}=1682$ lipids. The origin $\left(\Sigma_{ll},\Sigma_{ul}\right)=\left(0,0\right)$ of this leaflet tension space defines the relaxed reference state with tensionless leaflets, see the snapshot in Figure 5b. As explained in Section 3.4.1 above, the reference state of the planar bilayers with $N_{ll}+N_{ul}=1682$ can be obtained by focusing on the symmetric bilayers with $N_{ll}=N_{ul}=841$ and determining the unique bilayer with vanishing bilayer tension Σ=0. To characterize the elastic response of the reference state, it is useful to distinguish two types of elastic deformations, corresponding to the red and black data in Figure 9.

The red data represent elastic deformations of the reference state with
(6)$$\Sigma_{ll}=\Sigma_{ul}\;(\mathrm{equal}\;\mathrm{leaflet}\;\mathrm{tensions}),$$
for which we use the abbreviation ELT. Examples for ELT states are displayed in panels a and c of Figure 5. The ELT states are located in the main diagonal of the leaflet tension space in Figure 9. The black data in Figure 9 correspond to elastic deformations of the reference state with
(7)$$\Sigma_{ll}=-\Sigma_{ul}\;(\mathrm{opposite}\;\mathrm{leaflet}\;\mathrm{tensions}),$$
for which we use the abbreviation OLT. The OLT states are located on the diagonal which is orthogonal to the main diagonal in Figure 9. In what follows, this diagonal is called the ‘perpendicular diagonal’. All OLT states can be obtained from the reference state by reshuffling lipids from one leaflet to the other, keeping the total lipid number $N_{ll}+N_{ul}$ constant and imposing the constraint of vanishing bilayer tension $\Sigma =\Sigma_{ll}+\Sigma_{ul}=0$ for the OLT states. As a consequence, one leaflet becomes compressed by a negative leaflet tension whereas the other leaflet becomes stretched by a positive leaflet tension.

### 3.6. Area Compressibility Modulus of Symmetric Planar Bilayers

The elastic ELT deformations of the symmetric reference state with optimal area per lipid, $a_{ll}^{0}=a_{ul}^{0}\equiv a_{\mathrm{le}}^{0}$, lead to other symmetric bilayer states with a reduced or increased area per lipid, $a_{ll}=a_{ul}\equiv a_{\mathrm{le}}$, corresponding to the compression or stretching of the reference state. The associated area dilation is defined by
(8)$$\Delta a\equiv \frac{a_{\mathrm{le}}-a_{\mathrm{le}}^{0}}{a_{\mathrm{le}}^{0}}\;.$$

The ELT deformations can be described by the asymptotic equality
(9)$$\Sigma \approx K_{A}\Delta a\;\mathrm{for}\;\mathrm{small}\;\Delta a$$
between the bilayer tension Σ and the area dilation Δa, which defines the area compressibility modulus $K_{A}$. Analysing the simulation data of symmetric bilayers with $N_{ll}=N_{ul}=841$ as reviewed here leads to the numerical value $a^{0}=1.22\;d^{2}$ for the optimal area per lipid and to the value $K_{A}=27\pm 1\;k_{B}T/d^{2}$ for the area compressibility modulus [24,25,32].

### 3.7. Stress Asymmetry between Leaflets of Planar Bilayers

The stress asymmetry ΔΣ between the two leaflet tensions of a planar bilayer is defined by
(10)$$\Delta \Sigma \equiv \Sigma_{ul}-\Sigma_{ll}=\int_{z_{\mathrm{mid}}}^{+\frac{1}{2}\;L_{z}}\;dz\;s\left(z\right)-\int_{-\frac{1}{2}\;L_{z}}^{z_{\mathrm{mid}}}\;dz\;s\left(z\right)$$
where the second equality follows from the integral expressions for the leaflet tensions in Equation (Equation 3). The stress asymmetry ΔΣ vanishes for all ELT states of the planar bilayer, for which both leaflets contain the same number of lipids, i.e., for which $N_{ul}=N_{ll}$. In contrast, all OLT states of the planar bilayer exhibit a nonzero stress asymmetry because the OLT states are characterized by one stretched and one compressed leaflet. In fact, apart from the ELT states along the main diagonal of the two-dimensional leaflet tension space in Figure 9, all points $\left(\Sigma_{ll},\Sigma_{ul}\right)$ in this space have a nonzero stress asymmetry.

### 3.8. Spontaneous Curvature of Planar Bilayers

Another quantitative measure for the transbilayer asymmetry of a planar bilayer is provided by the first moment of the stress profile [24,25,27,101]. If one considers a tensionless bilayer with zero bilayer tension, Σ=0, the first moment of the stress profile can be interpreted as a torque per unit length, which is directly related to the product of two curvature-elastic parameters, the bending rigidity κ and the spontaneous curvature *m*. This relationship has the form [24,25,27,101]
(11)$$\int_{-\frac{1}{2}\;L_{z}}^{+\frac{1}{2}\;L_{z}}\;dz\;s\left(z\right)z=-2\kappa m\;\mathrm{for}\;\Sigma =\int_{-\frac{1}{2}\;L_{z}}^{+\frac{1}{2}\;L_{z}}\;dz\;s\left(z\right)=0\;.$$

The bending rigidity κ is always positive whereas the spontaneous curvature *m* can be positive or negative.

The constraint of zero bilayer tension, Σ=0, ensures that the first moment of the stress profile does not depend on the origin, $z=z_{0}$, of the *z*-coordinate. If Σ≠0, Equation (Equation 11) would introduce a $z_{0}$-dependence into the first moment of the stress profile [102]. The constraint of zero bilayer tension, $\Sigma =\Sigma_{ll}+\Sigma_{ul}=0$, implies that the leaflet tensions satisfy $\Sigma_{ul}=-\Sigma_{ll}$ which is the defining property of the bilayer’s OLT states. Therefore, the relationship in Equation (Equation 11) is only meaningful for OLT states, i.e., for those points $\left(\Sigma_{ll},\Sigma_{ul}\right)$ of the two-dimensional leaflet tension space in Figure 9 that are located on the perpendicular diagonal in this space. In contrast, the stress asymmetry ΔΣ in Equation (Equation 10) can be computed for all points $\left(\Sigma_{ll},\Sigma_{ul}\right)$ of the leaflet tension space. Thus, the stress asymmetry ΔΣ is applicable to any bilayer state whereas the interpretation of the first moment of the stress profile in terms of a torque per unit length is restricted to OLT states.

## 4. Planar Bilayers with Several Lipid Components

When a bilayer contains several lipid components, its elastic behavior depends on the lipid composition as well. For a two-component bilayer, the composition of this bilayer is uniquely described by the mole fraction of one component. However, both leaflets may differ in their composition, which implies that we need one mole fraction for each leaflet to characterize the lipid composition of both leaflets. For a three-component bilayer, we need to consider the mole fractions of two components to characterize the bilayer’s composition. When the two leaflets have different compositions, we need, in general, four composition variables to describe the three-component mixture in each of the two leaflets.

In this section, two relatively simple examples will be discussed. First, a bilayer with two lipid components that differ in the size of their headgroups [25]. Second, a bilayer with three lipid components, one of which undergoes frequent flip-flops [27].

### 4.1. Two Lipid Components with Small and Large Headgroups

Glycolipids such as GM1 have bulky headgroups consisting of several monosaccharides. When these lipids are added to phospholipid bilayers, they generate large membrane curvatures even for small compositional asymmetries between the two leaflets of the bilayers. On the micrometer scale, these bilayer asymmetries lead to the spontaneous tubulation of giant vesicles as observed by optical microscopy [85,103].

We model a binary mixture of a phospholipid and a glycolipid using two types of coarse-grained lipids, small-head lipids and large-head lipids. The small-head lipids are identical with those discussed in the previous section and have a headgroup consisting of three H beads, see Figure 2 and Figure 10a. The large-head lipids have a larger headgroup which is built from six H beads, see Figure 10b. The large-head lipids provide a coarse-grained model for a glycolipid such as GM1. The two lipid species are displayed in Figure 10a.

#### 4.1.1. Bilayer and Leaflet Compositions

To focus on the bilayer asymmetry arising from the different compositions of the two leaflets of the membrane, both leaflets are taken to contain the same number of lipids but different mole fractions of the two lipids [25]. The numbers of small-head and large-head lipids in the upper leaflet are denoted by $N_{ul}^{\mathrm{SH}}$ and $N_{ul}^{\mathrm{LH}}$, the corresponding numbers in the lower leaflet by $N_{ll}^{\mathrm{SH}}$ and $N_{ll}^{\mathrm{LH}}$. The imposed constraint on the lipid numbers has the form
(12)$$N_{ul}^{\mathrm{SH}}+N_{ul}^{\mathrm{LH}}=N_{ll}^{\mathrm{SH}}+N_{ll}^{\mathrm{LH}}\;.$$

For the simulations described in this section, lipid flip-flops between the leaflets were absent and the constraint in Equation (Equation 12) was conserved over the whole run time of the simulations. The mole fractions of the large-head lipid are given by
(13)$$\phi_{ll}^{\mathrm{LH}}\equiv \frac{N_{ll}^{\mathrm{LH}}}{N_{ll}^{\mathrm{SH}}+N_{ll}^{\mathrm{LH}}}\;\mathrm{in}\;\mathrm{the}\;\mathrm{lower}\;\mathrm{leaflet}$$
and by
(14)$$\phi_{ul}^{\mathrm{LH}}\equiv \frac{N_{ul}^{\mathrm{LH}}}{N_{ul}^{\mathrm{SH}}+N_{ul}^{\mathrm{LH}}}\;\mathrm{in}\;\mathrm{the}\;\mathrm{upper}\;\mathrm{leaflet}.$$

The two mole fractions $\phi_{ll}^{\mathrm{LH}}$ and $\phi_{ul}^{\mathrm{LH}}$ define another two-dimensional parameter space. The simulations described in [25] focussed on the one-dimensional subspace corresponding to $\phi_{ll}^{\mathrm{LH}}=0$ and variable $\phi_{ul}^{\mathrm{LH}}\geq 0$. This subspace includes the symmetric bilayer with $\phi_{ul}^{\mathrm{LH}}=\phi_{ll}^{\mathrm{LH}}=0$.

#### 4.1.2. Leaflet Tensions and Reference States

In order to determine the leaflet tensions of a two-component bilayer, one follows the same two-step procedure as for one-component bilayers. First, the stress profile s=s(z) across the bilayer is computed and then the position $z_{\mathrm{mid}}$ of the bilayer’s midplane, which divides the stress profile up into two leaflet contributions that determine the two leaflet tensions as in Equation (Equation 3). Using the CHAIN protocol for the position of the midplane, see Section 3.4.2, we obtained the leaflet tensions displayed in Figure 10c as a function of $\phi_{ul}^{\mathrm{LH}}$ for $\phi_{ll}^{\mathrm{LH}}=0$. The reference state with tensionless leaflets corresponds to the symmetric bilayer with $\phi_{ul}^{\mathrm{LH}}=\phi_{ll}^{\mathrm{LH}}=0$. In general, such a reference state with tensionless leaflets can be obtained for all symmetric bilayers with mole fractions $\phi_{ul}^{\mathrm{LH}}=\phi_{ll}^{\mathrm{LH}}$ by varying the base area of the simulation box for fixed lipid numbers $N_{ll}^{\mathrm{LH}}=N_{ul}^{\mathrm{LH}}$ and $N_{ll}^{\mathrm{SH}}=N_{ul}^{\mathrm{SH}}$.

#### 4.1.3. Small Compositional Asymmetries Generate Large Spontaneous curvatures

The simulations of two-component bilayers demonstrate that a small compositional asymmetry can generate a large spontaneous curvature of the bilayer. Using the first moment of the stress profile to determine the parameter combination κm in Equation (Equation 11) as well as independent simulations to obtain the bending rigidity κ, the spontaneous curvature *m* is found to increase linearly with the mole fraction $\phi_{ul}^{\mathrm{LH}}$ according to
(15)$$m=\phi_{ul}^{\mathrm{LH}}\;\frac{0.32}{d}\;\mathrm{for}\;\phi_{ul}^{\mathrm{LH}}\leq 0.6\;\mathrm{and}\;\phi_{ll}^{\mathrm{LH}}=0.$$

The positive value of *m* implies that the bilayer prefers to bulge towards the upper leaflet as one would expect intuitively because the large-head lipids in the upper leaflet occupy more space.

The spontaneous curvature *m* as given by Equation (Equation 15) is surprisingly large even for small compositional asymmetries. Using the value d=0.8 nm for the bead diameter, we obtain m=1/(250 nm) and m=1/(62.5nm) for mole fractions $\phi_{ul}^{\mathrm{LH}}=0.01$ and $\phi_{ul}^{\mathrm{LH}}=0.04$, respectively. For a GUV membrane, the latter value would lead to cylindrical nanotubes with a diameter of only 63 nm [85,104].

#### 4.1.4. Bending Rigidity and Fluctuation Spectrum of Bending Undulations

For symmetric bilayers, the bending rigidity κ can be obtained from a systematic analysis of the bilayer’s shape fluctuations. For a symmetric bilayer with one lipid component, the deduced κ-value was found to satisfy the simple relationship [24,36]
(16)$$\kappa =\frac{1}{48}K_{A}\ell_{\mathrm{me}}^{2}$$
with the area compressibility modulus $K_{A}$ as defined in Equation (Equation 9), and the bilayer thickness *ℓ* which is typically between 4 and 5 nm. The relationship as given by Equation (Equation 16) has been criticized by other groups who claimed that the prefactor 1/48 should be replaced by 1/24 [84,105].

The numerical coefficient 1/48 in Equation (Equation 16) was derived in [36] using classical elasticity theory for thin solid-like films and considering the limit in which the two-dimensional shear modulus vanishes. Using a polymer brush model, Rawicz et al. [84] obtained the same relationship as in Equation (Equation 16) but with the numerical coefficient 1/48 replaced by 1/24. However, the simulation study of a one-component bilayer [36] confirmed the value 1/48 within an accuracy of about 10 percent by measuring the three parameters κ, $K_{A}$, and $\ell_{\mathrm{me}}$ independently.

To corroborate the accuracy of the relationship in Equation (Equation 16), we also studied the shape fluctuations of symmetric and tensionless SH/LH bilayers with different values of the mole fraction $\phi_{ul}^{\mathrm{LH}}=\phi_{ul}^{\mathrm{LH}}\equiv \phi_{\mathrm{le}}$, using the Fourier mode analysis introduced in [36]. The resulting fluctuation spectra are displayed in Figure 11 as a function of wavenumber *q* for four different values of $\phi_{\mathrm{le}}$. All of these spectra are well fitted by the expression [24,36]
(17)$$S\left(q\right)\approx \frac{k_{B}T}{\kappa q^{4}}\;\mathrm{for}\;\mathrm{small}\;q\;\mathrm{and}\;\Sigma ≃0,\;$$
which depends only on a single parameter, the dimensionless bending rigidity $\kappa /(k_{B}T)$. Fitting the simulation data in Figure 11 to the functional form in Equation (Equation 17), we can deduce the bending rigidity κ as a function of the mole fraction $\phi_{\mathrm{le}}$. All of these κ-values satisfy the relationship in Equation (Equation 16) with the area compressibility $K_{A}$ and the bilayer thickness determined by independent data analysis. Therefore, the results of the Fourier mode analysis in Figure 11 agree with the prefactor 1/48 in Equation (Equation 16) but exclude the prefactor 1/24 as proposed in [84].

### 4.2. Three-Component Bilayer with Flip-Flopping Lipid Component

The two leaflets of a biomembrane typically differ in their lipid composition. Each lipid molecule stays within one leaflet of the bilayer before it undergoes a transition or flip-flop to the other leaflet. The corresponding flip-flop times are very different for different lipid components and vary over several orders of magnitude. Here, we consider a lipid bilayer with two phospholipid components that do not undergo flip-flops on the accessible time scales and thus may exhibit a wide range of leaflet tensions and stress asymmetries. When we now add a third lipid component that undergoes frequent flip-flops, these flip-flops tend to reduce the stress asymmetry between the leaflets. Furthermore, bilayers with a compositional asymmetry can acquire a significant spontaneous curvature even if both leaflets are tensionless.

#### 4.2.1. Relaxation of Leaflet Tensions

To comprehend the relaxation of the leaflet tensions, let us consider a planar and tensionless bilayer with Σ=0 which implies, in general, that the two leaflet tensions satisfy $\Sigma_{ll}=-\Sigma_{ul}$ as in Equation (Equation 24). As a consequence, one leaflet is stretched whereas the other leaflet is compressed. In such a situation, we can lower the elastic energy of the planar bilayer by reshuffling lipids (or other membrane molecules) from the compressed leaflet to the stretched one, thereby reducing both the positive tension in the stretched leaflet and the absolute value of the negative tension in the compressed leaflet. For a sufficiently large number of reshuffled lipids, we may then reach a bilayer state in which both leaflet tensions $\Sigma_{ul}$ and $\Sigma_{ll}$ vanish.

The relaxation process just described as a steered reshuffling of molecules between the two leaflets can also occur spontaneously if a molecular component is added to the bilayer that undergoes frequent flip-flops between the two leaflets. The latter process has been observed in molecular dynamics simulations for a lipid bilayer that contained the phospholipid POPC and the glycolipid (ganglioside) GM1, both of which did not undergo flip-flops, and, in addition, a model cholesterol, which moved frequently from one leaflet to the other on the time scale of the simulations, as schematically shown in Figure 12 [27].

The corresponding simulation data are displayed in Figure 13. The upper leaflet of the asymmetric bilayer contains 66 POPCs and 24 GM1s whereas the lower leaflet is composed of 87 POPCs. In addition, the bilayer contains 20 cholesterols that undergo frequent flip-flops between the two leaflets. As shown in Figure 13a, these cholesterol molecules are distributed in an asymmetric manner between the two leaflets, with an average number of 9 cholesterols in the upper leaflet and of 11 cholesterols in the lower leaflet. In this way, the flip-flopping cholesterol ensures that both leaflets are tensionless. The time-dependent relaxation towards these tensionless leaflets is depicted in Figure 13b. After the first 500 ps, the decay curve is well fitted by a single exponential with a time constant of 55 ns.

#### 4.2.2. Spontaneous Curvature of Two-Component Bilayers with Tensionless Leaflets

The reference state of a one-component bilayer with vanishing leaflet tension implies that the bilayer is symmetric and that the stress profile satisfies s(−z)=s(z) when we choose the *z*-coordinate in such a way that the midplane is located at $z_{\mathrm{mid}}=0$, see Figure 7a. Therefore, the spontaneous curvature *m*, which is proportional to the first moment of the stress profile, must vanish as well for the reference state of a one-component bilayer. Intuititively, one might expect that the spontaneous curvature vanishes for all bilayers with vanishing leaflet tensions. This expectation agrees with the view, originally proposed by Bancroft for surfactant monolayers in water-oil emulsions [106,107], that a thin fluid layer is bounded by two interfaces and that these interfaces typically differ in their tensions. Such a layer should have a tendency to bend or bulge towards the interface with the lower tension, because the layer can then reduce the area of the other interface with the higher tension.

The leaflet tensions $\Sigma_{ll}$ and $\Sigma_{ul}$ are defined by the partial integrals over the stress profile s=s(z) as given by Equation (Equation 3). On the other hand, the spontaneous curvature is proportional to the first moment of the stress profile, see Equation (Equation 11), and this first moment can be finite for an asymmetric planar bilayer even if both leaflet tensions vanish. The latter behavior has indeed been observed in molecular dynamics simulations for two-component bilayers containing the phospholipid POPC and the glycolipid (ganglioside) GM1. When these bilayers have a compositional asymmetry, they acquire a significant spontaneous curvature even if both leaflets are tensionless as shown in Figure 14 [27]. Therefore, for a multi-component bilayer, the reference state with tensionless leaflets can possess a nonzero spontaneous curvature.

## 5. Instabilities of Planar Bilayers with Large Stress Asymmetries

The planar bilayers discussed so far experienced moderate leaflet tensions $\Sigma_{ll}$ and $\Sigma_{ul}$, see Figure 9, and thus small stress asymmetries $\Delta \Sigma =\Sigma_{ul}-\Sigma_{ll}$. In this regime of small ΔΣ, the phospholipids do not undergo flip-flops on the time scales of the simulations. This stability of the lipid bilayers changes when we consider an extended range of leaflet tensions and larger stress asymmetries. Indeed, for sufficiently large stress asymmetries, the phospholipids undergo flip-flops between the two leaflets and the bilayers exhibit structural instabilities even for vanishing bilayer tension.

### 5.1. Stability Regime for Planar Bilayers with One Lipid Component

To avoid bilayer poration and membrane rupture, each bilayer is kept in an OLT state with (almost) zero bilayer tension by adjusting the total number of lipids assembled in the bilayer to the size of the simulation box. We consider such OLT states for one-component bilayers with a total number of 1682 lipids assembled in both leaflets. Such a bilayer is symmetric when each leaflet consists of 841 lipids but becomes asymmetric when the two leaflets contain different number of lipids. When the upper leaflet contains $N_{ul}=870$ lipids and the lower leaflet $N_{ll}=812$ lipids, for example, the upper leaflet is compressed and experiences a negative leaflet tension $\Sigma_{ul}<0$ whereas the lower leaflet is stretched and subject to a positive leaflet tension $\Sigma_{ll}>0$, see Figure 15. To simplify the mathematical formulas, we will use the short-hand notation $\gamma \equiv k_{B}T/d^{2}$ for the basic tension unit.

During the simulation time of 12.5μs, the lipids did not undergo any flip-flops in the stability regime which is characterized by lipid numbers $N_{ul}$ within the range $945\geq N_{ul}\geq 737$ and by upper leaflet tensions $\Sigma_{ul}$ with $-1.86\;\gamma \leq \Sigma_{ul}\leq +2.00\;\gamma$. Flip-flops from the upper to the lower leaflet were observed, however, for $N_{ul}\geq 957$, corresponding to the left instability regime in Figure 15. For $N_{ul}=957$, which defines the left instability line in Figure 15, the upper leaflet was initially compressed with $\Sigma_{ul}=-2.21\;\gamma$, which induced flip-flops from the compressed upper to the stretched lower leaflet. As the lipid number in the upper leaflet was increased to $N_{ul}>957$, that is, into the left instability regime in Figure 15, flip-flops became more frequent. On the other hand, for $N_{ul}\leq 725$ and $\Sigma_{ul}\geq 2.13\;\gamma$, which defines the right instability region in Figure 15, lipids underwent flip-flops from the compressed lower leaflet to the stretched upper one.

### 5.2. Stress-Induced Flip-Flops of Lipids in Planar Bilayers

To determine the kinetic rates of lipid flip-flops, we considered several ensembles of bilayers. Each ensemble consisted of more than 120 bilayers that were assembled from the same lipid numbers $N_{ul}$ and $N_{ll}$ and, thus, experienced the same leaflet tensions as long as they remained in their initial metastable states. More precisely, the bilayers experienced the same initial leaflet tensions until time $t_{1}$, at which the first flip-flop occurred. The statistics of $t_{1}$ can be described by the cumulative distribution function $P_{\mathrm{cdf}}\left(t\right)$ that represents the probability that the first flip-flop occurs at time $t_{1}\leq t$. From a numerical point of view, the cumulative distribution function $P_{\mathrm{cdf}}\left(t\right)$ is more reliable than the probability density function $dP_{\exp}/dt$ because, in contrast to the density function, the cumulative distribution function $P_{\mathrm{cdf}}\left(t\right)$ does not require any binning of the data. Furthermore, in the present context, the complementary probability distribution $1-P_{\mathrm{cdf}}\left(t\right)$ represents the survival probability of the metastable bilayer state without any flip-flops up to time *t*.

The measured cumulative distribution functions $P_{\mathrm{cdf}}\left(t\right)$ are displayed in Figure 16 for planar bilayers with $N_{ul}=986,1015$, and 1073 lipids in the upper leaflet and $N_{ll}=1682-N_{ul}$ lipids in the lower leaflet, corresponding to the black circles, red squares, and blue diamonds, respectively. All three data sets can be well fitted by a cumulative distribution function of the form
(18)$$P_{\mathrm{cdf}}\left(t\right)=1-\exp\left(-\omega_{\mathrm{pl}}\;t\right)\equiv P_{\exp}\left(t\right)\;,$$
which involves only a single fit parameter, the flip-flop rate $\omega_{\mathrm{pl}}$ for planar bilayers. The inverse rate $\omega_{\mathrm{pl}}^{-1}$ is equal to the average first flip-flop time $\langle t_{1}\rangle$, which also represents the average lifetime of the metastable bilayer states. Note that the exponential distribution in Equation (Equation 18) vanishes for t=0 and approaches the limiting value $P_{\exp}\left(t\right)\approx 1$ for large *t*, as required for any cumulative distribution function. The associated probability density function $dP_{\exp}/dt$ is normalized to one.

### 5.3. Stress-Induced Instability and Self-Healing of Planar Bilayers

In addition to flip-flops of the lipid molecules, the bilayers undergo structural instabilities outside of the stability regime in Figure 15, that is, when the lipid number $N_{ul}$ is increased beyond the left instability line at $N_{ul}=957$ or decreased beyond the right instability line at $N_{ul}=725$. One example for such a structural instability is shown in Figure 17. In this example, the bilayer was initially assembled with $N_{ul}=986$ lipids in the upper leaflet and $N_{ll}=1682-N_{ul}=696$ lipids in the lower leaflet.

After the initial assembly, the bilayer remained in a metastable state without flip-flops for hundreds of nanoseconds and bulged towards the upper leaflet, see the simulation snapshot after 200 ns in Figure 17a, before it became unstable. After 1000 ns, the upper leaflet has expelled about 100 lipids that form a globular micelle, see Figure 17b. The micelle consists of red-green lipids that were initially assembled in the compressed upper leaflet. Along the contact line between micelle and bilayer segment, we observed frequent exchanges of lipids towards the lower leaflet as can be concluded from Figure 17c. This lipid exchange increased the number of lipids in the lower leaflet and decreased the number of lipids in the upper leaflet. Finally, after 1700 ns, the conventional bilayer structure has been restored, see Figure 17d, where the lower leaflet now contains both the blue-purple lipids, from which this leaflet was initially assembled, and 93 red-green lipids that moved from the upper to the lower leaflet. After this self-healing process, the bilayer remained again stable without flip-flops until the end of the simulations. Note that the expelled lipids in Figure 17b form a micelle rather than a bilayer enclosing a certain amount of water.

## 6. Nanovesicles with One Lipid Component

Nanovesicles are closed, bubble-like surfaces with a diameter between 20 and 200 nm, formed by synthetic and cell-derived lipid bilayers. Electron microscopy (EM) studies have shown that these vesicles can attain both spherical and nonspherical shapes. One disadvantage of EM methods is that they provide only a single snapshot of each vesicle. In contrast, molecular dynamics simulations can reveal the spatio-temporal remodeling of each individual nanovesicle. We start with the assembly of spherical vesicles that enclose a certain volume of water and contain a certain total number of lipids. When we reduce their volume, the spherical vesicles transform into a multitude of nonspherical shapes such as oblates and stomatocytes as well as prolates and dumbbells. This surprising polymorphism can be controlled by redistributing a small fraction of lipids between the inner and outer leaflets of the bilayer membranes, which then experience different leaflet tensions.

### 6.1. Assembly of Lipids into Spherical Nanovesicles

Spherical vesicles were assembled by placing lipid molecules onto two spherical shells corresponding to the two leaflets of the vesicle bilayers. The size of these vesicles was primarily determined by the vesicle volume, that is, by the number of water beads enclosed by the inner leaflet of the membrane. This number was chosen in such a way that the headgroup layers of the inner and outer leaflets had a diameter of about 45d and 50d, respectively. For a given volume, we placed $N_{\mathrm{il}}$ and $N_{\mathrm{ol}}$ lipids onto the inner and outer leaflets, keeping the total lipid number $N_{\mathrm{lip}}=N_{\mathrm{il}}+N_{\mathrm{ol}}=10100$ fixed. Thus, for given volume and constant total lipid number, we are left with a single assembly parameter, which we take to be the lipid number $N_{\mathrm{ol}}$ in the outer leaflet. The spherical vesicles assembled in this manner were found to be stable for $N_{\mathrm{ol}}$-values within the range $5700\leq N_{ol}\leq 6300$. The lipid numbers $N_{il}=10100-N_{ol}$ in the inner leaflet are then given by $4400\geq N_{il}\geq 3800$.

Experimentally, the volume of a vesicle can be changed by osmotic deflation and inflation. In the simulations, the vesicle volume was varied by changing the number $N_{W}$ of water beads enclosed by the inner leaflet of the vesicle membrane. This number determines the volume $V\equiv N_{W}\;d^{3}/3$ where the factor 1/3 reflects the bulk water density $\rho_{W}=3/d^{3}$. To monitor the volume changes, the rescaled volume ν was used which is defined by
(19)$$\nu \equiv \frac{N_{W}}{N_{W}^{\mathrm{isp}}}$$
where $N_{W}^{\mathrm{isp}}$ is the number of water beads enclosed by the initial spherical vesicle. Thus, the initial vesicle is characterized by ν=1 and any volume reduction with $N_{W}<N_{W}^{\mathrm{isp}}$ leads to ν<1. Monitoring volume changes via the parameter ν is quite convenient here because we can directly change the number $N_{W}$ of water beads within the vesicle and thus compute the value of ν without the necessity to determine any membrane surface.

### 6.2. Bilayer Tension and Leaflet Tensions of Vesicle Bilayers

For a spherical nanovesicle, the density and stress profiles across the vesicle bilayer depend on the radial coordinate *r*, with r=0 corresponding to the center of the spherical shape. Integrating the stress profile s=s(r) over the radial coordinate, we obtain the bilayer tension
(20)$$\Sigma =\int_{0}^{r_{\max}}dr\;s\left(r\right)\;,$$
where $r_{\max}$ is an appropriate upper limit for the integration. In practice, one can reduce the integration over *r* to a small interval because the stress profile s=s(r) decays rapidly to zero as we move away from the bilayer.

In order to compute the leaflet tensions $\Sigma_{il}$ and $\Sigma_{ol}$ of the inner and outer leaflet, we introduce the average position $r=r_{\mathrm{mid}}$ of the leaflet-leaflet interface, which defines the midsurface of the vesicle bilayer. The leaflet tension $\Sigma_{il}$ and $\Sigma_{ol}$ of the inner and outer leaflet are then given by
(21)$$\Sigma_{il}=\int_{0}^{r_{\mathrm{mid}}}\;dr\;s\left(r\right)\;\mathrm{and}\;\Sigma_{ol}=\int_{r_{\mathrm{mid}}}^{r_{\max}}\;dr\;s\left(r\right)\;,$$
which represents the decomposition of the bilayer tension Σ in Equation (Equation 20) according to $\Sigma =\Sigma_{il}+\Sigma_{ol}$.

### 6.3. Midsurface of Leaflet-Leaflet Interface for Nanovesicles

#### 6.3.1. Midsurface from Density Profile of Hydrophobic Chains

The simplest procedure to determine the midsurface radius $r=r_{\mathrm{mid}}$ is again provided by the CHAIN protocol, that is, by placing $r_{\mathrm{mid}}$ at the maximum of the hydrophobic chain density. Using this procedure, one obtains the leaflet tensions displayed in Figure 18 versus the lipid number $N_{ol}$ in the outer leaflet, for constant total number of lipids, $N_{ol}+N_{il}=10,100$. The data in Figure 18a were obtained for vesicle volume ν=1, using $N_{W}^{\mathrm{isp}}=90,400$ for the initial number of interior water beads. The data in Figure 18b display the leaflet tensions that were computed for tensionless vesicle bilayers after slightly deflating the vesicles to $\nu =\nu_{0}<1$. For the five values of $N_{ol}$ displayed in Figure 18b, the nanovesicles with vanishing bilayer tension, Σ=0, have rescaled volumes $\nu_{0}$ within the range $0.966\leq \nu_{0}\leq 0.978$ [28]. All OLT states of the nanovesicles are obtained from the reference state with tensionless leaflets by reshuffling lipids from one leaflet to the other and adjusting the rescaled volume ν of the vesicle to $\nu =\nu_{0}$ in order to obtain tensionless bilayers, see also Section 6.6 below.

Interpolating the two sets of data in panel a and b of Figure 18, one can obtain two estimates for the reference state of the vesicle bilayer with tensionless leaflets. The data in Figure 18a lead to the estimate $N_{ol}=N_{ol}^{*}=5963$ lipids and $N_{il}=N_{il}^{*}=10,100-N_{ol}^{*}=4137$ lipids for the outer and inner leaflets of the reference state. The data in Figure 18b lead to the slightly different lipid numbers $N_{ol}=N_{ol}^{*}=5993$ and $N_{il}=N_{il}^{*}=10,100-N_{ol}^{*}=4107$. Both estimates imply that the reference state with tensionless leaflets is characterized by very different lipid numbers $N_{ol}^{*}$ and $N_{il}^{*}$ in the two lealets of the vesicle bilayer.

#### 6.3.2. Midsurface from Volume per Lipid via Voronoi Tesselation

The Voronoi tesselation for the bead volumes as described in Section 3.4.3 for planar bilayers can be directly extended to nanovesicles in water, by assigning again a polyhedral Voronoi cell to each bead of the molecular model. The volumes of the inner and outer leaflets of the vesicle bilayer are denoted by $V_{il}$ and $V_{ol}$ and are computed by summing up all bead volumes as in the case of planar bilayers, see Figure 8. The volumes per lipid of the inner and outer leaflets are then obtained by dividing the leaflet volumes by the number of lipids, $N_{il}$ and $N_{ol}$.

To compute the radial coordinate $r=r_{\mathrm{mid}}$ for the midsurface position of the vesicle bilayer, we consider a cubic simulation box with volume $L^{3}$, which is divided up into two separate water compartments by the closed surface of the vesicle, the inner water compartment with volume $V_{iW}$ and the outer water compartment with volume $V_{oW}$. The midsurface radius $r=r_{\mathrm{mid}}$ of the vesicle bilayer can then be computed using two geometric relationships. The first geometric relationship has the form
(22)$$\frac{4\pi }{3}r_{\mathrm{mid}}^{3}=V_{il}+V_{iW}\;,$$
and relates the midsurface radius $r_{\mathrm{mid}}$ to the volume $V_{il}$ of the inner leaflet and the volume $V_{iW}$ of the inner water compartment. The second geometric relationship is given by
(23)$$\frac{4\pi }{3}r_{\mathrm{mid}}^{3}=L^{3}-V_{oW}-V_{ol}\;,$$
which depends on the volume $L^{3}$ of the simulation box, the volume $V_{oW}$ of the outer water compartment, and the volume $V_{ol}$ of the outer leaflet. The two relationships in Equations (Equation 22) and (Equation 23) give identical values for $r_{\mathrm{mid}}$ within the numerical accuracy.

### 6.4. Shape Transformations of Nanovesicles by Changes of Vesicle Volume

Panels a and b of Figure 18 display the leaflet tensions $\Sigma_{il}$ and $\Sigma_{ol}$ of four and five different nanovesicles corresponding to four and five different values of $N_{ol}$ and $N_{il}=10,100-N_{ol}$. All nanovesicles have a spherical shape, both for ν=1 and for $\nu =\nu_{0}<1$, which implies that we cannot draw any conclusions about their leaflet tensions by looking at the shapes of the vesicles. This situation changes drastically as soon as we monitor the response of these vesicles to a reduction of their volume, mimicking the experimental procedure of osmotic deflation.

When we deflate the spherical vesicle with $N_{ol}=6300$ lipids in the outer leaflet, we observe the shape transformations displayed in Figure 19. The vesicle transforms into a dumbbell shape, which exhibits a closed membrane neck for ν≤0.9 and an out-bud that grows in size as the volume is decreased from ν=0.9 to ν=0.7. From the shape transformations in Figure 19, we can conclude that the inner leaflet of the spherical vesicle was stretched whereas its outer leaflet was compressed, in agreement with the data in Figure 18 for $N_{ol}=6300$.

On the other hand, deflating the spherical vesicle with $N_{ol}=5700$, a very different sequence of shapes is observed as shown in Figure 20. Now, the deflated vesicle develops an in-bud rather than an out-bud. Therefore, we can conclude that, before deflation, the inner leaflet of the spherical vesicle was compressed whereas its outer leaflet was stretched, in agreement with the data in Figure 18 for $N_{ol}=5700$. Therefore, the morphological responses of the spherical vesicles in Figure 19 and Figure 20 allow us to draw some definite conclusions about the compressed or stretched state of the two leaflets.

As a third example, let us consider the deflation-induced shape transformation of a spherical vesicle with $N_{ol}=5963$ lipids as displayed in Figure 21. The latter $N_{ol}$-value is obtained for the reference state with tensionless leaflets, using the interpolation in Figure 18a. In fact, we conclude from the data in Figure 18b that the tensionless bilayer of this vesicle has a slightly stretched outer leaflet and a slightly compressed inner leaflet.

### 6.5. Fission of Membrane Neck for In-Budded Nanovesicles

The deflation of a spherical nanovesicle with $N_{ol}=5700$ leads to a stomatocyte shape with an open membrane neck as shown in Figure 20. By reshuffling 200 lipids from the outer to the inner leaflet, we obtain a new initial vesicle state with $N_{ol}=5500$, which transforms into a stomatocyte shape with a closed neck that undergoes membrane fission within about 15μs, see Figure 22. As a result of this fission process, the vesicle is divided up into two nested daughter vesicles, which adhere to each other.

### 6.6. Two-Dimensional Leaflet Tension Space for Nanovesicles

The two leaflet tensions $\Sigma_{il}$ and $\Sigma_{ol}$ define a two-dimensional parameter space for nanovesicles. This space is depicted in Figure 23 for vesicle bilayers with a total number of $N_{ll}+N_{ul}=2525$ lipids. The origin $\left(\Sigma_{il},\Sigma_{ol}\right)=\left(0,0\right)$ of this leaflet tension space defines the relaxed reference state with tensionless leaflets. The reference state with tensionless leaflets, corresponding to $\Sigma_{il}=\Sigma_{ol}=0$, is obtained for a vesicle bilayer with $N_{il}=840$ lipids in the inner and $N_{ol}=1685$ lipids in the outer leaflet [32]. Thus, in contrast to planar bilayers, the reference state for nanovesicles cannot be obtained by symmetry arguments.

To characterize the elastic response of the reference state, it will now be useful to distinguish two types of elastic deformations, corresponding to the black and green data in Figure 23. The black data in Figure 23 correspond to elastic deformations of the reference state with
(24)$$\Sigma_{ll}=-\Sigma_{ul}\;\mathrm{or}\;\mathrm{opposite}\;\mathrm{leaflet}\;\mathrm{tensions}\;(\mathrm{OLTs}).$$

The OLT states are located on the perpendicular diagonal which is orthogonal to the main diagonal in Figure 23. All OLT states can be obtained from the reference state by reshuffling lipids from one leaflet to the other, keeping the total lipid number $N_{ll}+N_{ul}$ constant and imposing the constraint of vanishing bilayer tension $\Sigma =\Sigma_{ll}+\Sigma_{ul}=0$. As a consequence, one leaflet is compressed by a negative leaflet tension whereas the other leaflet is stretched by a positive leaflet tension.

The green data in Figure 23 represent the leaflet tensions arising from changes in the vesicle volume, corresponding to vesicle inflation or deflation (VID). To generate the green VID data, we start again from the reference state with $\Sigma_{il}=\Sigma_{ol}=0$ but now change the vesicle volume in order to increase or decrease the bilayer tension, thereby mimicking the experimental procedure of osmotic inflation or deflation. It is interesting to note that the green data do not follow the main diagonal. Therefore, during a VID step, both leaflet tensions are changed by different amounts.

Figure 23 does not display data for ELT deformations with $\Sigma_{il}=\Sigma_{ol}$. In order to obtain an ELT deformation, we could start from one of the VID data and add another step, in which lipids are again reshuffled between the leaflets. However, compared to the OLT deformations, the lipid reshuffling has to be iterated several times because we need to change both leaflet tensions in different ways. Thus, during each of these iterative steps, we would have to vary the lipid numbers, $N_{il}$ and $N_{ol}$, in the two leaflets, determine the corresponding midsurface of the vesicle bilayer, and compute the two leaflet tensions from the decomposition of the stress profile, a somewhat tedious and time-consuming procedure.

### 6.7. Stress Asymmetry of Vesicle Bilayers

The stress asymmetry $\Delta \Sigma_{\mathrm{ve}}$ between the two leaflet tensions of a vesicle bilayer is defined by [30,108]
(25)$$\Delta \Sigma_{\mathrm{ve}}\equiv \Sigma_{ol}-\Sigma_{il}=\int_{r_{\mathrm{mid}}}^{r_{\max}}\;dr\;s\left(r\right)-\int_{0}^{r_{\mathrm{mid}}}\;dr\;s\left(r\right)$$
where the second equality follows from the integral expressions for the leaflet tensions in Equation (Equation 21).

The stress asymmetry ΔΣ in Equation (Equation 25) vanishes for all ELT states of the vesicle bilayer. However, these ELT states for nanovesicles are not easy to characterize in terms of the corresponding lipid numbers $N_{il}$ and $N_{ol}$ of the inner and outer leaflets because, in contrast to planar bilayers, these lipid numbers are not related by any symmetry. Thus, for vesicle bilayers with $N_{il}+N_{ol}=2525$ as depicted in Figure 23, the reference state with tensionless leaflets is obtained for $N_{il}=840$ lipids in the inner and $N_{ol}=1685$ lipids in the outer leaflet.

All OLT states of the vesicle bilayer, corresponding to the black data in Figure 23, exhibit a nonzero stress asymmetry $\Delta \Sigma_{\mathrm{ve}}$ because the OLT states are characterized by one stretched and one compressed leaflet. In fact, apart from the ELT states along the main diagonal of the two-dimensional leaflet tension space in Figure 23, all points $\left(\Sigma_{il},\Sigma_{ol}\right)$ of this space lead to a nonzero stress asymmetry.

### 6.8. Spontaneous Curvature of Vesicle Bilayers

For a spherical vesicle with mean curvature *M*, bounded by a tensionless bilayer with Σ=0, the spontaneous curvature *m* satisfies the relation [28,102]
(26)$$2\kappa \left(M-m\right)=\int_{0}^{r_{\max}}dr\;s\left(r\right)\;r\;\mathrm{for}\;\Sigma =\int_{0}^{r_{\max}}dr\;s\left(r\right)=0\;,$$
which involves the bending rigidity κ and the first moment of the stress profile s(r). The constraint of vanishing bilayer tension, Σ=0, ensures that we can interpret the first moment of the stress profile as a torque per unit length that is applied to the midsurface of the vesicle bilayer but does not depend on the radius $r_{\mathrm{mid}}$ of this midsurface. Because of this constraint, the spontaneous curvature *m* as given by Equation (Equation 26) is only defined for the OLT states of the vesicle bilayer, corresponding to the black data of Figure 23. For mean curvature M=0, Equation (Equation 26) reduces to Equation (Equation 11) for planar bilayers.

## 7. Instabilities of Nanovesicles with Large Stress Asymmetries

The vesicle bilayers discussed so far experienced moderate leaflet tensions $\Sigma_{il}$ and $\Sigma_{ol}$, see Figure 23, and relatively small stress asymmetries $\Delta \Sigma =\Sigma_{ol}-\Sigma_{il}$. For small ΔΣ, the phospholipids do not undergo flip-flops on the time scales of the simulations. This stability of the vesicle bilayers changes when we consider larger leaflet tensions and larger stress asymmetries. Indeed, for sufficiently large stress asymmetries, the phospholipids undergo flip-flops between the two leaflets and the vesicle bilayers exhibit structural instabilities even for vanishing bilayer tension.

### 7.1. Stability Regimes of Spherical Nanovesicles

To illustrate the stability and instability of vesicle bilayers, we consider bilayers that are assembled from a fixed total number of 2875 lipids, with $N_{il}$ lipids in the inner leaflet and $N_{ol}=2875-N_{il}$ lipids in the outer one. We focus again on tensionless bilayers, for which the bilayer tension $\Sigma =\Sigma_{il}+\Sigma_{ol}$ is close to zero. Using the CHAIN protocol for the average position of the leaflet-leaflet interface, we obtain the leaflet tensions $\Sigma_{il}$ and $\Sigma_{il}$ as plotted in Figure 24 as functions of the lipid number $N_{ol}$ in the outer leaflet.

Inspection of Figure 24 shows that both leaflet tensions vanish when the outer leaflet contains between 1893 and 1935 lipids. Linear interpolation then leads to tensionless leaflets for $N_{ol}^{*}=1921$ lipids in the outer leaflet and $N_{il}^{*}=954$ lipids in the inner one. Thus, for these nanovesicles, the lipid number in the tensionless outer leaflet is more than twice as large as the one in the tensionless inner leaflet. As a consequence, the area per lipid in the tensionless outer leaflet is smaller than the area per lipid in the tensionless inner leaflet, which implies that the inner leaflet is more loosely packed than the tensionless outer leaflet.

When we start from the reference state with tensionless leaflets and reshuffle some lipids between these leaflets, we obtain nonzero leaflet tensions $\Sigma_{ol}$ and $\Sigma_{il}$. The two leaflets form bilayers that remain stable for the outer lipid number range $1775\leq N_{ol}\leq 2095$ and for the outer leaflet tension range $1.60\;k_{B}T/d^{2}\leq \Sigma_{ol}\leq -1.94\;k_{B}T/d^{2}$, see Figure 24. Flip-flops from the compressed inner to the stretched outer leaflet are observed for $N_{ol}\leq 1755$ and $\Sigma_{ol}\geq +1.78\;\gamma$, corresponding to the right shaded region in Figure 24. Flip-flops from the compressed outer to the stretched inner leaflet occur for $N_{ol}\geq 2105$ and $\Sigma_{ol}\leq -1.97\;\gamma$, which defines the left shaded region in Figure 24. Within these two instability regimes with $N_{ol}<1755$ and $N_{ol}>2105$, we also observe structural instabilities of the lipid bilayers in addition to the flip-flops.

### 7.2. Stress-Induced Flip-Flops between Leaflets of Nanovesicles

We now focus on the left instability regime in Figure 24, corresponding to compressed outer leaflets with $\Sigma_{ol}<0$ and stretched inner leaflets with $\Sigma_{il}>0$. Within this instability region, the lipids undergo flip-flops from the outer to the inner leaflets. To determine the kinetic rates, we examined again different ensembles of nanovesicles. Each ensemble consisted of more than 60 vesicles that were initially assembled from the same lipid numbers $N_{ol}$ and $N_{il}$ and, thus, experienced the same leaflet tensions as long as they remained in their initial states. More precisely, the bilayers experienced the same initial leaflet tensions until time $t_{1}$, at which the first flip-flop occurred. As in the case of planar bilayers, see Section 5.2, the statistics of $t_{1}$ is again described by the cumulative distribution function $P_{\mathrm{cdf}}\left(t\right)$ that represents the probability that the first flip-flop occurs at time $t_{1}\leq t$. This distribution function is depicted in Figure 25.

The cumulative distribution functions in Figure 25 have a sigmoidal shape with a point of inflection at intermediate times. This sigmoidal shape is qualitatively different from the exponential distribution in Equation (Equation 18) that was used to fit the onset of flip-flops in planar bilayers. Distribution functions with a sigmoidal shape can be obtained by generalizing the exponential distributions to Weibull distributions as provided by [109]
(27)$$P_{\mathrm{Wei}}\left(t\right)=1-\exp\left[-{\left(\omega_{\mathrm{ve}}t\right)}^{k}\right]\;,$$
which involve stretched exponentials. The Weibull distributions depend on two parameters, the rate parameter $\omega_{\mathrm{ve}}$ and the dimensionless shape parameter k>0. For k=1, the Weibull distribution in Equation (Equation 27) becomes identical to the exponential distribution in Equation (Equation 18). For k≠1, the empirical Weibull distribution has been applied to a large variety of different processes [110,111].

The inset of Figure 25 shows that the rate parameter $\omega_{\mathrm{ve}}$ increases monotonically with the absolute value of the stress asymmetry between the two leaflet tensions. This behavior demonstrates that the leaflet tensions and the associated stress asymmetry represent key parameters for the lipid flip-flops between the two bilayer leaflets.

In addition, the shape parameter *k* is found to be greater than one [30] which corroborates the sigmoidal shape of the distributions and implies that the instantaneous flip-flop rate increases with the age of the metastable state. In mathematical statistics, the instantaneous rate is known as the hazard rate and equal to the ratio of the probability density function $dP_{\mathrm{cdf}}/dt$ to the survival probability $1-P_{\mathrm{cdf}}\left(t\right)$ [112,113]. The only distribution that leads to a constant and age-independent hazard rate is the exponential distribution. Therefore, the sigmoidal shape of the cumulative distribution functions as observed here for the lifetime of the metastable bilayer states implies ageing of these nanovesicle states.

### 7.3. Stress-Induced Instability and Self-Healing of Nanovesicles

In addition to the flip-flops, the instability regimes in Figure 24 lead to structural instabilities followed by self-healing of the bilayers. One example is displayed in Figure 26 which corresponds to the left instability regime in Figure 24, characterized by a compressed outer and a stretched inner leaflet. The vesicle bilayer in Figure 26 is initially assembled from $N_{ol}=2105$ lipids in the outer and $N_{il}=770$ lipids in the inner leaflet. After adjusting the vesicle volume to obtain a tensionless bilayer, the outer leaflet is compressed by the negative leaflet tension $\Sigma_{ol}=-1.97\;k_{B}T/d^{2}$ whereas the inner leaflet is stretched by the positive leaflet tension $\Sigma_{il}=1.94\;k_{B}T/d^{2}$.

During the first 1300 ns, the bilayer of the vesicle undergoes shape fluctuations that appear to be ‘normal’ until the outer leaflet starts to develop a protrusion by expelling some lipids. This protrusion grows rapidly into a cylindrical micelle, which reaches its maximal extension after 1720 ns as shown in Figure 26b. Somewhat later, some lipids move into the inner leaflet, primarily along the contact line between the micelle and the bilayer, see snapshot at 2160 ns in Figure 26c. This lipid exchange decreases the number of lipids within the compressed outer leaflet and increases the number of lipids within the stretched inner leaflet until 111 red-green lipids have been moved from the outer to the inner leaflet and the ordered bilayer structure has been restored at 2710 ns as shown in Figure 26d. After this time point, the restored bilayer undergoes no further flip-flops until the end of the simulations.

## 8. Remodeling of Nanovesicles via the Adsorption of Small Solutes

In this section, we consider nanovesicles exposed to an exterior solution of small solutes, which adsorb onto the outer leaflet of the vesicle bilayers. The behavior of these systems depends on several key parameters. First, it depends on the key parameters that determine the vesicle behavior in the absence of solute. For one-component bilayers as considered here, these parameters are the lipid numbers $N_{il}$ and $N_{ol}$ in the inner and outer leaflet of the vesicle bilayer, the associated leaflet tensions, and the vesicle volume. The presence of solute contributes two additional key parameters, the solute concentration and the solvent conditions [29]. By definition, good solvent conditions imply that the solution remains spatially uniform for all solute concentrations. A poor solvent, on the other hand, leads to a certain range of solute concentrations, for which the solution undergoes liquid–liquid phase separation. Thus, in order to study the remodeling of the vesicles in a systematic manner, we will first discuss the phase diagram of the solution, which is described as a binary mixture of water and solute beads. The remodeling of the nanovesicles exposed to the adsorbing solutes will first be discussed for good solvent conditions. In this case, the behavior of the vesicles is qualitatively similar to their behavior in the absence of solute. In contrast, new morphological responses of the nanovesicles are observed for poor solvent conditions when the solute concentration is close to the binodal line of the phase diagram.

### 8.1. Phase Behavior of a Binary Solute-Water Mixture

A relatively simple model system for the solution is provided by a binary mixture consisting of water and solute molecules. The mixture is modeled in terms of water (W) and solute (S) beads, both of which represent small molecular groups. For computational simplicity, the two types of beads are taken to have the same diameter, *d*, and the interaction between two W beads is taken to be the same as the interaction between two S beads [26,29,31]. This symmetry implies that the phase diagram remains unchanged when we swap the W with the S beads and that this binary mixture has a particularly simple phase diagram as displayed in Figure 27. This binary mixture represents an off-lattice variant of the classical lattice gas model for binary mixtures.

The coordinates for the phase diagram in Figure 27 are the solute mole fraction $\Phi_{S}$ and the solubility ζ. The mole fractions $\Phi_{S}$ and $\Phi_{W}$ of solute and water are defined by
(28)$$\Phi_{S}=\frac{N_{S}}{N_{W}+N_{S}}\;\mathrm{and}\;\Phi_{W}=\frac{N_{W}}{N_{W}+N_{S}}=1-\Phi_{S}$$
where $N_{S}$ and $N_{W}$ are the numbers of solute and water beads within the exterior solution. The solubility ζ is defined in terms of the force parameters $f_{ij}$ for the interaction forces between the water (W) and solute (S) beads and given by [29]
(29)$$\zeta \equiv \frac{(f_{\mathrm{WW}}+f_{\mathrm{SS}})/2}{f_{\mathrm{WS}}}\;.$$
which involves the two force parameters $f_{\mathrm{WW}}$ and $f_{\mathrm{SS}}$ between a pair of water and solute beads as well as the force parameter $f_{\mathrm{WS}}=f_{\mathrm{SW}}$ for a water bead interacting with a solute bead. As we increase $f_{\mathrm{WS}}$, contacts between water and solute beads become energetically more costly. These contacts can be avoided by the segregation of the two types of beads. As a consequence, decreasing the solubility ζ by increasing $f_{\mathrm{WS}}$ leads to phase separation into a solute-poor and a solute-rich phase for intermediate values of the solute mole fraction $\Phi_{S}$.

Inspection of Figure 27 shows that the phase diagram is mirror symmetric with respect to $\Phi_{S}=1/2$. This symmetry implies horizontal tie lines, which are parallel to the $\Phi_{S}$-axis. The symmetry also implies that the critical demixing point is located at $\Phi_{S}=\Phi_{W}=1/2$. In the next subsections, we will focus on two values of the solubility ζ, corresponding to good and poor solvent conditions, see green and red data points in Figure 27b.

### 8.2. Remodeling of Vesicle Shape for Good Solvent Conditions

The green data in Figure 27b correspond to good solvent conditions with ζ=25/32=0.781 as defined by Equation (Equation 29). In this case, out-budded nanovesicles with closed membrane necks can be generated from spherical vesicle shapes using two different protocols, which both involve changes in the vesicle volume and in the exterior solute concentration but in reverse order. First, we can generate budded shapes with closed membrane necks by exposing a spherical nanovesicle to a sufficiently large solute concentration and then reducing the volume of this vesicle, see Figure 28a. Second, the same budded shapes can also be formed when we first deflate the vesicle in the absence of solute and subsequently increase the solute concentration in the exterior solution, see Figure 28b. For a deflated vesicle with rescaled volume ν=0.7, the membrane neck closes when we reach a threshold value of the solute concentration $\Phi_{S}$ that is larger than 0.09 and smaller than 0.1. The latter budding process is reversible as we demonstrated by decreasing and increasing the solute concentration several times.

### 8.3. Recurrent Shape Changes for Bad Solvent Conditions

We now consider poor solvent conditions with solubility ζ=25/40=0.625, corresponding to the red data in Figure 27b, and increase the solute concentration to mole fraction $\Phi_{S}=0.025$, which is close to the binodal line at $\Phi_{S}=\Phi_{S}^{*}=0.0275$. In this case, the vesicle is observed to form a budded shape for volume ν=0.75 as shown in Figure 29. However, for these parameter values of ζ and $\Phi_{S}$, the vesicle does not attain a stable shape but undergoes persistent shape changes with recurrent closure and opening of the membrane neck, see the time-lapse snapshots in Figure 29. The corresponding time series of the outer neck diameter is displayed in Figure 30. This recurrent behavior is reminiscent of the flickering fusion pores (kiss-and-run) that have been described for synaptic vesicles [114,115,116].

### 8.4. Division of Nanovesicles via Solute-Mediated Adhesion

We continue to consider poor solvent conditions with solubility ζ=25/40=0.625, corresponding to the red data in Figure 27b, and slightly increase the solute concentration from $\Phi_{S}=0.025$ to $\Phi_{S}=0.026$, which is even closer to the binodal line at $\Phi_{S}=\Phi_{S}^{*}=0.0275$. Now, the nanovesicle undergoes a complex budding process with subsequent membrane adhesion and neck fission provided the volume is below the threshold value, ν=0.85. For ν=0.9, the vesicle attains a stable prolate shape. In contrast, for smaller volumes with ν≤0.85, the vesicle divides into two daughter vesicles as shown in Figure 31 and Figure 32. The daughter vesicles adhere to one another by an intermediate adsorption layer of solute. This adhering couple of vesicles represents the stable vesicle morphology for the whole run time of the simulations, typically up to 100μs.

### 8.5. Fusion of Daughter Vesicles after Solute Removal

The adhesion of the two daughter vesicles as displayed in the last snapshots of Figure 31 is mediated by the adsorption layer of solute within the contact area. Thus, one would expect that the two daughter vesicles can be separated by removing all solute molecules, corresponding to the experimental procedure of solute ‘washout’. In the simulations, the complete removal can be obtained by transmuting all solute beads into water beads. As a result of this transmutation, the two daughter vesicles are observed to fuse with each other, as displayed by the time-lapse snapshots in Figure 33. The mechanism underlying this surprising behavior remains to be clarified. One possible mechanism is that the cleavage of the neck leads to a small defect in the contact area and that this defect acts as a nucleus for the fusion process. However, a systematic search for such a defect has not been successful so far.

## 9. Engulfment of Condensate Droplets by Planar Bilayers

In the previous section, the nanovesicles were exposed to a solute-water mixture that stayed in the one-phase region of the phase diagram, see Figure 27. In the present and in the next section, we will consider the two-phase region of this phase diagram in which the solute-water mixtures undergo liquid–liquid phase separation into two liquid phases, the solute-poor α and solute-rich β phase. More precisely, we will focus on the formation of small β droplets within the bulk α phase. The β droplets provide a simple model for condensate droplets.

### 9.1. Formation of Condensate Droplets In Vitro and In Vivo

Condensate droplets are formed in aqueous solutions of macromolecules that undergo phase separation into two liquid phases [108]. A well-studied example are aqueous solutions of the two polymers PEG and dextran, which have been used for a long time in biochemical analysis and biotechnology [117] and are intimately related to water-in-water emulsions [118].

The aqueous phase separation of PEG-dextran solutions provides an example for *segregative* phase separation, in which one phase is enriched in one macromolecular component such as PEG whereas the other phase is enriched in the other macromolecular component such as dextran. Segregative phase separation implies that the different species of macromolecules effectively repel each other. Another type of aqueous two-phase system is obtained by *associative* phase separation, for which one phase is enriched in the macromolecular components whereas the other phase represents a dilute aqueous solution of the macromolecules [119,120,121,122]. Associative phase separation implies that the different macromolecular species effectively attract each other. The latter type of phase separation is observed, for instance, in solutions of two, oppositely charged polyelectrolytes [121,122], a process also known as coacervation, which leads to coacervate droplets enriched in the polyelectrolytes.

Condensate droplets have also been observed in living cells where they provide separate liquid compartments which are not bounded by membranes. Examples for these condensates include germ P-bodies [123,124], nucleoli [125], and stress granules [126]. These biomolecular condensates are believed to form via liquid–liquid phase separation in the cytoplasm [123,127] and can be reconstituted in vitro [128,129,130,131]. They are enriched in certain types of proteins that have intrinsically disordered domains and interact via multivalent macromolecular interactions [127,130,131,132,133].

### 9.2. Interactions of Condensate Droplets with Biomembranes

Segregative phase separation of PEG-dextran solutions within giant unilamellar vesicles (GUVs) was initially reported by Christine Keating and coworkers [134]. Wetting transitions of condensate droplets at the GUV membranes were first observed in [135,136], complete engulfment of these droplets in [137,138]. GUVs in contact with coacervate droplets arising from associative phase separation have also been studied. These studies include the formation of coacervate droplets within GUVs [139,140], the exocytosis of such droplets from GUVs [141,142], and the endocytosis and uptake of coacervate droplets by GUVs [143]. A recent review of these processes is provided in [108].

Remodeling of cellular membranes by condensate-membrane interactions has been observed for P-bodies that adhere to the outer nuclear membrane [123], for lipid vesicles within a synapsin-rich liquid phase [144], for TIS granules interacting with the endoplasmic reticulum [145], for condensates at the plasma membrane [146,147,148], and for condensates that are enriched in the RNA-binding protein Whi3 and adhere to the endoplasmatic reticulum [149].

### 9.3. Geometry of Partial and Complete Engulfment

The adhesion of a small β droplet to a planar bilayer is displayed in Figure 34. The β droplet coexists with the bulk phase α. The system also involves a third liquid phase γ, which plays the role of an inert spectator phase. The adhesion of the β droplet to the bilayer leads to a bilayer-droplet morphology that involves three surface segments: the αβ interface between the β droplet and the α phase as well as two bilayer segments, the βγ segment in contact with the β droplet and the αγ segment exposed to the α phase. Furthermore, the αβ interface between the two liquid phases forms a contact line with the bilayer which provides the boundary of its βγ contact segment.

The partial engulfment of the β droplet in Figure 34 was obtained for solute mole fraction $\Phi_{S}=0.0126$ and solubility ζ=1/2 in the phase diagram of Figure 27. Furthermore, the planar bilayer in Figure 34 is symmetric in the sense that each leaflet contains the same number of lipid molecules. Thus, this planar bilayer is characterized by equal leaflet tensions in the absence of the adhering droplet.

The planar bilayer displayed in Figure 34 is subject to a significant bilayer tension arising from the periodic boundary conditions acting on the bilayer, which prevent the bilayer from increasing the area of its βγ contact segment with the β droplet. Therefore, the droplet is only partially engulfed by the bilayer which implies that the interfacial free energy of the αβ interface makes a significant contribution to the free energy of the bilayer-droplet system. This interfacial free energy contribution can be diminished by reducing the lateral size $L_{\|}$ of the simulation box, which allows the bilayer to increase the area of its βγ contact segment and to decrease the area of the αβ interface.

### 9.4. Complete Engulfment of Droplets with Tight-Lipped Membrane Necks

The lateral size $L_{\|}$ of the simulation box can be reduced in a continuous manner as displayed in Figure 35. In this example, the simulation starts with a lateral box size of $L_{\|}=130\;d$ which is then continuously reduced until it reaches its final value $L_{\|}=120\;d$ after 4μs. This reduction of the lateral box size allows the bilayer to engulf the droplet completely as one can conclude from the side views in Figure 35b. However, the bottom views in Figure 35a reveal that the complete engulfment proceeds in a non-axisymmetric manner and leads to an unusual tight-lipped membrane neck.

### 9.5. Negative Line Tension of Contact Line

The unusual tight-lipped shape of the membrane neck in Figure 35 arises from the negative line tension λ of the contact line. The numerical value of this line tension can be computed using the force balance between the three surface segments that meet along the contact line. This computation can be performed for a circular contact line and an axisymmetric membrane neck as shown in the first column of Figure 35.

The three surface segments that meet along the contact line experience three different surface tensions, the interfacial tension $\Sigma_{\alpha \beta }$, the bilayer tension $\Sigma_{\beta \gamma }^{m}$ of the βγ bilayer segment in contact with the β droplet, and the bilayer tension $\Sigma_{\alpha \gamma }^{m}$ of the αβ bilayer segment exposed to the α phase. All three surface tensions can be obtained from the stress profiles across the three surfaces [26]. In addition to the line tension and the three surface tensions, the force balance along the contact line also involves three geometric quantities, the intrinsic contact angle $\theta_{\alpha }^{*}$, the contact line radius $R_{\mathrm{co}}$, and the tilt angle $\psi_{\mathrm{co}}$ of the normal vector at the contact line, all of which can be deduced from the simulations. The tangential (or parallel) force balance as given by [136]
(30)$$\Sigma_{\beta \gamma }^{m}-\Sigma_{\alpha \gamma }^{m}=\Sigma_{\alpha \beta }\cos\theta_{\alpha }^{*}+\frac{\lambda }{R_{\mathrm{co}}}\cos\psi_{\mathrm{co}}\;,$$
then involves a single unknown quantity, the line tension λ. Using the measured values of the surface tensions and the geometric quantities, one finds that the line tension λ is negative for a wide range of parameters [26].

## 10. Engulfment of Rigid Nanoparticles by Planar Bilayers

### 10.1. Complete Engulfment of Nanoparticles with Tight-Lipped Membrane Necks

The unusual tight-lipped shape of the membrane neck, as described in the previous section for the complete engulfment of condensate droplets by planar bilayers, has also been observed for the complete engulfment of rigid nanoparticles. The nanoparticle was constructed using particle beads arranged on an fcc lattice with the number density $\rho =3/(d^{3})$ [150]. The closely packed particle beads left no space for water beads to penetrate into the nanoparticle.

Partial engulfment of such a nanoparticle by a planar bilayer is shown in Figure 36a, complete engulfment in Figure 36b. Surprisingly, the contact line between the bilayer and the nanoparticle can also become strongly non-circular during the particle engulfment, as follows from the bottom views in Figure 37.

### 10.2. Other Simulation Studies of Nanoparticle Engulfment

A variety of simulation methods has been previously used to study the engulfment of solid or rigid nanoparticles by lipid bilayers and nanovesicles. Engulfment of such particles has been investigated by Brownian dynamics [151], DPD simulations [150,152,153] and Monte Carlo simulations [154,155]. DPD simulations have also been applied to the translocation of relatively small nanoparticles through lipid bilayers [156]. However, none of these previous studies on the engulfment of rigid nanoparticles reported the tight-lipped membrane neck displayed in Figure 36 and Figure 37.

## 11. Engulfment and Endocytosis of Condensate Droplets by Nanovesicles

We now address the engulfment of a condensate droplet by the bilayer of a nanovesicle. After the droplet has come into contact with the vesicle, the engulfment process is induced by a reduction in the vesicle volume. This process can proceed via two different pathways that involve axisymmetric and non-axisymmetric membrane shapes [31]. The non-axisymmetric pathway for complete engulfment leads to a non-circular contact line, which is caused by a *negative* line tension, and to a tight-lipped shape of the membrane neck, which prevents the fission of this neck. Thus, the non-axisymmetric pathway is similar to the engulfment of a condensate droplet by a symmetric planar bilayer, as shown in Figure 35. The axisymmetric pathway, on the other hand, leads to a circular contact line, which is governed by a *positive* line tension, and to an axisymmetric membrane neck that undergoes fission, thereby dividing the nanovesicle up into two daughter vesicles. Furthermore, it turns out that the sign of the line tension is determined by the stress asymmetry between the two leaflet tensions: A sufficiently small stress asymmetry $\Delta \Sigma_{\mathrm{ve}}=\Sigma_{\mathrm{ol}}-\Sigma_{\mathrm{il}}$ as defined in Equation (Equation 25) leads to a negative line tension and, thus, to a tight-lipped membrane neck whereas a sufficiently large positive stress asymmetry leads to positive line tension and to the cleavage of the axisymmetric membrane neck, see Figures 42 and 43 below.

### 11.1. Partial Engulfment of Condensate Droplets

Complete engulfment of a droplet by volume reduction of the vesicle is only possible if the droplet is sufficiently small or the membrane area released by the volume reduction is sufficiently large. When we reduce the volume of a spherical vesicle by $\Delta V_{\mathrm{ve}}$, the droplet volume $V_{\mathrm{dr}}$ must satisfy the inequality $V_{\mathrm{dr}}\leq \Delta V$. This intuitive argument can be corroborated by applying the isoperimetric inequality $A^{3}\geq 36\pi V^{2}$ which is valid for any closed surface with surface area *A* and volume *V* [31,108].

If the droplet is too large or the released membrane area too small, the droplet can only partially be engulfed. An example for partial engulfment is displayed in Figure 38, which was obtained for solute mole fraction $\Phi_{S}=0.004$ and solubility ζ=25/70=0.36 as defined by Equations (Equation 28) and (Equation 29). For these parameter values, the nanovesicle is located within the two-phase coexistence region of the phase diagram in Figure 27. Initially, both the droplet and the nanovesicle are fully immersed in the liquid phase α as shown in Figure 38a. When the droplet comes into contact with the vesicle membrane, a small contact area is formed as in Figure 38b. After this onset of adhesion, the vesicle membrane starts to engulf the membrane. This process continues by pulling out membrane area from the thermally excited undulations. Eventually, a new stable morphology, corresponding to partial engulfment, is reached as shown in Figure 38c.

### 11.2. Different Pathways for Engulfment and Endocytosis

Now, let us consider a sufficiently large volume reduction of the nanovesicle so that the condensate droplet can become completely engulfed. When the vesicle bilayer contains $N_{ol}=5400$ lipids in its outer and $N_{il}=4700$ lipids in the inner leaflet, the complete engulfment process proceeds in an axisymmetric manner as shown in Figure 39a. In this case, the vesicle forms a closed membrane neck at time t=0.3μs which undergoes fission at t=2μs, thereby generating a small intraluminal vesicle enclosing the droplet. The same axisymmetric pathway of complete engulfment and endocytosis is obtained when the vesicle bilayer contains $N_{ol}=5500$ lipids in its outer and $N_{il}=4600$ lipids in the inner leaflet as displayed in Figure 39b. Now, the membrane neck closes at t=4μs and undergoes fission at t=9μs, again generating a small intraluminal vesicle.

Complete engulfment of a condensate droplet by a vesicle bilayer can also proceed via non-axisymmetric shapes. The latter pathway is observed when the vesicle bilayer contains $N_{ol}=5700$ lipids in its outer and $N_{il}=4400$ lipids in its inner leaflet as in Figure 40a or $N_{ol}=5963$ lipids in its outer and $N_{il}=4137$ lipids in its inner leaflet as in Figure 40b. In both cases, the complete engulfment process leads to a tight-lipped shape of the membrane neck, as indicated by the white dashed rectangles around the contact lines in Figure 40, which prevents the fission of this neck and the division of the vesicle.

In order to elucidate the mechanism underlying the different pathways for complete engulfment as displayed in Figure 39 and Figure 40, we will now look at the contact line tension λ and on the stress asymmetry between the two leaflets of the tensionless bilayers.

### 11.3. Contact Line Tensions and Stress Asymmetry of Vesicle Bilayers

The line tension of the contact line between the condensate droplet and the vesicle bilayer can again be computed using the force balance along the contact line as described by Equation (Equation 30). The result of this computation is displayed in Figure 41 for three different droplet diameters $D_{\mathrm{dr}}$.

Inspection of Figure 41 shows that the line tension λ is positive for sufficiently large values of the outer lipid number $N_{ol}$ but becomes negative for sufficiently small values of this lipid number. The line tension vanishes at a certain lipid number $N_{ol}=N_{ol}^{\left[0\right]}$, which depends weakly on the diameter $D_{\mathrm{dr}}$ of the β droplet and thus on the curvature of the αβ interface. Interpolation of the data in Figure 41 leads to $N_{ol}^{\left[0\right]}=5582$ for the smallest droplets with diameter $D_{\mathrm{dr}}=14\;d$, to $N_{ol}^{\left[0\right]}=5572$ for $D_{\mathrm{dr}}=18.3\;d$, and to $N_{ol}^{\left[0\right]}=5538$ for the largest droplets with diameter $D_{\mathrm{dr}}=24.5\;d$.

The data displayed in Figure 41 are obtained for OLT states of the vesicle bilayers with vanishing bilayer tension, $\Sigma_{ol}+\Sigma_{il}=0$, in the absence of the adhering droplets. The dependence of the two leaflet tensions $\Sigma_{ol}$ and $\Sigma_{il}$ on the outer leaflet number $N_{ol}$ is displayed in Figure 42. We now combine the $N_{ol}$-dependence of the leaflet tensions in Figure 42 with the $N_{ol}$-dependence of the line tension in Figure 41 to conclude that the line tension λ is positive for large stress asymmetries ΔΣ but negative for small stress asymmetries. This conclusion is consistent with the results on symmetric planar bilayers because the latter bilayers are characterized by vanishing stress asymmetry and negative line tension.

## 12. Adhesion and Fusion of Nanovesicles Controlled by Leaflet Tensions

Finally, in this last section, we emphasize that the leaflet tensions also play an important role for the adhesion and fusion of nanovesicles. As a simple example, we consider two identical vesicles, both of which are characterized by tensionless bilayers and by stretched outer leaflets when they come into contact. Depending on the magnitude of the positive outer leaflet tension $\Sigma_{ol}>0$, the vesicle adhere to each other as in Figure 43 or undergo fusion as in Figure 44.

Figure 43 displays the time evolution of the two identical vesicles when their bilayers contain $N_{il}=4400$ lipids in their inner and $N_{ol}=5700$ lipids in their outer leaflets. In this case, the inner leaflet of each vesicle is compressed by the negative leaflet tension $\Sigma_{il}=-0.82\;k_{B}T/d^{2}$ whereas the outer leaflet of each vesicle is stretched by the positive leaflet tension $\Sigma_{ol}=+0.87\;k_{B}T/d^{2}$. Thus, the vesicle bilayer is subject to the initial stress asymmetry $\Delta \Sigma =\Sigma_{ol}-\Sigma_{il}=1.69\;k_{B}T/d^{2}$. After slightly deflating both vesicles from volume ν=0.8 to volume ν=0.7, the two vesicles start to adhere to each other and to transform into oblate shapes, thereby increasing their contact area, see last snapshot in Figure 43.

When the initial stress asymmetry is slightly increased from $\Delta \Sigma =\;1.69\;k_{B}T/d^{2}$ to $\Delta \Sigma =2.04\;k_{B}T/d^{2}$, a topological remodeling process is observed as displayed Figure 44. Now, the two vesicles undergo a fast fusion process that is completed within 0.3μs, even without an intermediate deflation step.

## 13. Summary and Outlook

In this review, we described recent insights into the fluid-elastic behavior of lipid bilayers and nanovesicles as obtained from molecular dynamics simulations. We emphasized that this behavior is primarily controlled by the two leaflet tensions of the bilayer membranes. The position of the undulating leaflet-leaflet interface can be visualized by using different colors for the lipids in the two leaflets as in Figure 3 and Figure 5. In order to compute the two leaflet tensions, we considered the average position of the leaflet-leaflet interface, which defines the midplane for planar bilayers and the midsurface for vesicle bilayers. The behavior of planar bilayers was discussed in more detail in Section 3, Section 4 and Section 5. To determine the bilayer’s midplane, we used (i) the CHAIN protocol, which is based on the density profile of the hydrophobic lipid chains (Figure 7a–c), and (ii) the VORON protocol, based on Voronoi tessellation, by which we first compute the volume of the lipid and water molecules and subsequently add these molecular volumes up to obtain the volumes of the different subcompartments of the system, see Section 3.4.3 and Figure 8.

A global view about the elastic behavior of planar bilayers is obtained from its two-dimensional leaflet tension space in Figure 9. The origin of this space is provided by the reference state with tensionless leaflets. This reference state can be deformed into equal leaflet tension (ELT) states with $\Sigma_{ul}=\Sigma_{ll}$ and opposite leaflet tension (OLT) states with $\Sigma_{ul}=-\Sigma_{ll}$. The ELT states are located along the main diagonal of the leaflet tension space, the OLT states along the perpendicular diagonal, which is orthogonal to the main diagonal. The stress asymmetry $\Delta \Sigma =\Sigma_{ul}-\Sigma_{ll}$ is well-defined for all bilayer states, that is, for all points of the leaflet tension space whereas the spontaneous curvature is only defined for the OLT states.

Section 4 described the behavior of mixed lipid bilayers with two lipid components, corresponding to large-head and small-head lipids (Figure 10 and Figure 11), and with three components, corresponding to two phospholipid components and fast flip-flopping cholesterol (Figure 12 and Figure 13). The fluctuation spectrum of bending undulations for two-component bilayers as displayed in Figure 11 fully confirms Equation (Equation 16), which was originally derived in [36] for one-component bilayers. For two-component bilayers consisting of two phosopholipids and no cholesterol, the reference states with vanishing leaflet tensions can possess a nonzero spontaneous curvature, arising from the compositional asymmetry of the reference states (Section 4.2.2). Large stress asymmetries between the upper and lower leaflets of the planar bilayers lead to the onset of flip-flops between the two leaflets (Figure 16) and to structural instabilities of these bilayers (Figure 17) as reviewed in Section 5.

The fluid-elastic behavior of nanovesicles was discussed in Section 6, Section 7 and Section 8. In Section 6, we first described the protocols to determine the midsurface of the nanovesicles, the CHAIN protocol in Section 6.3.1 and the VORON protocol in Section 6.3.2. In response to changes in vesicle volume, spherical nanovesicles undergo a variety of shape transformations (Figure 19, Figure 20, Figure 21 and Figure 22) which are controlled by the leaflet tensions of the vesicle bilayers (Figure 18). For negative stress asymmetry, the nanovesicle can form in-buds with closed membrane necks that undergo fission (Figure 22).

A global view about the elastic behavior of nanovesicles is obtained via the two-dimensional leaflet tension space in Figure 23. The origin of this space is provided by the reference state with tensionless leaflets. By reshuffling lipids between the inner and outer leaflets, the reference state can be transformed into bilayer states with opposite leaflet tensions, $\Sigma_{ol}=-\Sigma_{il}$. Another physically relevant transformation is provided by vesicle inflation or deflation (VID), which generates a curved pathway in the leaflet tension space. Large stress asymmetries between the outer and inner leaflets of the vesicle bilayers lead to the onset of flip-flops between the two leaflets (Figure 25) and to structural instabilities of the vesicle bilayers (Figure 26).

Nanovesicles exposed to small solutes that adsorb onto the vesicle bilayers were considered in Section 8. The phase diagram of the binary water-solute mixture, which depends on the solute mole fraction $\Phi_{S}$ and on the solubility ζ of solute in water, is displayed in Figure 27. For poor solvent conditions with ζ=0.625, the binodal line is located at $\Phi_{S}=\Phi_{S}^{*}=0.0275$. When the solution is still in the one-phase region but close to the binodal line, the nanovesicles exhibit unusual shape transformations. For $\Phi_{S}=0.025<\Phi_{S}^{*}$, the dumbbell-shaped vesicles exhibit recurrent shape changes between dumbbells with closed and with open necks (Figure 29 and Figure 30). For $\Phi_{S}=0.026<\Phi_{S}^{*}$, the recurrent closing and reopening of the neck is truncated by neck fission which leads to two daughter vesicles that are bound to each other by a solute adsorption layer (Figure 31 and Figure 32). Removal of the solute leads to fusion and reunification of the two daughter vesicles (Figure 33). The mechanism underlying this fusion process remains to be clarified.

In Section 9, Section 10 and Section 11, we discussed the engulfment and endocytosis of condensate droplets and rigid nanoparticles. Complete engulfment of condensate droplets by a planar bilayer proceeds in a non-axisymmetric manner and leads to a closed membrane neck with an unusual tight-lipped shape (Figure 35). This shape of the membrane neck in Figure 35 arises from the negative line tension λ of the contact line. The negative sign of the line tension is obtained from the force balance along the contact line as given by Equation (Equation 30), which involves the surface tensions of the three surface segments that meet at the contact line. A tight-lipped neck shape has also been observed for the engulfment of rigid nanoparticles by planar bilayers (Figure 36 and Figure 37). On the other hand, the complete engulfment of condensate droplets by vesicle bilayers can proceed along two different pathways. One pathway involves axisymmetric shapes and leads to closed membrane necks that undergo fission (Figure 39). In this case, the line tension of the contact line is *positive* as follows from a combination of Figure 41 and Figure 42. The other pathway proceeds via non-axisymmetric shapes and necks. When these necks close, they attain a tight-lipped shape, which prevents the fission of these necks and the division of the vesicle (Figure 40). In the latter case, the line tension of the contact line is *negative* as follows again from a combination of Figure 41 and Figure 42.

All engulfment processes described in Section 9, Section 10 and Section 11 are consistent with the view that small and large stress asymmetries between the two bilayer leaflets lead to negative and positive line tensions, respectively. In order to corroborate this view, it will be rather useful to study the engulfment of condensate droplets by planar bilayers with large stress asymmetries, a system that has not been considered so far. In Section 12, we discussed the influence of the leaflet tensions on the adhesion and fusion of two nanovesicles, providing strong evidence that these two processes are also controlled by the stress asymmetry between the two bilayer leaflets. In fact, when the stress asymmetry is below a certain threshold value, the two vesicles adhere as in Figure 43 whereas they fuse as in Figure 44 when the stress asymmetry exceeds this threshold.

Because the flip-flop rate depends on the stress asymmetry between the two leaflet tensions, see Figure 16 for planar bilayers and Figure 25 for vesicle bilayers, one should be able to obtain additional insight into these tensions from experiments that measure how the flip-flop rates vary when we expose the bilayers to different external conditions. One approach is provided by osmotic deflation and inflation of nanovesicles, see the VID data in Figure 23. Furthermore, the intrinsic contact angle of condensate droplets that adhere to the outer leaflet of a nanovesicle, see Figure 38 and Equation (Equation 30), should also be quite sensitive to changes of the outer leaflet tension and, thus, of the stress asymmetry. The intrinsic contact angle can be measured by super-resolution microscopy such as STED [157]. Combining such super-resolution methods with osmotic inflation or deflation of the vesicles and with the VID data in Figure 23, one might be able to determine the dependence of the intrinsic contact angle on the outer leaflet tension.

In this study, we focused on one-component bilayers for the sake of computational simplicity but our results can be directly generalized to multi-component bilayers as well, provided the bilayer leaflets are fluid and have a laterally uniform composition. Indeed, for leaflets with a uniform molecular composition, the leaflet tensions describe the local stress within these leaflets, irrespective of the number of molecular components. Thus, for a multi-component bilayer, the different components will have different areas per lipid but the average molecular area in each leaflet will again determine the corresponding leaflet tension and vice versa.

Multi-component bilayers can also be studied experimentally to measure the cumulative distribution function for the onset of flip-flops (Figure 16 and Figure 25). To prepare a population of vesicles with a significant stress asymmetry, one might use the enzymatic conversion of phospholipid headgroups in the outer leaflet as demonstrated by the conversion of the PS to the PE headgroup [158] and of the PC headgroup to the PS and PE headgroups [159]. After such a vesicle population has been prepared, one would look for flip-flops and measure the corresponding flip-flop times, which are likely to exceed the flip-flop times observed in our simulations by several orders of magnitude. Studying a dilute dispersion of liposomes and adding local lipid probes to their membranes, it may also be possible to distinguish flip-flops in the same vesicle bilayer from those between different bilayers.

As discussed in Section 9.1 and Section 9.2, condensate droplets are also formed on the micrometer scale, both in vitro and in vivo. Furthermore, the adhesion of these droplets to synthetic or cellular membranes leads to a variety of remodeling processes of these membranes [108]. In the living cell, the condensates are enriched in certain types of proteins with intrinsically disordered regions. The binding of such proteins to membranes has been recently studied by all-atom molecular dynamics [160,161]. It would be rather interesting to map the all-atom force fields used in the latter studies to DPD force parameters in order to study the engulfment and endocytosis of the protein-enriched condensate droplets as well as the dependence of these processes on the leaflet tensions.

## Figures and Tables

**Figure 1 biomolecules-13-00926-f001:**
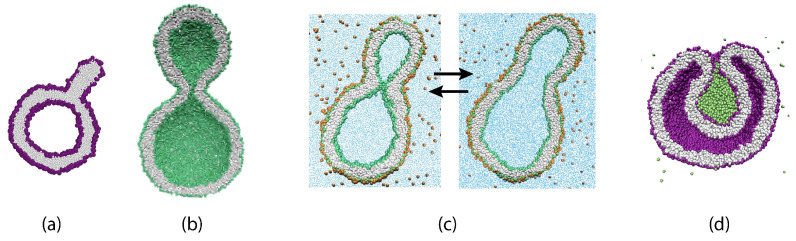
Simulation snapshots of nanovesicles that illustrate their remodeling for different leaflet tensions: (**a**) Cross-sectional view of a nanovesicle that undergoes a structural instability, generating a micellar protrusion [30]; (**b**) Half cut view of a vesicle that forms a dumbbell shape with a closed membrane neck [28]; (**c**) Cross-sectional views of recurrent shape changes between dumbbells with open and closed necks [29]; (**d**) Half cut view of a condensate droplet (green) that is completely engulfed by the in-bud of a nanovesicle and will divide into two daughter vesicles [31]. All vesicle bilayers in (**b**–**d**) have the same surface area, which is about 3.6 times larger than the area of the vesicle bilayer in (**a**).

**Figure 2 biomolecules-13-00926-f002:**
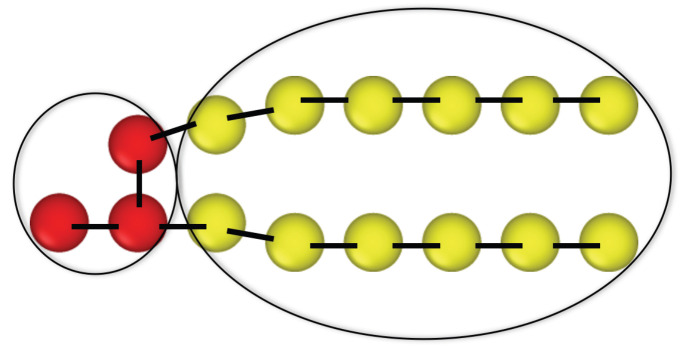
Coarse-grained molecular architecture of a phospholipid with three hydrophilic headgroup beads (red) and two hydrophobic chains, each of which contains six chain beads (yellow). All beads have the same diameter, *d*, which also provides the range of the interactions between the beads. The tensionless bilayer has a thickness of about 5d which implies that the diameter *d* is about 0.8 nm in physical units [25].

**Figure 3 biomolecules-13-00926-f003:**
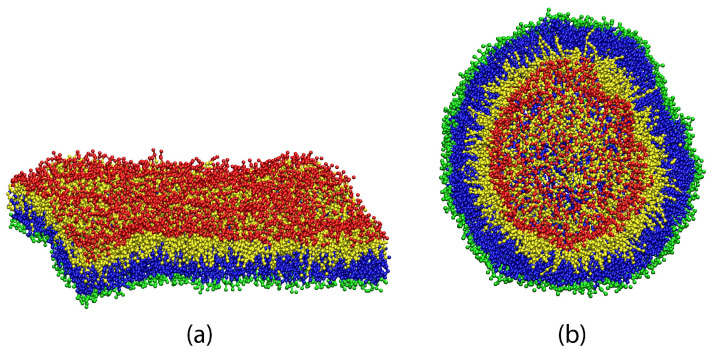
(**a**) Oblique view onto a planar lipid bilayer consisting of 841 lipid molecules in each leaflet; and (**b**) Half cut view of a nanovesicle with $N_{ol}=1685$ lipids in its outer and $N_{il}=840$ lipids in its inner leaflet. In order to highlight the interface between the two leaflets, the lipids in the upper and inner leaflet are drawn with red headgroups and yellow chains whereas the lipids in the lower and outer leaflet have green headgroups and blue chains, even though all lipids are identical. For the bilayers displayed in (**a**,**b**), both leaflets are tensionless which implies that each lipid attains its optimal area and its optimal volume. The lipid bilayers are immersed in plenty of water but the water molecules are not displayed for visual clarity [32].

**Figure 4 biomolecules-13-00926-f004:**
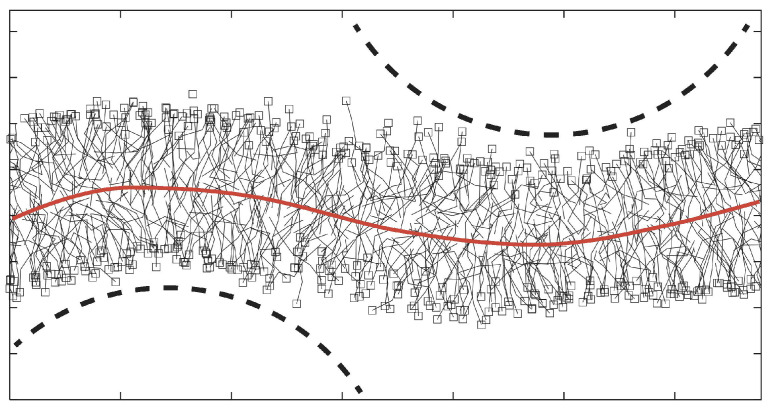
Emergence of membrane curvature in molecular dynamics simulations of a symmetric and tensionless bilayer. The lipid bilayer has a thickness of about 4 nm, the smallest curvature radius of its midsurface (red) is about 6 nm. For comparison, two circles (broken lines) with a radius of 6 nm are also displayed. The red line indicates the position of the fluctuating leaflet-leaflet interface [36].

**Figure 5 biomolecules-13-00926-f005:**
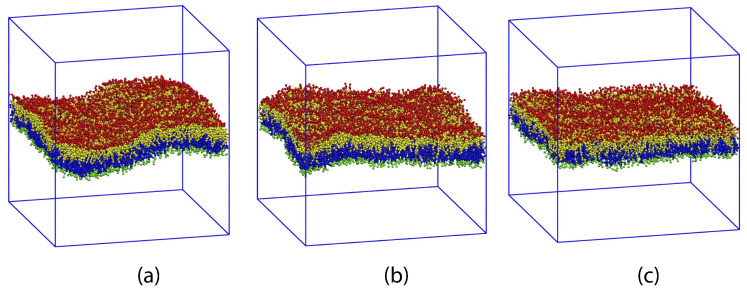
Three symmetric planar bilayers which contain the same number of lipids, $N_{ll}=N_{ul}=841$ in their lower and upper leaflets. The three bilayers differ in the base area of simulation box which provides another control parameter for the leaflet tensions, $\Sigma_{ll}$ and $\Sigma_{ul}$, acting in the two leaflets. (**a**) Compressed bilayer with negative leaflet tensions, $\Sigma_{ll}=\Sigma_{ul}<0$, and (projected) area per lipid, $a=1.18\;d^{2}$; (**b**) Reference state of planar bilayer with tensionless leaflets, $\Sigma_{ll}=\Sigma_{ul}=0$, and optimal area per lipid, $a=a^{0}=1.22\;d^{2}$; and (**c**) Stretched bilayer with positive leaflet tensions $\Sigma_{ll}=\Sigma_{ul}>0$ and increased area per lipid $a=1.29\;d^{2}$ [32].

**Figure 6 biomolecules-13-00926-f006:**
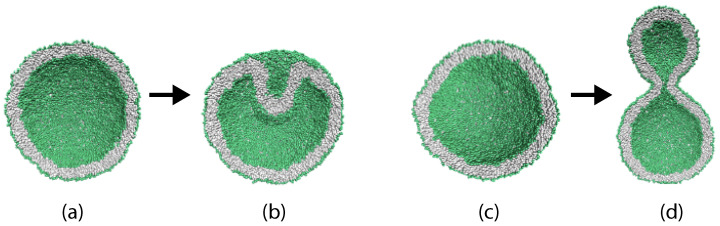
Shape transformations of spherical nanovesicles induced by volume reduction: (**a**,**b**) Transformation of a spherical vesicle with $N_{il}=4400$ lipids in the inner and $N_{ol}=5700$ lipids in the outer leaflet. For the slightly deflated vesicle with vanishing bilayer tension, the inner leaflet is compressed and the outer leaflet is stretched which implies that the bilayer prefers to bulge towards the inner leaflet; and (**c**,**d**) Transformation of a spherical vesicle with $N_{il}=3800$ lipids in the inner and $N_{ol}=6300$ lipids in the outer leaflet. For the slightly deflated vesicle with vanishing bilayer tension, the inner leaflet is stretched and the outer leaflet is compressed which implies that the bilayer prefers to bulge towards the outer leaflet [28].

**Figure 7 biomolecules-13-00926-f007:**
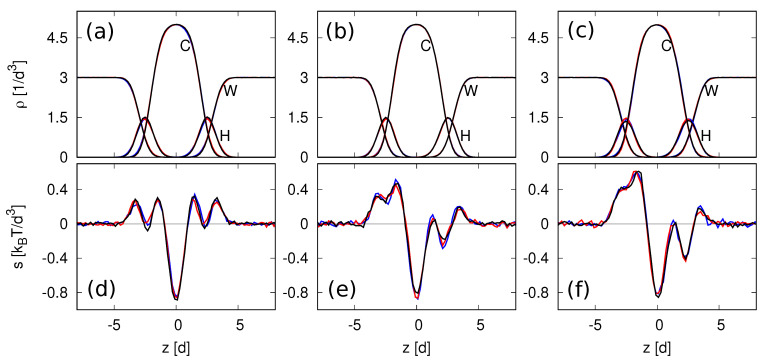
Density and stress profiles across lipid bilayers as functions of the coordinate *z* perpendicular to the bilayers. All bilayers are tensionless, i.e., the bilayer tension Σ as given by the integral over *z*, see Equation (Equation 2), is close to zero. The bilayers contain $N_{ll}$ lipids in their lower leaflet and $N_{ul}$ lipids in their upper leaflet: (**a**–**c**) Density profiles of the water (W), headgroup (H), and hydrophobic chain (C) beads; and (**d**–**f**) Stress profiles across the same bilayers as in the upper row. The bilayer in (**a**,**d**) is symmetric with $N_{ll}=N_{ul}=841$ lipids in each leaflet. The bilayer in (**b**,**e**) is asymmetric with $N_{ll}=815$ and $N_{ul}=862$. The bilayer in (**c**,**f**) has a larger transbilayer asymmetry with $N_{ll}=801$ and $N_{ul}=875$. The blue, red, and black lines correspond to different heights of the simulation box, as given by 32d, 40d, and 48d. Comparison of these three lines implies that finite-size effects arising from the box height can be ignored [24].

**Figure 8 biomolecules-13-00926-f008:**
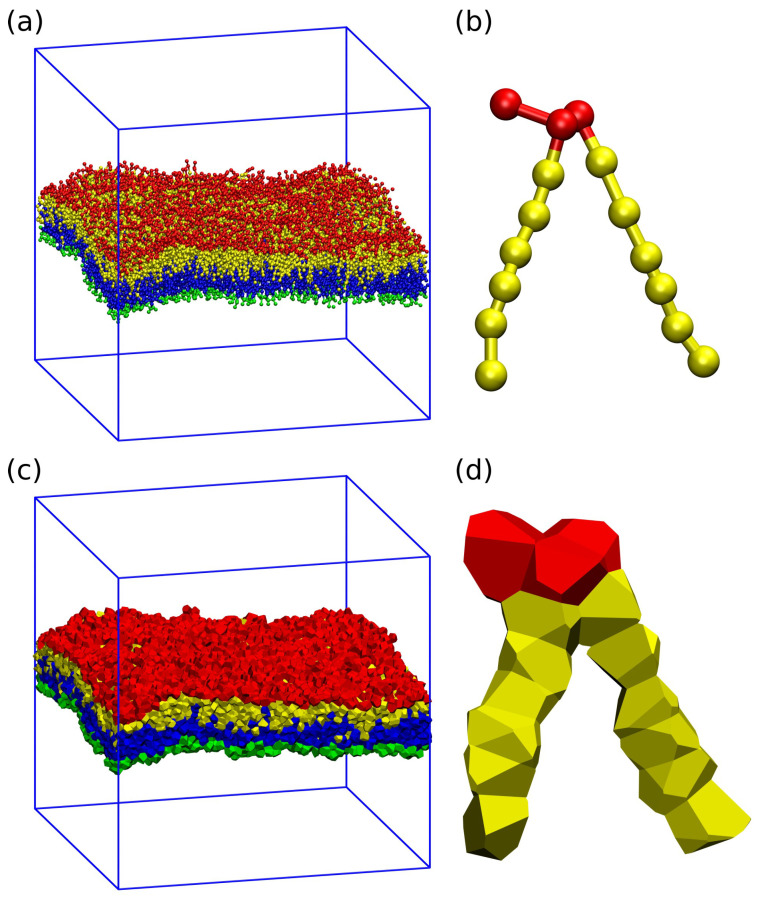
Three-dimensional Voronoi tessellation of a planar bilayer and a single lipid molecule: (**a**) Simulation snapshot of planar bilayer with tensionless leaflets and $N_{ll}=N_{ul}=841$. The lipids in the lower leaflet have green headgroups and blue hydrocarbon chains; those in the upper leaflet have red headgroups and yellow hydrocarbon chains; (**b**) Typical conformation of a single lipid molecule within the upper leaflet in ball-and-stick representation; (**c**) Voronoi cells assigned to each bead of the bilayer in panel a; and (**d**) Voronoi cells assigned to each bead of the lipid molecule in panel (**b**). Voronoi cells are also assigned to each water bead but these cells are not shown here for visual clarity [32].

**Figure 9 biomolecules-13-00926-f009:**
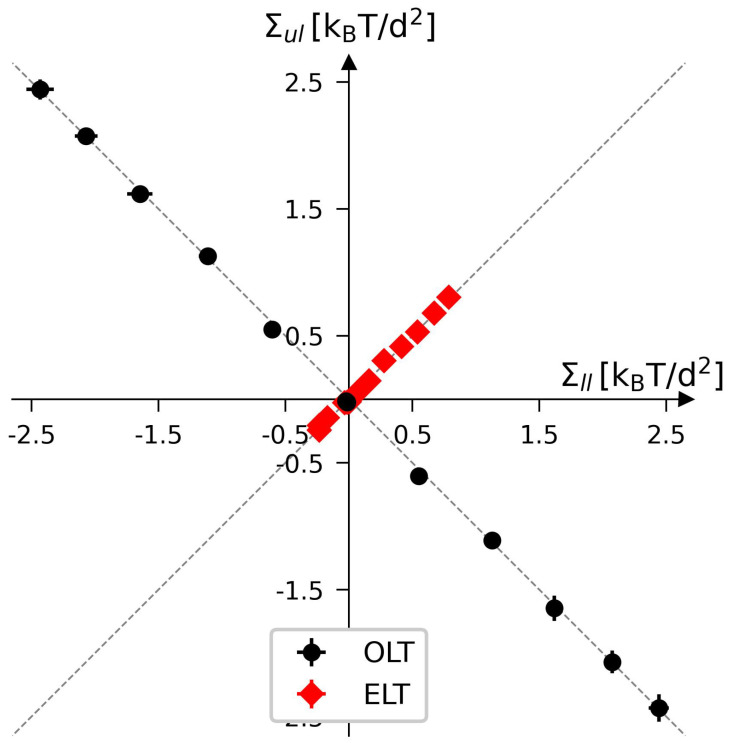
Leaflet tension space for planar bilayers that contain a total number of $N_{ll}+N_{ul}=1682$ lipids. The two coordinates are the leaflet tensions $\Sigma_{ll}$ and $\Sigma_{ul}$ in the lower and upper leaflets. Negative and positive leaflet tensions describe compressed and stretched leaflets. The reference state with tensionless leaflets, corresponding to $\Sigma_{ll}=\Sigma_{ul}=0$, is obtained for the symmetric bilayer with $N_{ll}=N_{ul}=841$ lipids. The red data points describe elastic deformations with equal leaflet tensions (ELT), $\Sigma_{ll}=\Sigma_{ul}$. The black data represent bilayers with opposite leaflet tensions (OLT), $\Sigma_{ll}=-\Sigma_{ul}$. All OLT states can be obtained from the reference state by reshuffling lipids from one leaflet to the other and adjusting the base area of the simulation box to obtain tensionless bilayers. The midplanes of the asymmetric OLT states were calculated using the CHAIN protocol [32].

**Figure 10 biomolecules-13-00926-f010:**
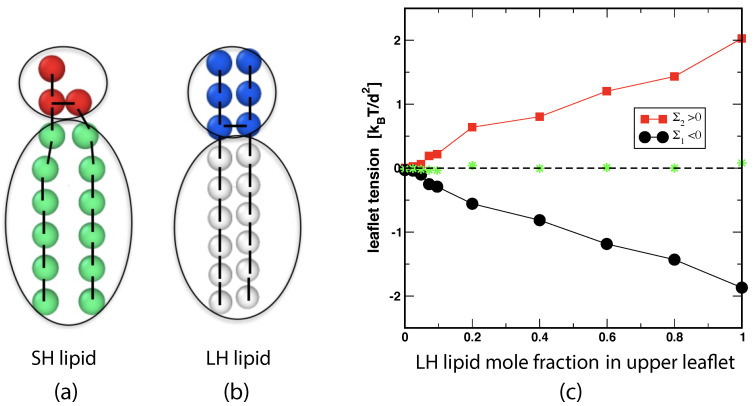
Tensionless bilayers assembled from two lipid components, small-head (SH) lipids and large-head (LH) lipids: (**a**,**b**) Molecular architecture of small-head (SH) and large-head (LH) lipids. In both cases, the two hydrocarbon chains are modeled by 2×6 chain (C) beads (green or gray). The headgroup of the SH lipid consists of three H beads (red) whereas the headgroup of the LH lipid has six H beads (blue); and (**c**) Upper leaflet tension $\Sigma_{1}\equiv \Sigma_{ul}<0$ (black circles) and lower leaflet tension $\Sigma_{2}\equiv \Sigma_{ll}=-\Sigma_{ul}>0$ (red squares) as a function of the mole fraction $\phi \equiv \phi_{ul}^{\mathrm{LH}}$ of the LH lipids in the upper leaflet for a lower leaflet that contains only SH lipids. The two-component bilayer was assembled from $N_{ul}^{\mathrm{SH}}+N_{ul}^{\mathrm{LH}}=N_{ll}^{\mathrm{SH}}+N_{ll}^{\mathrm{LH}}=841$ lipids. The bilayer tension $\Sigma =\Sigma_{ul}+\Sigma_{il}$ (green stars) is close to zero, demonstrating that the bilayers attain OLT states. The midplanes of the bilayers were determined by the CHAIN protocol.

**Figure 11 biomolecules-13-00926-f011:**
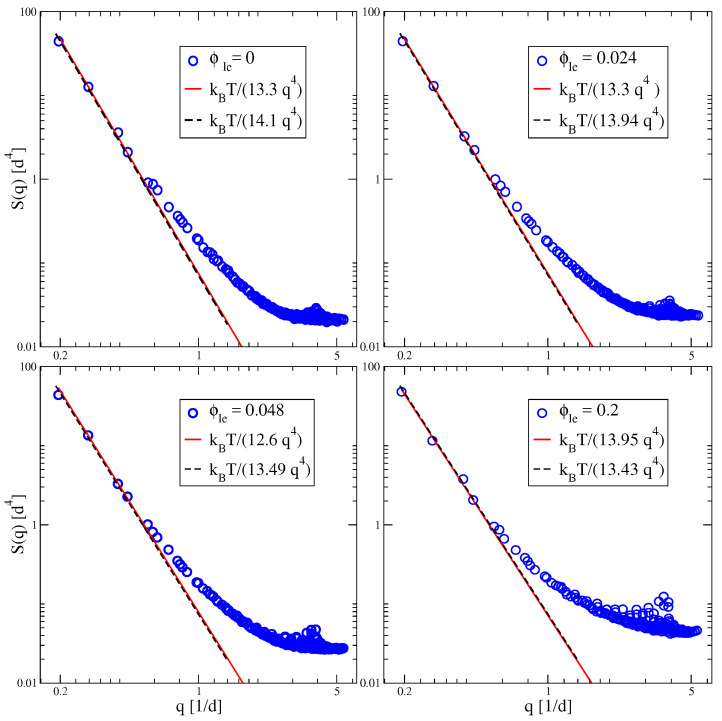
Fluctuation spectra of bending undulations for planar and symmetric bilayers with two lipid components as functions of wavenumber *q*. The two leaflets contain the same lipid number, $N_{ul}=N_{ll}=841$, and the same mole fraction $\phi_{\mathrm{le}}$ of the large-head (LH) lipids. The different panels correspond to different mole fractions. The box size is adjusted in such a way that each bilayer is tensionless and has (almost) zero bilayer tension $\Sigma =\Sigma_{ul}+\Sigma_{ll}=0$. It then follows from the symmetry between the two leaflets that both leaflet tensions vanish, $\Sigma_{ul}=\Sigma_{ll}=0$. For all mole fractions, the low-*q* part of the spectrum behaves as $S\left(q\right)\approx k_{B}T/\left(\kappa q^{4}\right)$ as in Equation (Equation 17) with composition-dependent κ-values. The bending rigidities κ, as obtained from the spectra, the area compressibility modulus $K_{A}$, and the membrane thickness $\ell_{\mathrm{me}}$, as determined by independent analysis, are found to satisfy the relationship in Equation (Equation 16) with the prefactor 1/48 for all lipid compositions [25].

**Figure 12 biomolecules-13-00926-f012:**
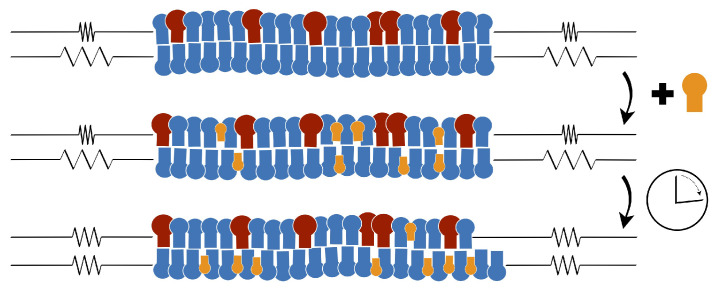
Relaxation of leaflet tensions in a lipid bilayer which is composed of two phospholipids and cholesterol (orange): The top row shows a bilayer membrane with two lipid components (blue and red) that do not undergo flip-flops from one leaflet to the other. The bilayer is tensionless in the sense that the bilayer tension $\Sigma =\Sigma_{ll}+\Sigma_{ul}$ is (close to) zero. However, the upper leaflet of the bilayer is compressed by a negative leaflet tension $\Sigma_{ul}<0$ whereas the lower leaflet is stretched by a positive leaflet tension $\Sigma_{ll}>0$, as indicated by the schematic springs on the left and on the right of the bilayer. As a third component, cholesterol (orange) is added to both leaflets so that they initially contain the same number of cholesterol molecules, as depicted in the middle row. After the cholesterol has been redistributed by flip-flops, both leaflets have attained a tensionless state as indicated by the relaxed springs. The cartoon at the bottom also indicates that the two tensionless leaflets typically differ in the preferred areas that they would assume in a symmetric bilayer [27].

**Figure 13 biomolecules-13-00926-f013:**
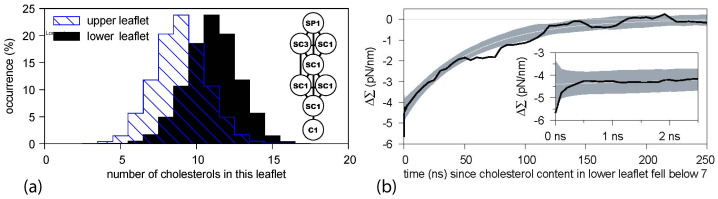
Simulation data for an asymmetric bilayer with 66 POPCs and 24 GM1s in the upper leaflet, 87 POPCs in the lower leaflet, as well as 20 cholesterols that undergo frequent flip-flops between the two leaflets: (**a**) Different distributions for the number of cholesterol molecules in the two bilayer leaflets, with an average number of 9 cholesterols in the upper leaflet (striped blue) and of 11 cholesterol in the lower leaflet (black). The inset shows a sketch of the Martini cholesterol model [44]; and (**b**) Time-dependent relaxation (black solid line) of the stress asymmetry $\Delta \Sigma =\Sigma_{ul}-\Sigma_{ll}$ towards the state with tensionless leaflets and ΔΣ=0. After the first 500 ps, the relaxation curve is well fitted by a single exponential with a time constant of 55 ns. The inset displays the non-exponential behavior observed during the first 2.5 ns [27].

**Figure 14 biomolecules-13-00926-f014:**
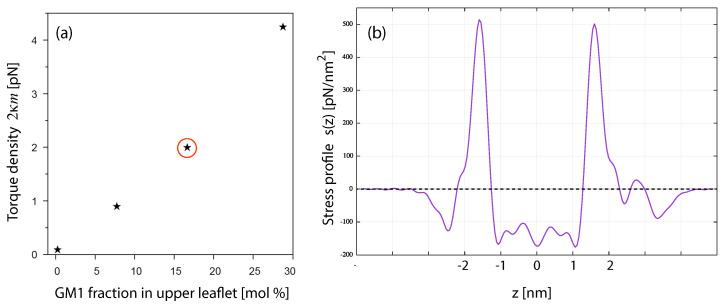
Asymmetric bilayers composed of POPC in the lower leaflet and a mixture of POPC and GM1 in the upper leaflet. The upper leaflet contains a fixed number of 100 lipid molecules while the lower leaflet contains a varible number of POPCs: (**a**) Torque density 2κm, corresponding to the negative first moment of the stress profile s(z) as in Equation (Equation 11), versus GM1 mole fraction in the upper leaflet. All bilayers have vanishing bilayer tension; and (**b**) Stress profile s(z) for a bilayer with 17 GM1 molecules and thus 17 mol% of GM1 in the upper leaflet and 98 POPCs in the lower leaflet, corresponding to the red encircled data point in (**a**). Even though the stress profile in (**b**) is clearly asymmetric, as follows by comparison of the two headgroup layers around z=±2.5 nm, the two leaflet tensions $\Sigma_{ll}$ and $\Sigma_{ul}$, which are obtained by integrating s(z) over negative and positive values of *z* as in Equation (Equation 3), are both close to zero [27].

**Figure 15 biomolecules-13-00926-f015:**
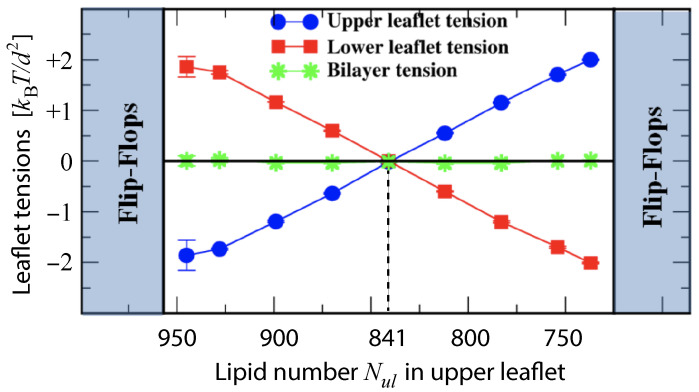
Upper leaflet tension $\Sigma_{ul}$ (blue) and lower leaflet tension $\Sigma_{ll}$ (red) versus lipid number $N_{ul}$ for constant total lipid number $N_{ul}+N_{ll}=1682$. The lipid number $N_{ul}$ is reduced by moving lipids from the upper to the lower leaflet, thereby increasing the upper leaflet tension and decreasing the lower one. Both leaflet tensions vanish for $N_{ul}=841$ (vertical dashed line) which defines the relaxed reference state of the planar bilayers. The green data display the bilayer tension $\Sigma =\Sigma_{ul}+\Sigma_{ll}$, which is close to zero, demonstrating that all bilayer states are OLT states. For $945\geq N_{ul}\geq 737$, the bilayer remains stable and the lipids do not undergo flip-flops from one leaflet to the other during the first 12.5 μs of the simulations. The midplanes of the OLT states were obtained via the CHAIN protocol [30].

**Figure 16 biomolecules-13-00926-f016:**
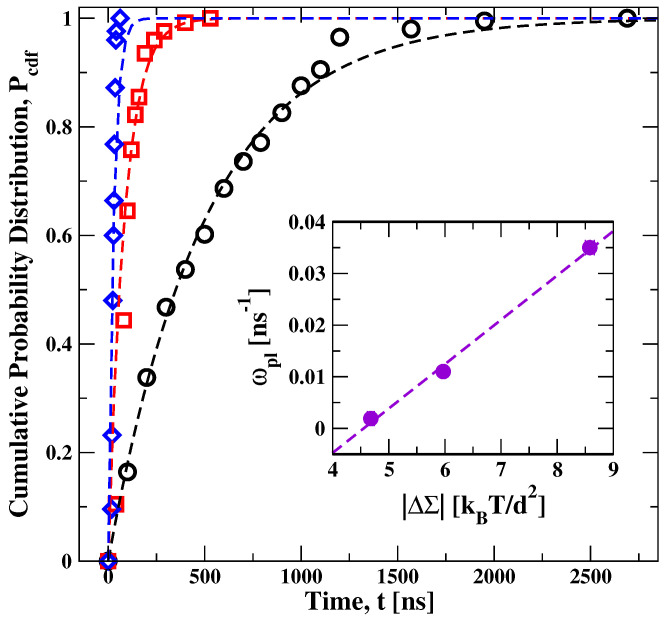
Cumulative distribution function $P_{\mathrm{cdf}}$ versus time *t*, for a planar bilayer with $N_{ul}$ lipids in the upper leaflet and $N_{ll}=1682-N_{ul}$ lipids in the lower leaflet. Three sets of data for $N_{ul}=986$ (black circles), $N_{ul}=1015$ (red squares), and $N_{ul}=1073$ (blue diamonds). These data sets are well fitted, using least squares, by an exponential distribution (broken lines) as in Equation (Equation 18), which involves only a single fit parameter, the flip-flop rate $\omega_{\mathrm{pl}}$. Each data set represents the outcome of more than 120 statistically independent simulations. Inset: Monotonic increase of the flip-flop rate $\omega_{\mathrm{pl}}$ with the absolute value $\left|\Delta \Sigma |=\right|\Sigma_{ul}-\Sigma_{ll}|$ of the stress asymmetry between the two leaflets [30].

**Figure 17 biomolecules-13-00926-f017:**
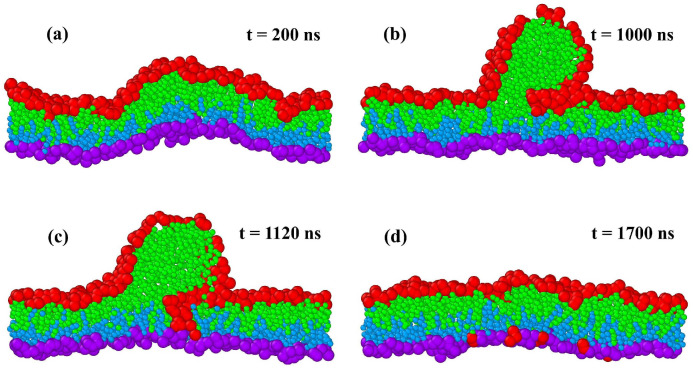
Structural instability and self-healing of a tensionless planar bilayer. At time t=0, the bilayer is initially assembled from $N_{ul}=986$ red-green lipids in the compressed upper leaflet and from $N_{ll}=696$ purple-blue lipids in the stretched lower leaflet: (**a**) At t=200 ns, the metastable bilayer bulges towards the upper leaflet; (**b**) At t=1000 ns, a globular micelle has been formed from about 100 red-green lipids that were expelled from the upper leaflet; (**c**) At t=1120 ns, red-green lipids move towards the stretched lower leaflet along the contact line between micelle and bilayer; and (**d**) This lipid exchange leads to a self-healing process of the bilayer that is completed at t=1700 ns. At this time point, 93 red-green lipids have moved from the upper to the lower leaflet. The restored bilayer remains stable without flip-flops until the end of the simulations at t=12.5μs [30].

**Figure 18 biomolecules-13-00926-f018:**
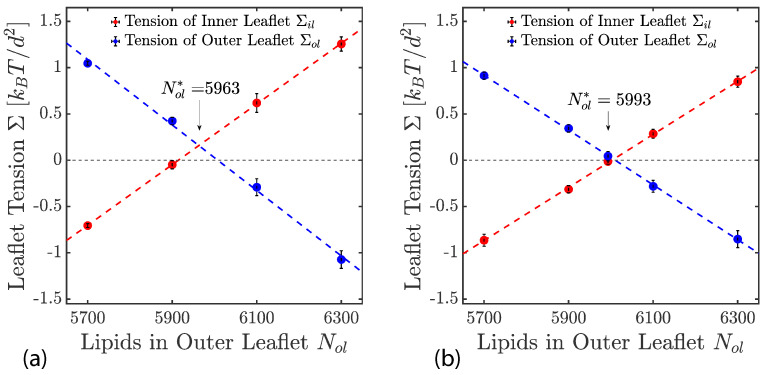
Leaflet tensions $\Sigma_{il}$ and $\Sigma_{ol}$ of the inner and outer leaflets for nanovesicles enclosed by bilayers with constant $N_{il}+N_{ol}=10,100$. The lipid number $N_{ol}$ is increased by reshuffling lipids from the inner to the outer leaflet: (**a**) Spherical vesicles with rescaled volume ν=1 in (**a**) and with $\nu =\nu_{0}<1$, corresponding to tensionless bilayers, in (**b**). The reference state with tensionless leaflets is estimated by linear intrapolation, which leads to $N_{ol}=N_{ol}^{*}=5963$ in (**a**) and to $N_{ol}=N_{ol}^{*}=5993$ in (**b**). In (**b**), all vesicle bilayers attain OLT states, for which the midsurface was calculated using the CHAIN protocol [28].

**Figure 19 biomolecules-13-00926-f019:**
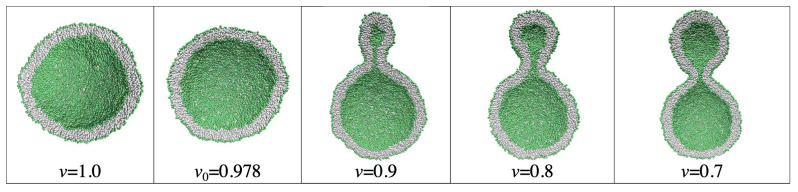
Shape transformations of a spherical vesicle which contains $N_{ol}=6300$ lipids in its outer and $N_{il}=3800$ lipids in its inner leaflet, induced by the reduction of the vesicle volume from ν=1 to ν=0.7, mimicking the experimental procedure of osmotic deflation. In the second panel with $\nu_{0}=0.978$, the vesicle has a tensionless bilayer, for which the outer leaflet is compressed with $\Sigma_{ol}=-0.99\;k_{B}T/d^{2}$ and the inner leaflet is stretched with $\Sigma_{il}=+0.99\;k_{B}T/d^{2}$. For rescaled volume ν≤0.9, the nanovesicle exhibits an out-bud with a closed membrane neck [28].

**Figure 20 biomolecules-13-00926-f020:**
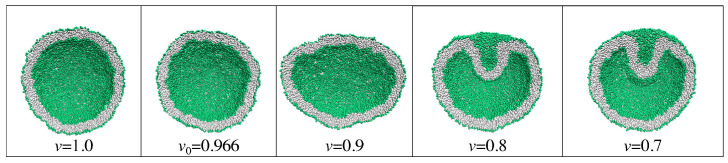
Shape transformations of a spherical nanovesicle with $N_{ol}=5700$ and $N_{il}=4400$, induced by the reduction of vesicle volume from ν=1 to ν=0.7. The second panel with $\nu =\nu_{0}=0.966$ represents a tensionless bilayer: the outer leaflet is stretched with $\Sigma_{ol}=+0.87\;k_{B}T/d^{2}$ whereas the inner leaflet is compressed with $\Sigma_{il}=-0.82\;k_{B}T/d^{2}$. For volume ν=0.7, the vesicle exhibits an in-bud with an open neck [28].

**Figure 21 biomolecules-13-00926-f021:**
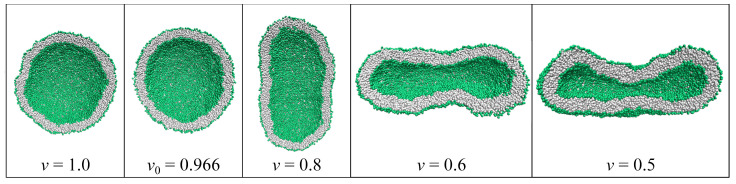
Shape transformations of a spherical vesicle with $N_{ol}=5963$ and $N_{il}=4137$ as driven by the reduction of its volume from ν=1 to ν=0.5. The second panel with $\nu_{0}=0.966$ displays the reference state of the bilayer, for which both leaflet tensions are close to zero. More precisely, for $\nu =\nu_{0}$, the outer leaflet is slightly stretched by the positive leaflet tension $\Sigma_{ol}=+0.03\;k_{B}T/d^{2}$ and the inner leaflet is slightly compressed by the negative leaflet tension $\Sigma_{il}=-0.02\;k_{B}T/d^{2}$. As the volume is further reduced. the vesicle attains a prolate shape for vs. = 0.8 and vs. = 0.7, an oblate or discocyte shape for vs. = 0.6, and a stomatocyte shape for vs. = 0.5 [28].

**Figure 22 biomolecules-13-00926-f022:**
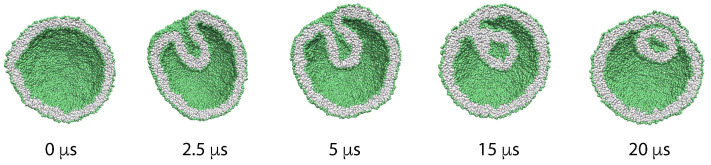
Time-dependent shape evolution of a nanovesicle with $N_{ol}=5500$ and $N_{il}=4600$ lipids. The vesicle has a spherical shape with volume ν=1 until time t=0μs, when the vesicle volume is reduced to ν=0.8. After this volume reduction, the vesicle develops an in-bud with a membrane neck that is closed at t=5μs. The neck is cleaved at about t=15μs, thereby generating an interluminal daughter vesicle that adheres to the larger daughter vesicle. The two vesicles remain in this adhering state at least until t=40μs. [Simulations by Rikhia Ghosh].

**Figure 23 biomolecules-13-00926-f023:**
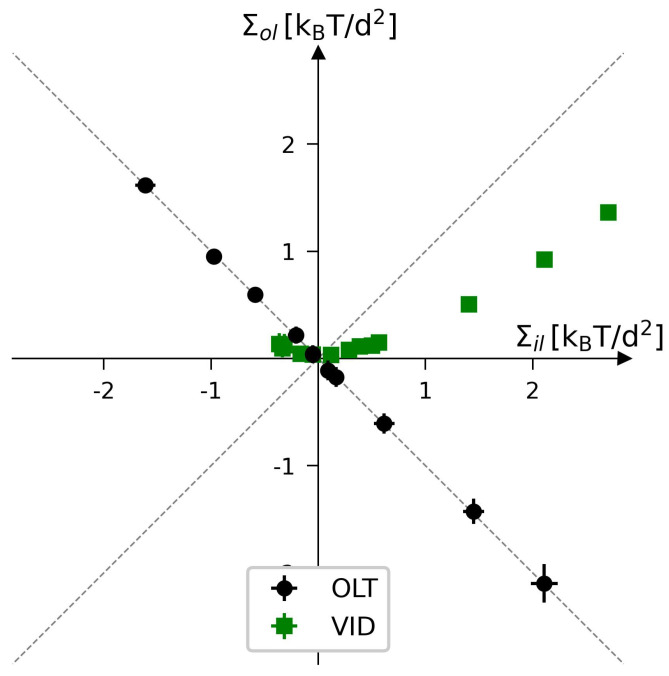
Leaflet tension space for the closed bilayers of nanovesicles with a total number of $N_{il}+N_{ol}=2525$ lipids in both leaflets. The two coordinates are the leaflet tensions $\Sigma_{il}$ and $\Sigma_{ol}$ in the inner and outer leaflets. Negative and positive leaflet tensions describe compressed and stretched leaflets. The reference state with tensionless leaflets, corresponding to $\Sigma_{il}=\Sigma_{ol}=0$, is obtained for a vesicle bilayer with $N_{il}=840$ lipids in the inner leaflet and $N_{ol}=1685$ lipids in the outer one. The black data represent elastic OLT deformations obtained from the reference state by reshuffling lipids from one leaflet to the other and adjusting the vesicle volume to obtain tensionless bilayers with $\Sigma =\Sigma_{il}+\Sigma_{ol}=0$. The green data represent the elastic deformations arising from changes in vesicle volume, corresponding to vesicle inflation or deflation (VID). For all data shown here, the midsurface of the vesicle bilayer was determined by the VORON protocol [32].

**Figure 24 biomolecules-13-00926-f024:**
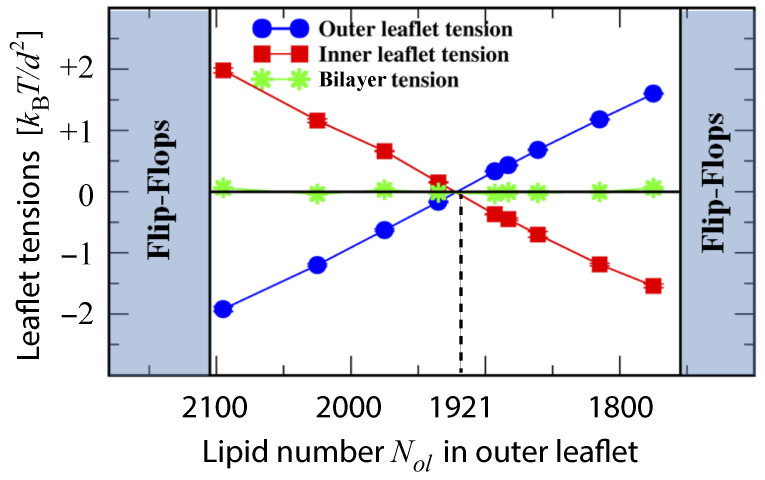
Outer leaflet tension $\Sigma_{ol}$ (blue) and inner leaflet tension $\Sigma_{il}$ (red) versus lipid number $N_{ol}$ for constant $N_{ol}+N_{il}=2875$. The lipid number $N_{ol}$ is reduced by moving lipids from the outer to the inner leaflet, thereby increasing the outer leaflet tension and decreasing the inner one. Both leaflet tensions vanish for $N_{ol}=1921$ (vertical dashed line) and $N_{il}=954$, which defines the relaxed reference state of the nanovesicles. The green data correspond to the bilayer tension $\Sigma =\Sigma_{ol}+\Sigma_{il}$, which is close to zero, demonstrating that all bilayer states are OLT states. During the run time of 12.5μs, we observed no flip-flops within the stability regime (white), corresponding to $2095\geq N_{ol}\geq 1775$. The left vertical line at $N_{ol}=2105$ represents the instability line at which the lipids start to undergo flip-flops from the compressed outer to the stretched inner leaflet. The right vertical line at $N_{ol}=1755$ represents the instability line at which the lipids start to undergo flip-flops from the compressed inner to the stretched outer leaflet. The vesicles have a diameter of 23.8d. The midsurface of the OLT states was obtained via the CHAIN protocol [30].

**Figure 25 biomolecules-13-00926-f025:**
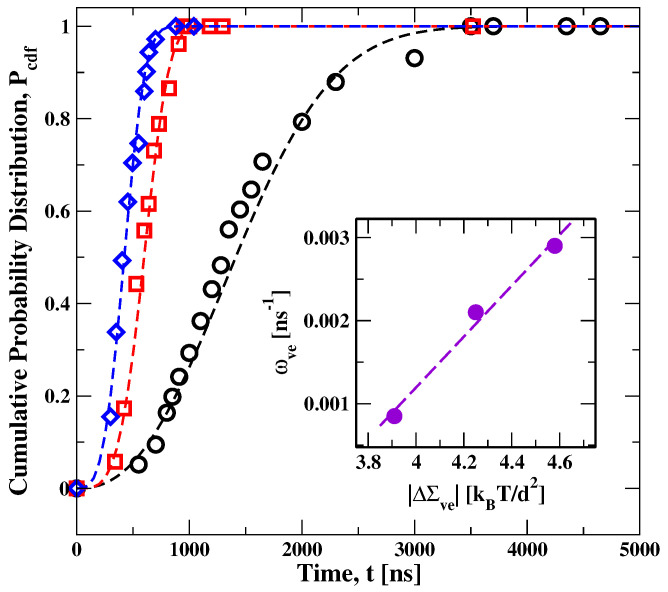
Cumulative distribution function $P_{\mathrm{cdf}}$ versus time *t* for flip-flops in tensionless bilayers of nanovesicles, assembled from $N_{ol}+N_{il}=2875$ lipids. Three sets of data are displayed with $N_{ol}=2105$ (black circles), $N_{ol}=2125$ (red squares), and $N_{ol}=2150$ (blue diamonds) lipids in the outer leaflet, which belong to the left instability regime in Figure 24. The three sets of data are well fitted, using least squares, to Weibull distributions (broken lines) as in Equation (Equation 27), which depend on two parameters, the shape parameter *k* and the rate parameter $\omega_{\mathrm{ve}}$. Each data set represents the outcome of at least 70 statistically independent simulations. Inset: Monotonic increase of the rate parameter $\omega_{\mathrm{ve}}$ with the absolute value $|\Delta \Sigma_{\mathrm{ve}}|$ of the stress asymmetry as defined by Equation (Equation 25) [30].

**Figure 26 biomolecules-13-00926-f026:**
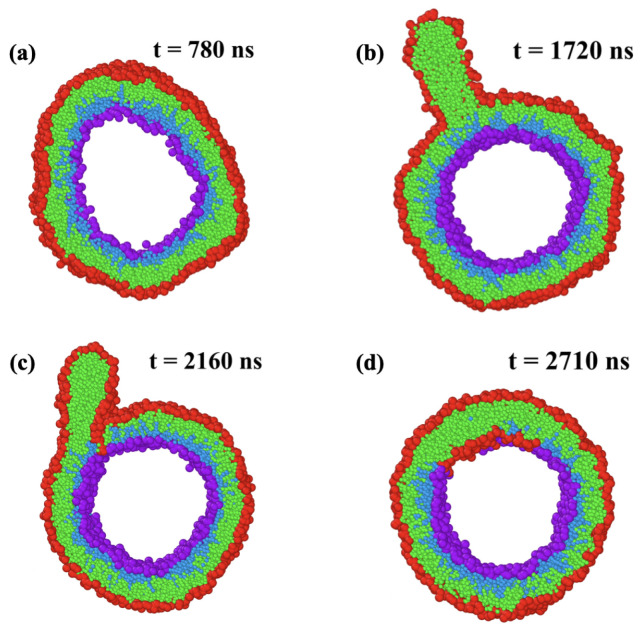
Structural instability and self-healing process of vesicle bilayer. At time t=0, the bilayer is assembled from $N_{ol}=2105$ and $N_{il}=770$ lipids and the vesicle volume is adjusted in such a way that the bilayer tension is close to zero, which leads to a compressed outer leaflet with negative leaflet tension $\Sigma_{ol}=-1.97\;k_{B}T/d^{2}$: (**a**) At t=780ns, the compressed outer leaflet exhibits some kinks; (**b**) At t=1720ns, a cylindrical micelle has been formed from about 180 red-green lipids that were expelled from the outer leaflet; (**c**) At t=2160ns, lipids move towards the stretched inner leaflet along the contact line between micelle and bilayer; and (**d**) At t=2710ns, the self-healing process via stress-induced lipid exchange has been completed and 111 red-green lipids have moved to the inner leaflet. The restored bilayer undergoes no further flip-flops until the end of the simulations.

**Figure 27 biomolecules-13-00926-f027:**
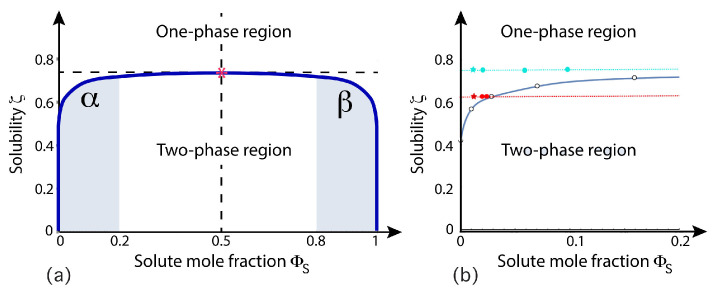
Phase diagram for a binary mixture of water and solute molecules as a function of solute mole fraction $\Phi_{S}$ and solubility ζ of the solute molecules in water [29]: (**a**) Global phase diagram for $0\leq \Phi_{S}\leq 1$. The phase diagram is mirror symmetric with respect to the dashed vertical line at $\Phi_{S}=1/2$, which implies horizontal tie lines. The critical demixing point (red star) with coordinates $\left(\Phi_{S},\zeta \right)=\left(1/2,0.746\right)$ is located at the crossing point of the dashed vertical line and the binodal line (dark blue). The binary mixture forms a uniform phase above the binodal line and undergoes phase separation into a water-rich phase α with $\Phi_{S}<0.5$ and a solute-rich phase β with $\Phi_{S}>0.5$; and (**b**) Phase diagram for $0\leq \Phi_{S}\leq 0.2$ corresponding to the left blue-shaded region of panel a. The green points correspond to good solvent conditions with ζ=25/32=0.781 above the critical point, the red points to poor solvent conditions with ζ=25/40=0.625 below the critical point. Replacing the solubility by temperature leads to a very similar phase diagram.

**Figure 28 biomolecules-13-00926-f028:**
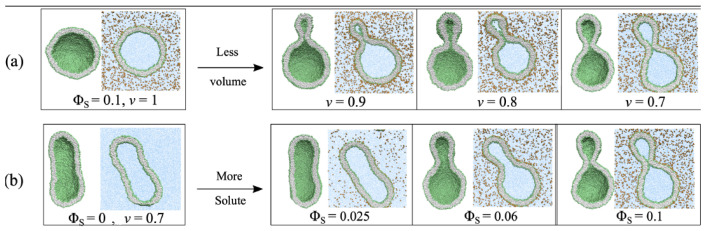
Budding of nanovesicles for good solvent condition: (**a**) A spherical vesicle with volume ν=1 is exposed to solute mole fraction $\Phi_{S}=0.1$ and subsequently deflated to volume ν=0.7; and (**b**) The same spherical vesicle is first deflated to volume ν=0.7 and then exposed to an increasing solute mole fraction from $\Phi_{S}=0$ up to $\Phi_{S}=0.1$. Both protocols lead to the same final dumbbell morphology with a closed membrane neck, corresponding to the rightmost snapshots. The solute-induced budding process in panel b is reversible as demonstrated by decreasing and increasing the solute concentration several times [29].

**Figure 29 biomolecules-13-00926-f029:**
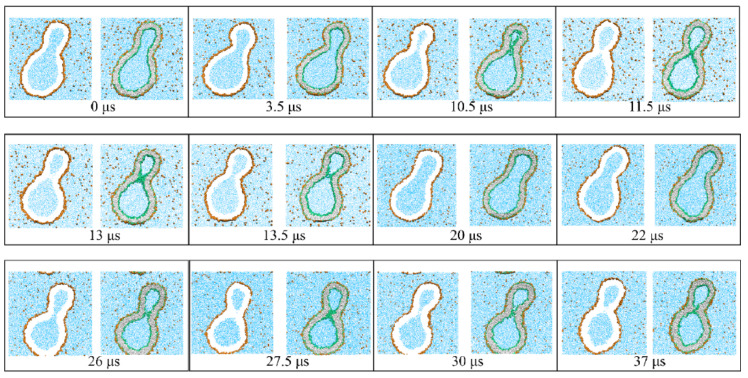
Time evolution of an individual nanovesicle exposed to solute concentration $\Phi_{S}=0.025$ close to the binodal line, which is located at $\Phi_{S}^{*}=0.0275$ for solubility ζ=0.625: At time t=0μs, the volume of the vesicle is reduced from ν=0.80 to ν=0.75 and then kept constant at this latter value. The vesicle responds to this volume decrease by closing and reopening its neck in a recurrent fashion. This recurrent process of neck closure and neck opening persists for at least 90μs, see next Figure 30, which displays the time evolution of the neck diameter [29].

**Figure 30 biomolecules-13-00926-f030:**
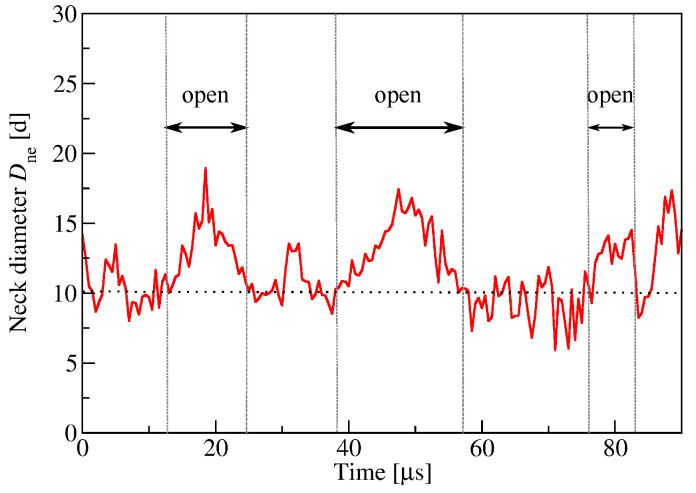
Time evolution of outer neck diameter $D_{\mathrm{ne}}$ corresponding to the time-lapse snapshots of the nanovesicle in Figure 29 for volume ν=0.75. The membrane neck repeatedly closes and opens up again. We consider the neck to be closed for $D_{\mathrm{ne}}≲10\;d$ and to be open for $D_{\mathrm{ne}}≳10\;d$. where $D_{\mathrm{ne}}≃10\;d$ represents the outer diameter of the closed neck [29].

**Figure 31 biomolecules-13-00926-f031:**
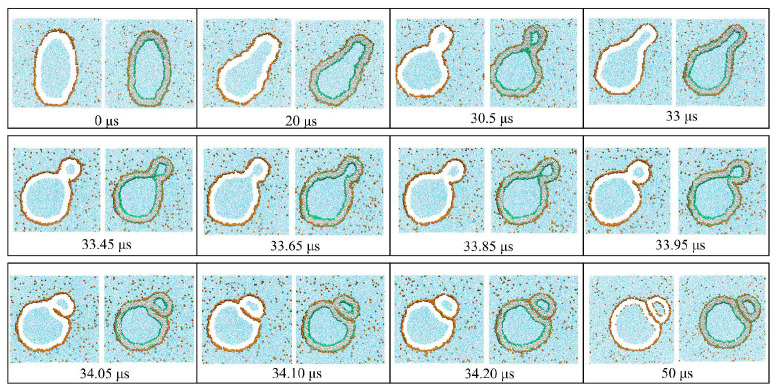
Nanovesicle exposed to exterior solution with solute concentration $\Phi_{S}=0.026$ close to the binodal line, which is located at $\Phi_{S}^{*}=0.0275$ for solubility ζ=0.625: At time t=0μs, we start from a prolate shape with volume ν=0.9 and reduce the vesicle volume to ν=0.85, which is then kept fixed for all later times. The vesicle transforms into a dumbbell shape with a membrane neck that is closed at t=30.5μs and reopens fast within about one μs. The cross-sectional snapshot at t=33.85μs indicates that the geometry of the neck has changed by adhesion of two membrane segments close to the neck. This adhesion is mediated by a layer of adsorbed solutes (orange dots), which generate a rapidly expanding contact area until the neck is cleaved and the vesicle is divided into two daughter vesicles. These two separate vesicles continue to adhere to each other via an intermediate adsorption layer of solutes and form a stable morphology for later times t≥34.20μs. The time dependence of the neck diameter and the growing contact area are displayed in Figure 32 [29].

**Figure 32 biomolecules-13-00926-f032:**
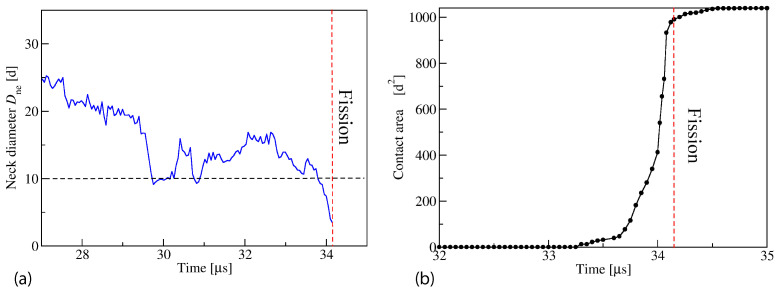
Time evolution of (**a**) neck diameter $D_{\mathrm{ne}}$ and (**b**) contact area between the two adhering membrane segments close to the neck, corresponding to the time series of snapshots in Figure 31. In this example, the neck was cleaved and the nanovesicle divided after a fission time of 34.15 μs. Note that the fission process involves a free energy barrier that has to be overcome by thermal noise. Therefore, the fission time varies from one fission event to another [29].

**Figure 33 biomolecules-13-00926-f033:**
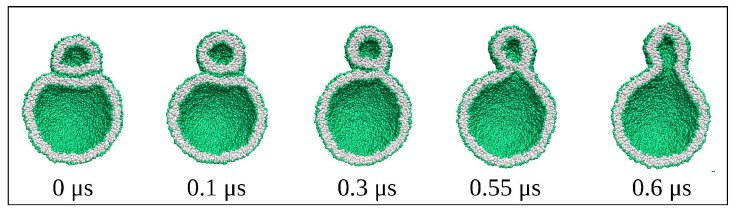
Starting from two adhering daughter vesicles as displayed in the last snapshot of Figure 31, we remove the solute molecules from the solution by transmuting all solute beads into water beads at time t=0μs. As a result of this removal, a fusion pore starts to appear in the contact zone after 0.3μs and leads to complete fusion of the two adhering daughter vesicles into a prolate vesicle within less than 0.6μs. [Simulations by Rikhia Ghosh].

**Figure 34 biomolecules-13-00926-f034:**
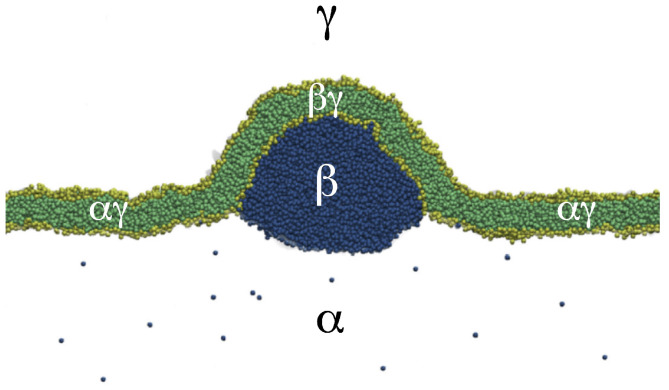
Partial engulfment of a condensate nanodroplet (β, dark blue) by a planar bilayer, consisting of lipids with yellow headgroups and green lipid tails as studied by molecular dynamics simulations [26]. The αβ interface between the droplet and the liquid bulk phase α forms a contact line with the bilayer which partitions this bilayer into a βγ segment in contact with the β droplet and into an αγ segment exposed to the α phase.

**Figure 35 biomolecules-13-00926-f035:**
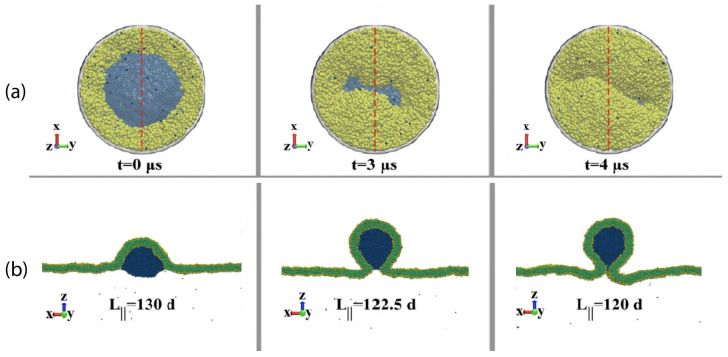
Formation of a non-circular, tight-lipped membrane neck generated by a nanodroplet (dark blue) that adheres to a planar bilayer [26]. This process was induced by a time-dependent reduction of the lateral size $L_{\|}$ of the simulation box, keeping the box volume fixed: (**a**) Bottom views of circular bilayer segments (yellow) around the αβ interface (blue) of the β droplet, separated by the contact line which is circular at time t=0μs, strongly non-circular after t=3μs, and has closed into a tight-lipped shape after t=4μs; and (**b**) Side views of the same bilayer-droplet morphology, with perpendicular cross-sections through membrane (green) and droplet (blue) taken along the red dashed lines in panel (**a**). The non-circular shape of the membrane neck is caused by the negative line tension of the contact line and prevents membrane fission. Same color code as in Figure 34.

**Figure 36 biomolecules-13-00926-f036:**
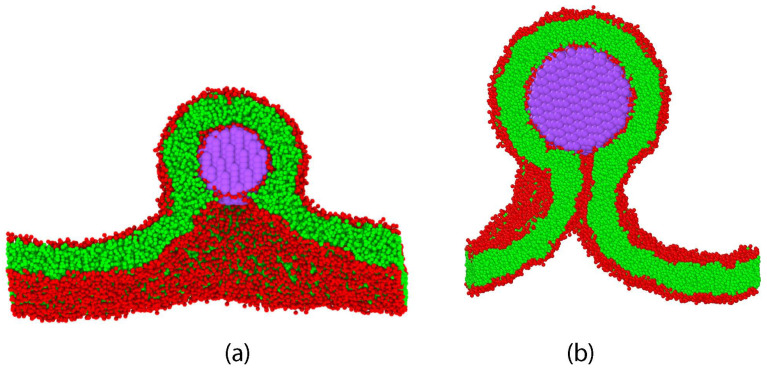
Engulfment of spherical nanoparticles by planar bilayers: (**a**) Partial engulfment of a nanoparticle with radius 5d; [25] and (**b**) Complete engulfment of particle with radius 8d. [Simulations by Aparna Sreekumari]. The complete engulfment process leads to a tight-lipped membrane neck as follows from the bottom views of the same bilayer-nanoparticle system displayed in Figure 37. The bilayer in panel b contains 59×59 lipids (red-green) in each leaflet. The bilayer edges indicate the lateral size of the simulation box.

**Figure 37 biomolecules-13-00926-f037:**
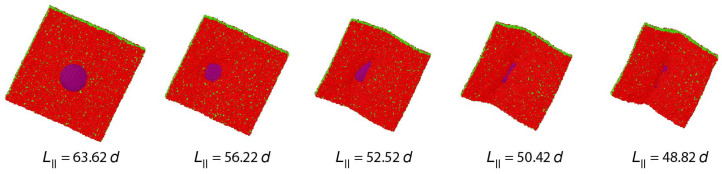
Engulfment of a spherical nanoparticle (purple) by a symmetric planar bilayer: The radius of the nanoparticle is equal to 8d. The planar bilayer contains 59×59=3481 lipids (red-green) in each leaflet. The different snapshots show the bottom view of the equilibrated bilayer-particle system for decreasing lateral sizes $L_{\|}$ of the simulation box. The initial snapshot displays the system for $L_{\|}=63.62\;d$, with a circular contact line between particle and bilayer. The contact line becomes strongly non-circular for $L_{\|}=52.52\;d$ and is hardly visible in the last snapshot for $L_{\|}=48.82\;d$. For $L_{\|}=43.62\;d$, corresponding to the side view in Figure 36b, the nanoparticle is no longer visible from the bottom. [Simulations by Aparna Sreekumari].

**Figure 38 biomolecules-13-00926-f038:**
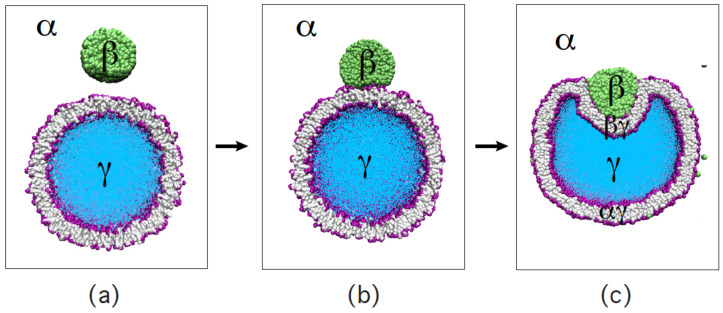
Partial engulfment of a condensate droplet (green) by the lipid bilayer (purple-grey) of a nanovesicle. The vesicle encloses the aqueous solution γ (blue). Both the nanodroplet and the nanovesicle are immersed in the aqueous bulk phase α (white): (**a**) Initially, the droplet is well separated from the vesicle which implies that the outer leaflet of the bilayer is only in contact with the α phase; (**b**) When the droplet is attracted towards the vesicle, it spreads onto the lipid bilayer, thereby increasing its contact area with the vesicle bilayer; and (**c**) Partial engulfment of the droplet by the membrane after the vesicle-droplet couple has relaxed to a new stable state. The contact area between bilayer and β droplet defines the βγ segment of the bilayer membrane whereas the rest of the bilayer represents the αγ segment still exposed to the α phase [31].

**Figure 39 biomolecules-13-00926-f039:**
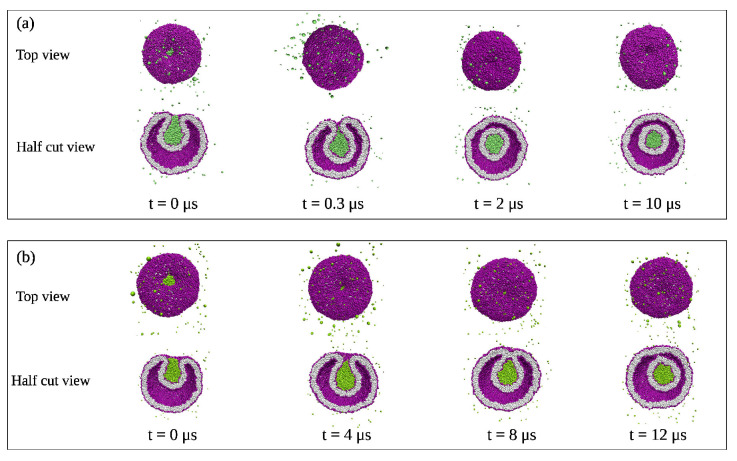
Complete axisymmetric engulfment of condensate droplets (green) followed by the division of the nanovesicle (purple-grey) into two nested daughter vesicles: (**a**) Vesicle bilayer with $N_{ol}=5400$ lipids in the outer and $N_{il}=4700$ lipids in the inner leaflet. At time t=0, the droplet is partially engulfed by the vesicle membrane, which forms an open membrane neck. At t=0.3, the neck closes and the droplet becomes completely engulfed. The neck undergoes fission at t=2μs, thereby generating a small intraluminal vesicle around the droplet; and (**b**) Vesicle bilayer with $N_{ol}=5500$ lipids in the outer and $N_{il}=4600$ lipids in the inner leaflet. The membrane neck now closes at t=4μs and undergoes fission at t=9μs, again generating a small intraluminal vesicle around the droplet. In both panels (**a**) and (**b**), the vesicle volume is equal to ν=0.6 during the whole endocytic process [31].

**Figure 40 biomolecules-13-00926-f040:**
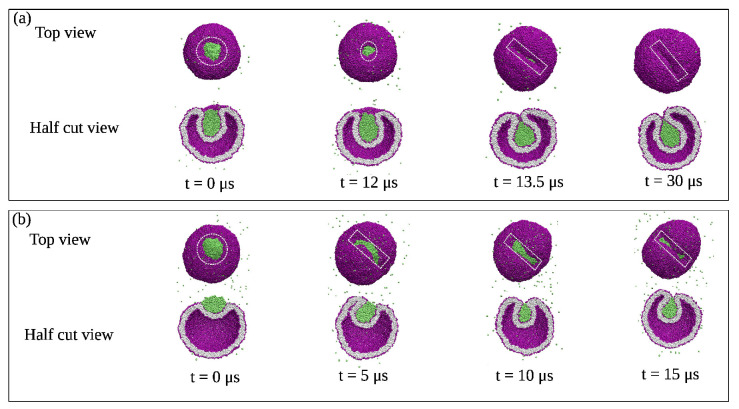
Complete non-axisymmetric engulfment of condensate droplets (green), which impedes the division of the nanovesicle (purple-grey): (**a**) Vesicle bilayer with $N_{ol}=5700$ lipids in its outer and $N_{il}=4400$ lipids in its inner leaflet. At time t=0, both the vesicle with volume ν=0.7 and the partially engulfed droplet (green) are axisymmetric, a morphology that persists until t=12μs, see white dashed circles around the contact lines. At t=12μs, we reduce the vesicle volume from ν=0.7 to ν=0.6, which leads to complete engulfment of the droplet via non-axisymmetric shapes. The broken rotational symmetry is directly visible from the strongly non-circular and highly elongated contact line, see white dashed rectangles around the contact lines at t=13.5μs and t=30μs; and (**b**) Vesicle bilayer with $N_{ol}=5963$ lipids in its outer and $N_{il}=4137$ lipids in its inner leaflet. Now, the vesicle volume is kept at the constant value ν=0.7. At t=0, the droplet is partially engulfed by the vesicle membrane with an axisymmetric contact line, see white dashed circle. The axial symmetry is broken at t=5μs, as follows from the strongly non-circular and highly elongated contact lines for t≥5μs, see white dashed rectangles [31].

**Figure 41 biomolecules-13-00926-f041:**
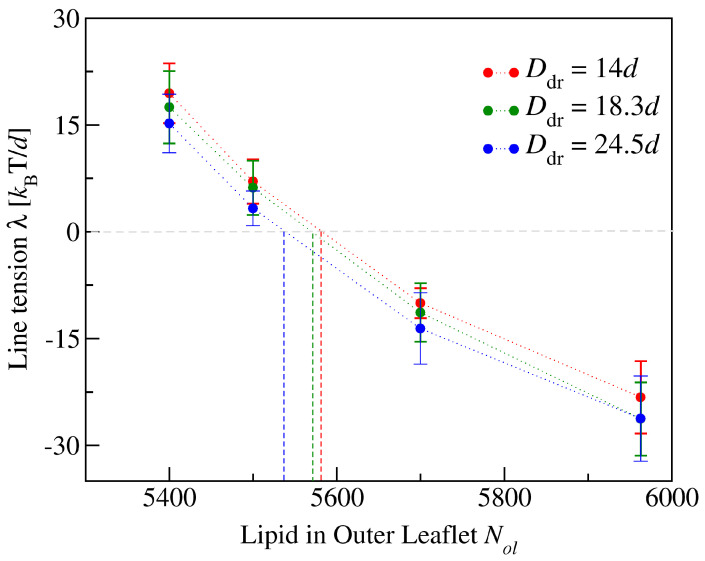
Line tension λ of contact line between droplet and vesicle membrane as a function of lipid number $N_{ol}$ in the outer leaflet for constant total lipid number $N_{ol}+N_{il}=10,100$, corresponding to OLT states of the vesicle bilayers. The line tension is calculated for droplets with three different diameters $D_{\mathrm{dr}}$, see inset, via the force balance relationship in Equation (Equation 30). As we increase $N_{ol}$, the line tension undergoes a transition from positive to negative values. The line tension is positive for $N_{ol}=5400$ and 5500, for which the whole engulfment process remains axisymmetric as in Figure 39. On the other hand, for $N_{ol}=5700$ and 5963, the line tension has a negative value and leads to the formation of a tight-lipped membrane neck as in Figure 40. The dashed vertical lines provide estimates for the lipid numbers $N_{ol}=N_{ol}^{\left[0\right]}$, at which the line tension vanishes. The numerical values of $N_{ol}^{\left[0\right]}$ vary from $N_{ol}^{\left[0\right]}=5582$ for the smallest droplets to $N_{ol}^{\left[0\right]}=5538$ for the largest droplets [31].

**Figure 42 biomolecules-13-00926-f042:**
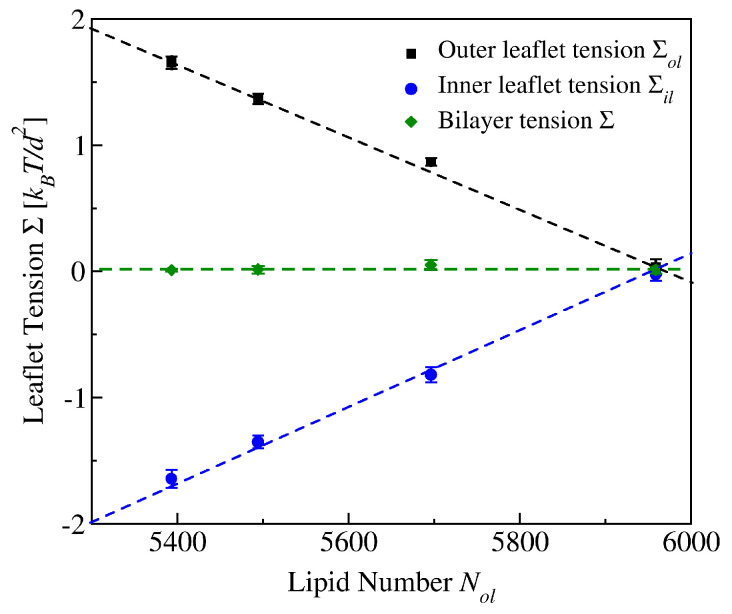
Stress asymmetry of tensionless bilayers for spherical nanovesicles with volume $\nu =\nu_{0}$: Leaflet tensions $\Sigma_{ol}$ (black squares) and $\Sigma_{il}$ (blue circles) of the outer and inner leaflet as a function of the lipid number $N_{ol}$ assembled in the outer leaflet which implies the lipid number $N_{il}=10,100-N_{ol}$ in the inner leaflet. For $5400\leq N_{ol}<5963$, the outer leaflet is stretched by the positive tension $\Sigma_{ol}$ whereas the inner leaflet is compressed by the negative tension $\Sigma_{il}$. Both leaflet tensions vanish for $N_{ol}=5963$. In all cases, the bilayer tension $\Sigma =\Sigma_{ol}+\Sigma_{il}$ (green diamonds) is close to zero [31].

**Figure 43 biomolecules-13-00926-f043:**
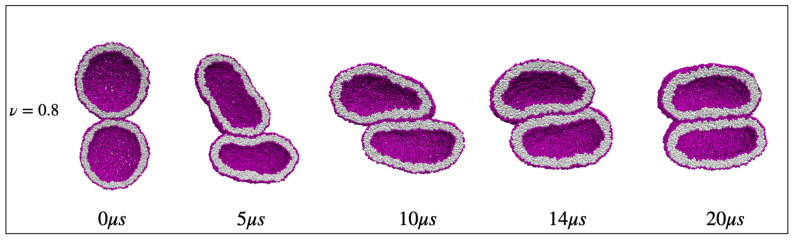
Two identical vesicles that come into contact adhere to each other when their outer leaflets are stretched provided the outer leaflet tensions remain below a certain threshold value. Each vesicle contains $N_{il}=4400$ lipids in its inner and $N_{ol}=5700$ lipids in its outer leaflet. Up to time t=0μs, each vesicle has the volume ν=0.8 and its outer leaflet is stretched by the positive leaflet tension $\Sigma_{ol}=+0.87\;k_{B}T/d^{2}$ whereas its inner leaflet is compressed by the negative leaflet tension $\Sigma_{il}=-0.82\;k_{B}T/d^{2}$. At t=0μs, the volume of each vesicle is reduced from ν=0.8 to ν=0.7. The vesicles then transform into oblate shapes that form a large contact area as shown in the last snapshot at t=20μs. [Simulations by Rikhia Ghosh].

**Figure 44 biomolecules-13-00926-f044:**
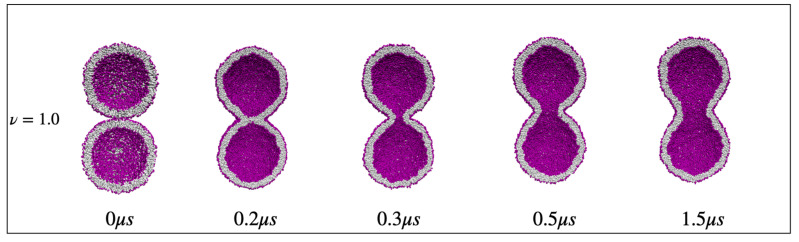
Two identical vesicles that come into contact undergo fusion when their outer leaflets are stretched by a sufficiently large leaflet tension and their bilayers experience a sufficiently large stress asymmetry. Initially, each vesicle contains $N_{il}=4500$ lipids in its inner and $N_{ol}=5600$ lipids in its outer leaflet. Thus, compared to the initial state in Figure 43, 100 lipids have been moved from the outer to the inner leaflet, thereby increasing the tension in the outer leaflet to $\Sigma_{ol}=+1.02\;k_{B}T/d^{2}$ whereas the inner leaflet tension is $\Sigma_{il}=-1.02\;k_{B}T/d^{2}$. Each vesicle has the initial volume ν=1 which remains unchanged during the whole process, in contrast to the adhesion process displayed in Figure 43. At time t=0, the two vesicles are brought into contact and promptly undergo fusion within 0.3μs without any volume reduction. [Simulations by Rikhia Ghosh].

## Data Availability

Not applicable.

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
