# Peer review of "Leaflet Tensions Control the Spatio-Temporal Remodeling of Lipid Bilayers and Nanovesicles"

_biomolecules, 2023, doi:10.3390/biom13060926_

Round 1

Reviewer 1 Report

Overall, this is a very timely and valuable review on membrane remodeling.  I have a number of comments that’s aimed at further improving the review paper. 

1. The review almost exclusively focuses on the authors’ own work.  This is justified because this body of work of the authors represents significant advances in an area of great current interest.  However, a brief summary of other studies will present a more balanced view of the literature.  For example, while the work reviewed here is based on coarse-grained simulations, significant progress has been made recently using all-atom simulations, in particular on membrane association of intrinsically disordered proteins [PMID: 36959324PMID: 36710643], which can lead to protein aggregation or condensate formation and to membrane curvature formation. 

2. The review summarizes many interesting computational predictions, but the authors should speculate on (a) how to verify these predictions by experiments; (b) use their formalisms to help measure membrane properties.  In particular, the last section, “Summary and Outlook”, is very long on summary and very short on outlook.  I’d ask the authors to cut down on summary and expand significantly on outlook. 

3. Minor corrections:

p. 2: nanovesicles -- they authors should point out early the more common term is SUV, rather than at a later point (p. 3)

p. 3: “technical details ... will not be discussed here” — actually some details will be very useful for the reader to appreciate the results.  E.g., the authors mention both DPD GROMACS+Martin. Is the GROMACS work (ref. 27) based on a type of simulations (e.g., MD plus thermostat for T regulation) that’s different from DPD?  If so, does the difference in dynamics simulation (in particular thermostat) affect the results?  Did ref. 27 use the Martini CG model, but the other papers use a different CG model?  If so, do the CG models affect conclusions?

p. 3, line 110, extraneous “d”

p. 9, lines 341 and 342 — some cross referencing is good, but here the cross referencing interrupts the flow of the text.

p. 12, Fig.7 — 3 colors are displayed but w/o explanation

p. 14, Voronoi volumes — are Voronoi volumes around water beads also needed in order to define V_lW and V_uW?  If so, state as such.

p. 15, the (Σll,Σul) = (0,0state — how was this state attained?  By varying the cross sectional area in a trial and error fashion?  This needs elaboration.

p. 24, Eq 17 and line 696 — P_exp should be P_CDF

p. 27, Fig. 17b — v0 values need to be stated.  How are the OLT states achieved here?  By varying v0?

p. 29, line 810 — resulting => initial

p. 31, Eq 25 — might be worth pointing out that this reduces to Eq 10 upon setting M=0

p. 34, Eq 26 — need to point out this is more commonly known as the stretched exponential

p. 37, Fig. 27b, first panel — “ΦS = 0.1” should probably be “ΦS = 0

p. 38, Fig. 31 needs to be cited in the text, not just in a figure caption

Author Response

First, it is confusing that the heading mentions Reviewer 1 but the report is actually the report of Reviewer 3. 

Second, my detailed point-by-point responses to Reviewer 3 are contained in the uploaded document Responses_to_Reviewer-3.pdf 

Reviewer 2 Report

The review entitled "Leaflet Tensions Control the Spatio-Temporal Remodeling of Lipid Bilayers and Nanovesicles" by Lipowsky and coauthors, provides a thorough overview of modern lipid membrane research. The work is well structured and worthy of publication.

As a reader, I would appreciate an, even brief, introduction to the Gauss-Bonnet theorem, which I believe is essential to elucidate remodeling and transformations that can take place in lipid membranes. Section 2.3 would be the best place to introduce the description.

Author Response

According to my counting, this is the review of Reviewer 1. 

My detailed responses to the report by Reviewer 1 are in the uploaded document

"Responses_to_Reviewer-1.pdf"

Reviewer 3 Report

The work by Lipowsky et al. is a well-constructed, described and written research of an importance to the field.

Authors wrote that “the main purpose of this review is to advertise and promote the concept of leaflet tensions, a new concept that has been recently introduced and further developed by our group.”

In my opinion, they have performed their task. The review is very detailed and comprehensive and can be interesting for scientists working in this field.

Several small remarks:

Stress profile is one of the basic parameters of this work. How is it calculated?

There is technical defect on the page 37, lines 1014-1016.

In my opinion, because most calculations were performed with CGMD, a short paragraph concerning the shortcomings of CGMD and the Martini force field compared with all-atom MD would be useful. It could give a more realistic picture of the possibilities and disadvantages of the method for calculating such characteristics as bilayer tension, compressibility and so on.

Author Response

According to my files, this is the report of Reviewer 2. 

My point-by-point responses to this report are contained in the uploaded document "Responses_to_Reviewer-2.pdf"
